# A Two-Layer Framework for Joint Online Configuration Selection and Admission Control

Owen Shen [*1]   Haoran Xu [*2]   Yinyu Ye [2]   Peter Glynn [2]   Patrick Jaillet [1]

## Abstract

We study online configuration selection with admission control problem, which arises in LLM serving, GPU scheduling, and revenue management. In a planning horizon with $T$ periods, we consider a two-layer framework for the decisions made within each time period. In the first layer, the decision maker selects one of the $K$ configurations (ex. quantization, parallelism, fare class) which induces distribution over the reward-resource pair of the incoming request. In the second layer, the decision maker observes the request and then decides whether to accept it or not. Benchmarking this framework requires care. We introduce a **switching-aware fluid oracle** that accounts for the value of mixing configurations over time, provably upper-bounding any online policy. We derive a max-min formulation for evaluating the benchmark, and we characterize saddle points of the max-min problem via primal-dual optimality conditions linking equilibrium, feasibility, and complementarity. This guides the design of **SP-UCB–OLP** algorithm, which solves an optimistic saddle point problem and achieves $\tilde{O}(\sqrt{KT})$ regret.

## 1. Introduction

Many resource-constrained systems require *two sequential decisions* at each time step. First, the decision maker selects a *system configuration* that determines the type of incoming request. Second, after observing the request's characteristics, the decision maker makes an *admission decision*—whether to accept or reject the request, subject to a cumulative resource budget. Neither decision alone captures the full problem: the configuration shapes what the decision maker will see, and the admission control determines what resources are consumed. It is the *coupling* of both decisions that creates the central challenge.

This two-layer structure arises naturally in several domains. In LLM serving, the system first selects a serving configuration (quantization level, batching strategy) (Yu et al., 2022; Kwon et al., 2023), then observes the incoming request's prompt length and estimated compute cost, and finally decides whether to admit the job under its current resource budget. In cluster scheduling, the system first chooses a parallelism configuration (Gu et al., 2019; Narayanan et al., 2020; Qiao et al., 2021), then observes the job's resource footprint, and decides whether to schedule it. In revenue management, the firm first selects a fare class (Gallego & van Ryzin, 1994; Talluri & van Ryzin, 2004), then observes a customer's willingness-to-pay, and decides whether to accept the booking. In all cases, the first decision (configuration) is made *before* the request is revealed, while the second decision (admission) is made *after* observing the request—a **select–observe–admit** protocol.

We formalize this protocol as follows. There are $K$ *configurations*, each inducing a distribution over reward–resource pairs $(r, \mathbf{a})$. At each period, the decision maker (1) **selects** a configuration $\theta_t$, (2) **observes** the reward–resource pair $(r_t, \mathbf{a}_t) \sim \mathcal{D}_{\theta_t}$, and (3) **admits or rejects** the request subject to a cumulative budget $\mathbf{B}$. All decisions are irrevocable. Two existing frameworks each capture one layer of this problem but not both. Bandits with knapsacks (BwK) (Badanidiyuru et al., 2018) models configuration selection under budget constraints, but uses *sample-and-commit*: pulling an arm immediately consumes resources with no opportunity to reject after observing the outcome. Online linear programming (OLP) (Agrawal et al., 2014; Li & Ye, 2022) models observation-based admission—the decision maker sees $(r_t, \mathbf{a}_t)$ before committing resources—but assumes a *fixed* request distribution with no configuration selection. Our framework unifies both: configuration selection from BwK with observation-based admission from OLP.

**Main Contributions.**

*Equal contribution   [1]Operations Research Center, Massachusetts Institute of Technology, Cambridge, MA, USA [2]Department of Management Science and Engineering, Stanford University, Stanford, CA, USA. Correspondence to: Owen Shen <owenshen@mit.edu>.

*Proceedings of the 43$^{rd}$ International Conference on Machine Learning*, Seoul, South Korea. PMLR 306, 2026. Copyright 2026 by the author(s).

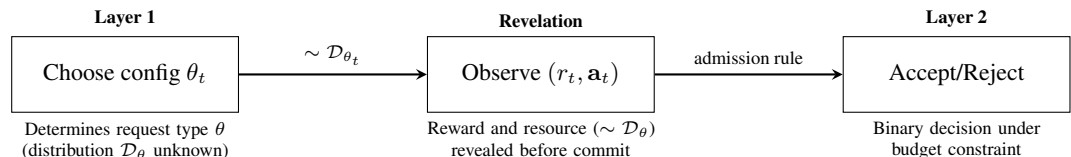

*Figure 1.* Two-layer decision structure with data revelation.

1. **Switching-aware fluid oracle.** Designing an appropriate oracle is essential in solving an online decision making problem. An oracle solves a so-called offline optimization problem whose optimal value serves as a benchmark for evaluating the performance of feasible online policies. A natural oracle of our problem is the fixed configuration oracle, i.e., fix a single best configuration and solve the offline resource allocation problem of the selected configuration. The offline resource allocation problem is given in (Li & Ye, 2022), and its optimal value depends on the reward-resource distribution. The best configuration in this natural oracle refers to the one whose reward-resource distribution gives the highest optimal value of the offline resource allocation problem. However, this oracle which is based on the naive "best-response" interpretation of our problem fails to upper bound the optimal online policy when there are heterogeneous resources. In those situations, a feasible online policy that switches among configurations may have higher total reward than the fixed configuration oracle by exploiting complementary budgets (Example 1).

   In this paper, we propose a *switching-aware* oracle which provides a valid upper bound on all feasible online policies (Theorem 4). This eliminates the "beating the oracle" pathology of the fixed-configuration oracle.

2. **Primal-dual characterization of the oracle.** We derive an equivalent max-min formulation of the offline optimization problem in the switching-aware oracle by duality. We build the connection between the saddle points of the max-min problem and the primal optimal solutions of the offline optimization problem via threshold rules, feasibility, and complementarity conditions (Theorems 2–3), which also corrects the naive "best-response" interpretation.

3. **Sublinear regret algorithm.** We design SP-UCB–OLP Algorithm by repeatedly solving the saddle points of an optimistic version of the max-min formulation of the offline optimization problem in the switching-aware oracle. We prove the regret of SP-UCB-OLP Algorithm is $\tilde{O}(\sqrt{KT})$ (Theorem 5)

where $T$ is the number of periods in the planning horizon and $K$ is the number of configurations. We also conduct numerical experiments to validate the regret upper bound.

## 2. Related Work

**Bandits with knapsacks (BwK) and stochastic packing.** BwK (Badanidiyuru et al., 2018) is a framework that deals with multi-armed bandit problems under budget constraints. Many variants of BwK problems including adversarial input (Immorlica et al., 2019), contextual features (Agrawal & Devanur, 2016; Agrawal et al., 2016), non-linear reward and constraint (Agrawal & Devanur, 2019) and non-stationary environments (Liu et al., 2022) have been studied in the literature. The LP-relaxation and dual-price perspective is also central in stochastic packing and online allocation (Agrawal et al., 2014; Agrawal & Devanur, 2015; Balseiro et al., 2020; Li & Ye, 2022). Although BwK and stochastic packing have components of both configuration selection and resource allocation, they use *sample-and-commit*: pulling an arm immediately consumes resources without the option to reject after observing the outcome. This makes our framework significantly different because it leverages *observe-then-decide* admission, where the agent sees $(r_t, \mathbf{a}_t)$ before committing resources.

**Online linear programming (OLP) and threshold-based decision rule.** OLP (Agrawal et al., 2014) is a framework of online resource allocation problems that uses *observe-then-decide*. Dual-based algorithms (Buchbinder & Naor, 2009; Devanur & Hayes, 2009; Agrawal & Devanur, 2015; Balseiro et al., 2020; Li & Ye, 2022; Bray, 2025) are the most common way to solve OLP problems. These methods learn dual prices and apply threshold rule to admission control. However, OLP assumes a *fixed* reward-resource distribution of the requests and does not explore over configurations. In this paper, our proposed SP-UCB-OLP algorithm learns both a probability distribution over the configurations and a dual price. The distribution guides the configuration selection decision, and the dual price gives a threshold-based admission rule.

Our threshold-based admission rule also relates to prophet inequalities (Krengel & Sucheston, 1977; Samuel-Cahn, 1984), which compare online policies to omni-

scient prophets. Modern extensions handle matroid constraints (Kleinberg & Weinberg, 2012) and pricing mechanisms (Dütting et al., 2017). Unlike prophet settings with adversarial inputs, reward and resource consumption of the requests in our model are stochastic but configuration-dependent. Dynamic pricing problems in revenue management (Gallego & van Ryzin, 1994; Talluri & van Ryzin, 2004; Besbes & Zeevi, 2012) are also closely related to the threshold-based admission rule.

**Saddle-point of max-min problem.** The offline optimization problem solved by our switching-aware oracle can be expressed as a max-min optimization problem, and our SP-UCB-OLP algorithm solves the joint online configuration selection and admission control problem by learning the saddle point of this max-min optimization problem. Thus our work also aligns with saddle-point learning and multiplicative weights (Freund & Schapire, 1997; Arora et al., 2012).

**Our positioning.** We unify configuration exploration (from BwK) with observe-then-decide threshold admission (from OLP) through a switching-aware oracle. This switching-aware oracle provides a valid upper bound for the total reward earned by any feasible online policies, and learning the saddle point of the max-min optimization problem corresponding to this oracle leads us to the SP-UCB-OLP algorithm that achieves sublinear regret.

## 3. Model

**Two-layer decision model.** The planning horizon has $T \in \mathbb{N}$ time periods, and one request arrives at each of the time periods. Let $d$ be the number of resource types, and $\mathbf{B} \in \mathbb{R}_+^d$ be the total budget vector. Denote $(r_t, \mathbf{a}_t) \in \mathbb{R}_+ \times \mathbb{R}_+^d$ as the reward-resource pair of request arriving in time period $t$. There are $K$ configurations $\Theta = \{1, \ldots, K\}$. Each configuration $\theta$ induces a reward–resource distribution $\mathcal{D}_\theta$. At each time $t = 1, \cdots, T$, the decision maker (1) selects a configuration $\theta_t \in \Theta$, (2) observes $(r_t, \mathbf{a}_t) \sim \mathcal{D}_{\theta_t}$ of the request arriving in time period $t$, (3) chooses $x_t \in \{0, 1\}$ after observing $(r_t, \mathbf{a}_t)$. If $x_t = 1$, the decision maker collects reward $r_t$ and consumes resources $\mathbf{a}_t$. $\{x_t\}_{t=1}^T$ must satisfy the **pathwise budget constraint**:

$$\sum_{t=1}^T \mathbf{a}_t x_t \leq \mathbf{B} \quad \text{(componentwise, almost surely)}.$$

In addition, the decisions made at each time period is irrevocable. The goal of the decision maker is to maximize the expected total reward $\mathbb{E}[\sum_{t=1}^T r_t x_t]$.

### 3.1. Assumptions

**Assumption 1** (Stationarity and Independence). *If an algorithm select configuration $\theta$ at time period $t$, $(r_t, \mathbf{a}_t)$ is drawn independently from $\mathcal{D}_\theta$. In addition, the distribution $\mathcal{D}_\theta$ is time-invariant for all $\theta \in \Theta$.*

**Assumption 2** (Boundedness). *There exist constants $R_{\max}, A_{\max} > 0$ such that for all $\theta$ and $(r, \mathbf{a}) \sim \mathcal{D}_\theta$:*

$$0 \leq r \leq R_{\max}, \quad \|\mathbf{a}\|_\infty \leq A_{\max} \quad a.s.$$

**Assumption 3** (Budget Scaling). *There exists $\mathbf{b} \in \mathbb{R}_+^d$ such that $B = T \cdot \mathbf{b}$.*

In the analysis of this paper, we fix the vector $\mathbf{b}$ and discuss the dependence of the performance of algorithms as the number of periods $T$ goes to infinity. Under this asymptotic regime, Assumption 3 indicates that we consider the "large resource budget" scenario in which the total budget is proportional to the total number of time periods.

**Assumption 4** (Continuous Valuation). *For any $\theta \in \Theta$, if $(r, \mathbf{a}) \sim \mathcal{D}_\theta$, then the conditional distribution of $r$ given $\mathbf{a} = \alpha$ is continuous for any $\alpha$ in the support of $\mathbf{a}$.*

We will later show that we do admission control through a price-based threshold method, in which we have a price $\mathbf{p} \in \mathbb{R}^d$ and accept a request with reward-resource pair $(r, \mathbf{a})$ if $r > \mathbf{a}^T \mathbf{p}$. Assumption 4 serves as a "No-Tie" assumption, i.e., for any $\mathbf{p} \in \mathbb{R}^d$, $\mathbb{P}(r = \mathbf{a}^T \mathbf{p}) = 0$.

## 4. Switching-Aware Fluid Oracle

We define an oracle and the corresponding offline optimization problem, which provides a benchmark that we can define the regret of an online algorithm with respect to. Recall $\mathbf{b} := \mathbf{B}/T$ is the per-period budget given in Assumption 3.

### 4.1. Fixed Configuration Oracle Leads to Negative Regret

Let $(r_{t,\theta}, \mathbf{a}_{t,\theta})$ be the reward-resource pair observed if the decision maker selects configuration $\theta$. Define

$$V_\theta^{off}(\mathbf{b}) := \mathbb{E}\Big[\max\Big\{\sum_{t=1}^T r_{t,\theta} x_{t,\theta} \Big| \sum_{t=1}^T \mathbf{a}_{t,\theta} x_{t,\theta} \leq T \cdot \mathbf{b},$$
$$0 \leq x_{t,\theta} \leq 1 \ \forall t\Big\}\Big].$$

The optimization problem inside the expectation defining $V_\theta^{off}(\mathbf{b})$ is the so-called offline LP in (Li & Ye, 2022). $V_\theta^{off}(\mathbf{b})$ represents the maximal expected total reward earned if the decision maker always selects configuration $\theta$ and knows the reward-resource pairs of all the incoming requests in advance. Let $\{x_{t,\theta}^*\}_{t=1}^T$ be the optimal solution of the maximization problem inside the expectation.

Then, a natural oracle of our problem is to always select $\theta^* = \arg\max_{\theta\in\Theta} V_\theta^{off}(\mathbf{b})$ and accepting requests according to $\{x_{t,\theta}^*\}_{t=1}^T$. Then, the expected total reward earned by this fixed configuration oracle is $V^*(\mathbf{b}) := V_{\theta^*}^{off}(\mathbf{b})$. However, the following example indicates that it is possible for an online policy having higher expected total reward than $V^*(\mathbf{b})$.

**Example 1** (Complementary Resources)**.** *Consider $T = 100$ periods, $d = 2$ resources with $\mathbf{b} = [0.5, 0.5]$, and two configurations with deterministic resource consumption: $(r \sim Uniform(0,2), \mathbf{a} = [1,0])$ and $(r = Uniform(0,2), \mathbf{a} = [0,1])$, each consuming only one type of resource. The fixed configuration oracle achieves $V^*(\mathbf{b}) = \frac{7550}{101} \approx 74.76$ (one budget wasted), but the optimal online policy is to select configuration 1 when $t \le 50$, select configuration 2 when $t > 50$ and accept all the requests, which earns $100$ as the expected total reward.*

This example motivates the following derivation of the switching-aware oracle.

## 4.2. Primal Mixed Fluid Relaxation

Let $\Delta_K = \{\mathbf{w} \in \mathbb{R}_+^K : \sum_\theta w_\theta = 1\}$ be the probability simplex. For each $\theta \in \Theta$, let $x_\theta : \mathbb{R}_+ \times \mathbb{R}_+^d \to [0,1]$ be a measurable acceptance rule. We define the **mixed fluid value** as:

$$V^{\mathrm{mix}}(\mathbf{b}) := \max_{\mathbf{w}\in\Delta_K} \max_{\{x_\theta\}} \left\{ \sum_{\theta\in\Theta} w_\theta \mathbb{E}_\theta[r\, x_\theta(r,\mathbf{a})] \;\middle|\; \sum_{\theta\in\Theta} w_\theta \mathbb{E}_\theta[\mathbf{a}\, x_\theta(r,\mathbf{a})] \le \mathbf{b} \right\}.$$

$$(1)$$

The max-max form represents the two-layer framework in our problem. The outer maximization represents the configuration selection, and the inner maximization represents the admission control after a configuration is selected. $V^{\mathrm{mix}}(\mathbf{b})$ also shows how OLP and BwK are connected with our problem. When $K = 1$, our problem is reduced to an OLP problem, and $V^{\mathrm{mix}}(\mathbf{b})$ is also equivalent to the fluid relaxation of OLP given by (Chen et al., 2024) in this case. When the acceptance rule is predetermined and not part of the decision, i.e., $\{x_\theta\}_{\theta\in\Theta}$ is fixed, our problem is reduced to a BwK problem, and $V^{\mathrm{mix}}(\mathbf{b})$ is also equivalent to the LP relaxation of BwK given by (Badanidiyuru et al., 2018) in this case.

The switching-aware fluid oracle knows the optimal solution $(\mathbf{w}^*, \{x^*(\theta)\})$. At each time period $t$, the oracle randomly selects a configuration $\theta_t$ following the probability distribution given by the **mixture** $\mathbf{w}^*$. After observing $(r_t, \mathbf{a}_t)$, the oracle applies the acceptance rule $x_{\theta_t}$ and earns $r_t x_{\theta_t}(r_t, \mathbf{a}_t)$. The switching-aware fluid oracle ignores the budget constraint. Thus, the total expected reward earned

by the switching-aware fluid oracle is $T \cdot V^{\mathrm{mix}}(\mathbf{b})$. In the following subsections, we characterize $V^{\mathrm{mix}}(\mathbf{b})$ and show $T \cdot V^{\mathrm{mix}}(\mathbf{b})$ is a valid benchmark.

## 4.3. Dual Form and Envelope Structure

**Envelope and threshold consumption.** For $\mathbf{p} \in \mathbb{R}^d$, define the **surplus** and **threshold consumption**:

$$g_\theta(\mathbf{p}) := \mathbb{E}_\theta[(r - \langle\mathbf{p},\mathbf{a}\rangle)_+],$$
$$h_\theta(\mathbf{p}) := \mathbb{E}_\theta[\mathbf{a}\,\mathbf{1}\{r > \langle\mathbf{p},\mathbf{a}\rangle\}].$$

For a mixture $\mathbf{w} \in \Delta_K$, define

$$L(\mathbf{w},\mathbf{p}) := \langle\mathbf{p},\mathbf{b}\rangle + \sum_\theta w_\theta g_\theta(\mathbf{p}).$$

**Theorem 1** (Primal–Dual Form of the Mixed Fluid Oracle)**.** *Under Assumptions 2 and 3, let $b_{min}$ be the smallest component of $\mathbf{b}$ (in particular, $b_{\min} > 0$) and $P_{max}$ be $2R_{max}/b_{min}$. With price domain $\mathcal{P} = [0, P_{\max}]^d$,*

$$V^{\mathrm{mix}}(\mathbf{b}) = \max_{\mathbf{w}\in\Delta_K} \min_{\mathbf{p}\in\mathcal{P}} \left\{ \langle\mathbf{p},\mathbf{b}\rangle + \sum_\theta w_\theta g_\theta(\mathbf{p}) \right\}$$

$$= \min_{\mathbf{p}\in\mathcal{P}} \left\{ \langle\mathbf{p},\mathbf{b}\rangle + \max_{\theta\in\Theta} g_\theta(\mathbf{p}) \right\}.$$

$$(2)$$

Define the induced consumption $H(\mathbf{w},\mathbf{p}) := \sum_\theta w_\theta h_\theta(\mathbf{p})$. Let $A(\mathbf{p}) := \arg\max_\theta g_\theta(\mathbf{p})$ be the **active (envelope) set**, and $\mathrm{supp}(\mathbf{w})$ be the indices of positive components of $\mathbf{w}$. The following theorem establishes the relationship between the primal form and the max-min form of $V^{\mathrm{mix}}(\mathbf{b})$.

**Theorem 2** (Primal–dual optimality (saddle/KKT conditions))**.** *Assume Assumptions 1–4. A pair $(\mathbf{w}^\star, \mathbf{p}^\star) \in \Delta_K \times \mathcal{P}$ is a saddle point of $\max_{\mathbf{w}\in\Delta_K} \min_{\mathbf{p}\in\mathcal{P}} L(\mathbf{w},\mathbf{p})$ (and hence attains $V^{\mathrm{mix}}(\mathbf{b})$) if and only if:*

*(i) (**Envelope support**) $\mathrm{supp}(\mathbf{w}^\star) \subseteq A(\mathbf{p}^\star)$.*

*(ii) (**Feasibility**) $H(\mathbf{w}^\star, \mathbf{p}^\star) \le \mathbf{b}$ (componentwise).*

*(iii) (**Complementarity**) $\langle\mathbf{p}^\star, \mathbf{b} - H(\mathbf{w}^\star, \mathbf{p}^\star)\rangle = 0$.*

*Moreover, the mixture $\mathbf{w}^\star$ together with threshold admission $x_\theta(r,\mathbf{a}) = \mathbf{1}\{r > \langle\mathbf{p}^\star, \mathbf{a}\rangle\}$ for all $\theta \in \Theta$, is primal-optimal for (1) and $V^{\mathrm{mix}}(\mathbf{b}) = L(\mathbf{w}^\star, \mathbf{p}^\star)$.*

Theorem 2 indicates that, at equilibrium, $\mathbf{w}^\star$ randomizes only among envelope-optimal configurations at $\mathbf{p}^\star$, and the induced threshold consumption matches the budget on priced resources (complementarity). The following theorem further characterizes the saddle points of the max-min form of $V^{\mathrm{mix}}(\mathbf{b})$.

**Theorem 3** (Characterization of all saddle points)**.** *Assume Assumptions 1–4. Define the envelope objective*

$$f(\mathbf{p}) := \langle \mathbf{p}, \mathbf{b} \rangle + \max_{\theta \in \Theta} g_\theta(\mathbf{p}), \qquad \mathcal{P}^\star := \arg\min_{\mathbf{p} \in \mathcal{P}} f(\mathbf{p}).$$

*Then the set of saddle points of L is exactly*

$$\mathcal{S} = \Big\{ (\mathbf{w}, \mathbf{p}) \in \Delta_K \times \mathcal{P} : \ \mathbf{p} \in \mathcal{P}^\star, \ \mathrm{supp}(\mathbf{w}) \subseteq A(\mathbf{p}),$$
$$H(\mathbf{w}, \mathbf{p}) \leq \mathbf{b}, \ \langle \mathbf{p}, \mathbf{b} - H(\mathbf{w}, \mathbf{p}) \rangle = 0 \Big\}.$$

### 4.4. Oracle Justification

Denote $R_T^\pi$ as the total reward earned by an online policy $\pi$.

**Theorem 4** (Switching-Aware Oracle Upper Bound)**.** *Under Assumptions 1–2, for any online policy $\pi$ satisfying the pathwise budget constraint,*

$$\mathbb{E}[R_T^\pi] \leq T \cdot V^{\mathrm{mix}}(\mathbf{b}).$$

*Proof sketch.* Fix $\mathbf{p} \in \mathcal{P}$, $r_t x_t \leq \langle \mathbf{p}, \mathbf{a}_t \rangle x_t + (r_t - \langle \mathbf{p}, \mathbf{a}_t \rangle)_+$ almost surely. By $\sum_t \mathbf{a}_t x_t \leq \mathbf{B}$ yields $R_T^\pi \leq \langle \mathbf{p}, \mathbf{B} \rangle + \sum_{t=1}^T (r_t - \langle \mathbf{p}, \mathbf{a}_t \rangle)_+$ almost surely. Taking expectation and upper bounding the average surplus by $\max_\theta g_\theta(\mathbf{p})$ gives $\mathbb{E}[R_T^\pi] \leq T(\langle \mathbf{p}, \mathbf{b} \rangle + \max_\theta g_\theta(\mathbf{p}))$. Minimizing over $\mathbf{p}$ completes the proof. Full proof is in Appendix C. □

**Remark 1.** *In Example 1, $T \cdot V^{\mathrm{mix}}(\mathbf{b}) = 100$ which provides a valid upper bound.*

## 5. Algorithm: SP-UCB–OLP

The switching-aware oracle motivates the following algorithm. At each time period $t$, we use the observed data to learn the surplus functions $\{g_\theta\}_{\theta \in \Theta}$. Using the sample averages $\{\hat{g}_{\theta,t}\}_{\theta \in \Theta}$ with respect to the observed data is the most straightforward way to learn surplus functions; however, to balance exploration and exploitation, we apply the idea of UCB and add confidence radii $\{\beta_{\alpha,\theta}(t)\}$ to $\{\hat{g}_{\theta,t}\}_{\theta \in \Theta}$ to get our final estimates of the surplus functions. We replace the surplus functions in (2) with the estimated surplus function to get an optimistic saddle point problem and solve this problem to get $(\mathbf{w}_t, \mathbf{p}_t)$. We then randomly select a configuration $\theta_t$ following the probability distribution given by $\mathbf{w}_t$ and accept the request if $r_t > \mathbf{a}_t^T \mathbf{p}_t$ and we have enough resources. We introduce the formulation of confidence radii $\beta_{\alpha,\theta}(t)$ and the details of algorithm in the following sections.

### 5.1. Confidence Radii and Optimistic Saddle Point Problem

We store every observed $(r_t, \mathbf{a}_t)$ regardless of whether it is admitted, so that surplus estimates are unbiased. For each configuration $\theta$, maintain a dataset $\mathcal{S}_\theta$ of all observed samples (accepted or rejected) with count $N_\theta = |\mathcal{S}_\theta|$. Define the empirical surplus:

$$\hat{g}_{\theta,t}(\mathbf{p}) := \frac{1}{N_\theta \vee 1} \sum_{(r,\mathbf{a}) \in \mathcal{S}_\theta} (r - \langle \mathbf{p}, \mathbf{a} \rangle)_+.$$

By Lemma 15 and its anytime version (Corollary 16) in the Appendix, there exist a constant $\underline{\alpha} > 0$ such that if $\alpha \geq \underline{\alpha}$, then for any $\delta \in (0, 1)$, with probability at least $1 - \delta$, the following holds simultaneously for all $\theta \in \Theta$ and all rounds $t \leq T$:

$$\sup_{\mathbf{p} \in \mathcal{P}} \big| g_\theta(\mathbf{p}) - \hat{g}_{\theta,t}(\mathbf{p}) \big| \ \leq \ \beta_{\alpha,\theta}(t),$$

where the confidence radius is

$$\beta_{\alpha,\theta}(t) := \alpha \sqrt{\frac{d \log t + \log(KT/\delta)}{N_\theta \vee 1}}.$$

Equivalently, estimating each surplus function uniformly to accuracy $\epsilon$ requires $N_\theta = \tilde{O}(d/\epsilon^2)$ samples per configuration. Then, at each round, we solve the following **optimistic saddle point problem**:

$$(\mathbf{w}_t, \mathbf{p}_t) \in \arg\max_{\mathbf{w} \in \Delta_K} \arg\min_{\mathbf{p} \in \mathcal{P}} L_t^{\mathrm{opt}}(\mathbf{w}, \mathbf{p})$$
$$= \arg\max_{\mathbf{w} \in \Delta_K} \arg\min_{\mathbf{p} \geq 0} L_t^{\mathrm{opt}}(\mathbf{w}, \mathbf{p}), \qquad (3)$$

where $L_t^{\mathrm{opt}}(\mathbf{w}, \mathbf{p}) := \langle \mathbf{p}, \mathbf{b}^{\mathrm{safe}} \rangle + \sum_\theta w_\theta(\hat{g}_{\theta,t}(\mathbf{p}) + \beta_{\alpha,\theta}(t))$, and $\mathbf{b}^{\mathrm{safe}} := (1 - \varepsilon)\mathbf{b}$ with $\varepsilon = \sqrt{\log T/T}$. The conservative per-period budget $\mathbf{b}^{\mathrm{safe}}$ is used to do feasibility control, which is consistent with literature practice such as in (Agrawal et al., 2014). The equality that replace $\mathbf{p} \in \mathcal{P}$ with $\mathbf{p} \geq 0$ is by Assumption 2. For the proof of this equality, please see Appendix F. After the warm start, $N_\theta \geq 1$ for all $\theta$, and the saddle point in (3) is computed by the finite LP

$$
\begin{array}{ll}
\min_{\mathbf{p}, z, \{y_{j,\theta}\}} & \mathbf{b}^{\mathrm{safe}\top} \mathbf{p} + z \\
\text{s.t.} & z \geq \frac{1}{N_\theta} \sum_{j=1}^{N_\theta} y_{j,\theta} + \beta_{\alpha,\theta}(t), \quad \forall \theta, \\
& y_{j,\theta} \geq r_{j,\theta} - \mathbf{a}_{j,\theta}^\top \mathbf{p}, \quad \forall \theta, j, \\
& \mathbf{p} \geq 0, \ y_{j,\theta} \geq 0, \quad \forall \theta, j.
\end{array}
$$

The primal optimizer gives $\mathbf{p}_t$, while the optimal dual multipliers of the first group of constraints give $\mathbf{w}_t$. This LP has $d + 1 + \sum_\theta N_\theta(t)$ primal variables and $K + \sum_\theta N_\theta(t)$ main constraints. With the doubling schedule in Appendix F, it is solved only $O(K \log T)$ times over the horizon.

### 5.2. Algorithm Details

Algorithm 1 provides the full details of our **SP-UCB-OLP** algorithm.

---

**Algorithm 1** SP-UCB–OLP:

---

1: **Input:** Configurations $\Theta$ ($|\Theta| = K$), budget $\mathbf{B}$, horizon $T$, exploration parameter $\alpha$, slack $\varepsilon = \sqrt{\log T/T}$
2: **Initialize:** $\mathbf{B}^{\mathrm{rem}} \leftarrow \mathbf{B}$, $\mathbf{b} \leftarrow \mathbf{B}/T$, $\mathbf{b}^{\mathrm{safe}} \leftarrow (1-\varepsilon)\mathbf{b}$
3: For each $\theta$: $N_\theta \leftarrow 0$, $\mathcal{S}_\theta \leftarrow \emptyset$
4: **for** $t = 1$ to $T$ **do**
5:     **// Pure observation phase: initialize sample sets without admission**
6:     **if** $t \leq K$ **then**
7:         $\theta_t \leftarrow t$ {Round-robin initialization}
8:         Observe $(r_t, \mathbf{a}_t) \sim \mathcal{D}_{\theta_t}$
9:         $\mathcal{S}_{\theta_t} \leftarrow \mathcal{S}_{\theta_t} \cup \{(r_t, \mathbf{a}_t)\}$;    $N_{\theta_t} \leftarrow N_{\theta_t} + 1$
10:        $x_t \leftarrow 0$ {Observe only, no admission}
11:        **continue**
12:     **end if**
13:     **// Compute confidence radii**
14:     **for** each $\theta \in \Theta$ **do**
15:         $\beta_{\alpha,\theta}(t) \leftarrow \alpha\sqrt{\frac{d\log t + \log(KT/\delta)}{N_\theta \vee 1}}$
16:     **end for**
17:     **// Empirical surpluses**
18:     $\widehat{g}_{\theta,t}(\mathbf{p}) \leftarrow \frac{1}{N_\theta \vee 1}\sum_{(r,\mathbf{a})\in\mathcal{S}_\theta}(r - \langle \mathbf{p}, \mathbf{a}\rangle)_+$
19:     **// Solve optimistic saddle point problem**
20:     $(\mathbf{w}_t, \mathbf{p}_t) \in \arg\max_{\mathbf{w}\in\Delta_K} \arg\min_{\mathbf{p}\geq 0} L_t^{\mathrm{opt}}(\mathbf{w}, \mathbf{p})$ {$L_t^{\mathrm{opt}}$ from (3)}
21:     **// Sample a configuration from the mixture**
22:     Draw $\theta_t \sim \mathbf{w}_t$
23:     Observe $(r_t, \mathbf{a}_t) \sim \mathcal{D}_{\theta_t}$
24:     $\mathcal{S}_{\theta_t} \leftarrow \mathcal{S}_{\theta_t} \cup \{(r_t, \mathbf{a}_t)\}$;    $N_{\theta_t} \leftarrow N_{\theta_t} + 1$
25:     **// Admission decision (bid-price with hard feasibility)**
26:     **if** $\mathbf{a}_t \leq \mathbf{B}^{\mathrm{rem}}$ and $r_t > \langle \mathbf{p}_t, \mathbf{a}_t\rangle$ **then**
27:         Accept: $x_t \leftarrow 1$;    $\mathbf{B}^{\mathrm{rem}} \leftarrow \mathbf{B}^{\mathrm{rem}} - \mathbf{a}_t$
28:     **else**
29:         Reject: $x_t \leftarrow 0$
30:     **end if**
31: **end for**

---

### 5.3. Theoretical Results

Define the **switching-aware regret** of an online policy $\pi$ as:

$$\mathrm{Reg}_\pi^{\mathrm{mix}}(T) := T \cdot V^{\mathrm{mix}}(\mathbf{b}) - \mathbb{E}[R_T^\pi].$$

**Theorem 5** (Main Theorem: Regret vs. Switching-Aware Fluid Oracle). *Denote Algorithm 1 as $\pi_1$, under Assumptions 1–4, run Algorithm 1 with $\varepsilon = \sqrt{\log T/T}$, confidence level $\delta = T^{-2}$, and exploration parameter $\alpha \geq \underline{\alpha}$. Then*

$$\mathrm{Reg}_{\pi_1}^{\mathrm{mix}}(T) \leq C_1\,\alpha\,\sqrt{KT \cdot d\log T} + C_2\,\sqrt{T\log T} + KR_{\max},$$

*where constants $\underline{\alpha}, C_1, C_2 > 0$ depend only on $(d, R_{\max}, A_{\max}, \mathbf{b})$. In particular, for fixed $d$,*

$$\mathrm{Reg}_{\pi_1}^{\mathrm{mix}}(T) = \tilde{O}(\alpha\sqrt{KT}).$$

**Remark 2.** *Algorithm 1 may perform well empirically under small value of $\alpha$, but we cannot guarantee the regret upper bound in Theorem 5 holds because the high-probability argument in our regret analysis may not hold. See Appendix D for the details.*

## 6. Experiments

Our experiments pursue three goals: (i) validate that SP-UCB–OLP achieves $\tilde{O}(\sqrt{T})$ regret as predicted by Theorem 5; (ii) validate that the switching-aware benchmark $V^{\mathrm{mix}}$ is necessary by demonstrating the complementarity gap; and (iii) test the algorithm on real GPU cluster traces. In all experiments, the optimistic saddle point problem is solved via the LP formulation in Appendix F. In addition to **SP-UCB–OLP**, we also test the other three algorithms: (1) **Greedy** runs **SP–UCB–OLP** with $\alpha = 0$ yielding pure exploitation. (2) **Random** selects $\theta_t$ from $\Theta$ uniformly at random and accepts as long as the budget constraint is not violated. (3) **Oracle** runs **SP-UCB–OLP** by replacing $\mathbf{w}_t$ and $\mathbf{p}_t$ with the true optimal mixture $\mathbf{w}^*$ and price $\mathbf{p}^*$ computed by the switching-aware oracle at each round.

In addition to regret $\mathrm{Reg}_\pi^{\mathrm{mix}}(T)$, we also consider the competitive ratio $\mathrm{CR}_\pi = \mathbb{E}[R_T^\pi]/(T \cdot V^{\mathrm{mix}}(\mathbf{b}))$ as another performance measurement. Unless otherwise noted, experiments use $d = 3$ resource dimensions. Budget is parameterized by a scaling factor $\rho > 0$: $\mathbf{b} = \rho \cdot \mathbf{b}_0$, where $\mathbf{b}_0$ is a scenario-specific baseline per-period budget (see Appendix G); $\rho < 1$ corresponds to tighter budgets and $\rho > 1$ to looser ones.

### 6.1. Regret Scaling with Horizon

We run SP-UCB–OLP with the theory-compliant exploration parameter $\alpha = 1.5$ on a $K = 5$ Gaussian scenario (S0) with $d = 3$ resources and budget scaling $\rho = 0.7$. Each configuration $\theta \in \{0, \ldots, 4\}$ generates rewards $r \sim \mathcal{N}(\mu_r^\theta, (\sigma_r^\theta)^2)$ and per-resource consumptions $a_j \sim \mathcal{N}(\mu_{a,j}^\theta, (\sigma_{a,j}^\theta)^2)$, independently truncated to $[0.01, R_{\max}]$ and $[0.01, A_{\max}]$ respectively to satisfy Assumption 2, with $R_{\max} = A_{\max} = 2.0$. The five configurations span a range of reward–consumption trade-offs; full parameter tables appear in Appendix G. The per-period budget is $\mathbf{b} = \rho \cdot \mathbf{b}_0$ where $\mathbf{b}_0$ is the mean consumption vector of the most resource-intensive configuration, and the baseline budget $\mathbf{b}_0$ is specified in Appendix G. Algorithm parameters are $P_{\max} = 2.0$, $\delta = T^{-2}$, $\varepsilon = \sqrt{\log T/T}$, $c_g = 0.0707$ (empirically tuned to match the theoretical $\sqrt{T}$ scaling), with warm-start of $K$ rounds (round-robin, no budget consumption) and saddle-point re-solves on a doubling schedule.

We vary $T \in \{100, 200, 500, 1000, 2000\}$ with 50 independent seeds. Table 1 and Figure 2 show that Regret/$\sqrt{T}$ remains approximately constant ($\approx 1.6$) across all horizon

*Table 1.* Regret scaling with $\alpha = 1.5$ (S0, 50 seeds).

| $T$ | Regret | Regret/$\sqrt{T}$ | CR |
|---|---|---|---|
| 100 | $15.9 \pm 4.1$ | $1.59 \pm 0.41$ | 0.63 |
| 200 | $22.5 \pm 6.0$ | $1.59 \pm 0.42$ | 0.74 |
| 500 | $35.2 \pm 10.2$ | $1.58 \pm 0.46$ | 0.84 |
| 1000 | $50.6 \pm 13.4$ | $1.60 \pm 0.42$ | 0.88 |
| 2000 | $71.8 \pm 24.7$ | $1.61 \pm 0.55$ | 0.92 |

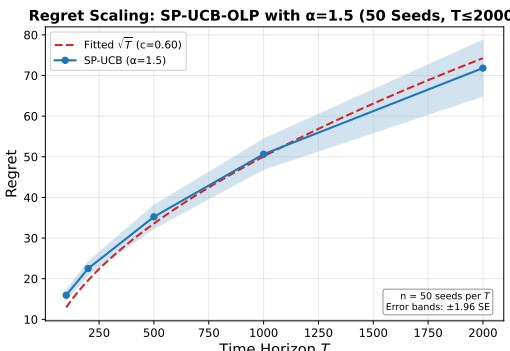

*Figure 2.* Regret scaling with $\alpha = 1.5$ (S0, 50 seeds).

*Table 2.* Alibaba regime configurations with reward coefficients.

| Regime | Description | $c_1$ (CPU) | $c_2$ (Mem) |
|---|---|---|---|
| 0 | CPU-heavy | 2.0 | 0.5 |
| 1 | Memory-heavy | 0.5 | 2.0 |
| 2 | Balanced | 1.2 | 1.2 |

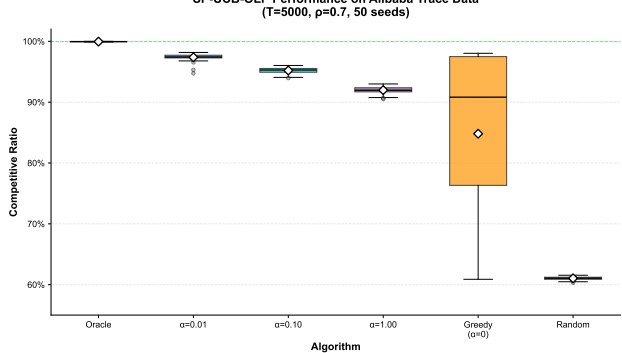

*Figure 3.* Alibaba traces: competitive ratio across 50 seeds ($T = 5,000$, $\rho = 1.0$). Greedy ($\alpha = 0$) exhibits high variance due to regime lock-in; minimal exploration ($\alpha = 0.01$) dramatically stabilizes performance.

lengths, confirming the $\tilde{O}(\sqrt{T})$ scaling predicted by Theorem 5.

### 6.2. Real-World Validation: Alibaba Traces

To validate beyond synthetic settings, we test on real cluster traces from Alibaba (Alibaba Group, 2018) with $d = 2$ resource dimensions (CPU and memory), $K = 3$ regime configurations, and $T = 5,000$ arrivals processed in original temporal order.

**Data and reward modeling.** We use real resource consumption from the trace: $\text{cpu} = \text{plan\_cpu}/100$ and $\text{mem} = \text{plan\_mem}/100$, both normalized to $[0, 1]$. The Alibaba trace exhibits significant resource imbalance: mean CPU utilization is $83.6\%$ while mean memory is $34.9\%$. Since the raw trace contains no native reward field, we construct reward as:

$$r = c_1[\theta] \cdot \text{cpu} + c_2[\theta] \cdot \text{mem} + \epsilon, \quad \epsilon \sim \mathcal{N}(0, 0.1^2)$$

where $c_1[\theta], c_2[\theta]$ are regime-specific coefficients. This creates a *stationary LP structure*: the optimal regime depends on which resource is more abundant, creating a meaningful learning problem.

**Configuration table.** Table 2 summarizes the three regime configurations. The CPU-heavy regime rewards CPU-intensive tasks; the Memory-heavy regime rewards memory-intensive tasks; the Balanced regime provides equal reward per unit of either resource. Full details appear in Appendix G.5.

**The value of minimal exploration.** Figure 3 shows competitive ratio distributions across 50 seeds with $\rho = 1.0$. A striking finding is the dramatic benefit of minimal exploration: Greedy ($\alpha = 0$) achieves $84.8\% \pm 14.2\%$ CR with 10 out of 50 seeds stuck below 70%, while $\alpha = 0.01$ achieves $97.4\% \pm 0.6\%$ CR. This $24\times$ reduction in variance demonstrates that even minimal exploration prevents lock-in to suboptimal regimes. SP-UCB-OLP with $\alpha = 0.01$ approaches Oracle performance ($99.95\%$) while Random provides a lower bound at $61.1\%$.

### 6.3. Benchmark Validation: Complementarity Gap

To test whether the switching-aware benchmark $V^{\text{mix}}$ is necessary, we construct a diagnostic scenario S4 with $K = 2$ configurations having orthogonal resource usage ($d = 2$). Configuration 0 has reward $r \approx 1$ (with uniform noise $\pm 0.01$) and consumption $\mathbf{a} \approx [1, 0]$, consuming only resource 1; configuration 1 has $r \approx 1$ and $\mathbf{a} \approx [0, 1]$, consuming only resource 2. The per-period budget is $\mathbf{b} = [0.5, 0.5]$ with $\rho = 0.7$. By design, the fixed-configuration oracle achieves $V^*(\mathbf{b}) \approx 0.5$ (one resource wasted), whereas the switching-aware oracle achieves $V^{\text{mix}}(\mathbf{b}) \approx 1.0$ (both resources utilized), yielding a complementarity gap of $\approx 2$. We use $T = 5,000$ and 10 random seeds; full specifications appear in Appendix G.

Recall $V^*(\mathbf{b})$ provided by the fixed configuration oracle defined in Section 4.1 and define $CR^* = \mathbb{E}[R_T^\pi]/V^*(\mathbf{b})$. Table 3 shows that all learning algorithms achieve $\text{CR}_\pi^* > 1$ (exceeding the fixed oracle), while $\text{CR}_\pi^{\text{mix}} \leq 1$ for all algo-

*Table 3.* Benchmark validation on S4 ($K = 2$, $d = 2$, $\rho = 0.7$, $T = 5,000$, 10 seeds). $\mathrm{CR}^* > 1$ confirms the fixed oracle is beatable; $\mathrm{CR}^{\mathrm{mix}} \leq 1$ confirms $V^{\mathrm{mix}}$ is a valid upper bound.

| Algorithm | $\mathrm{CR}^{\mathrm{mix}}$ | $\mathrm{CR}^*$ | Gap |
|---|---|---|---|
| SP-UCB-OLP | $0.91 \pm 0.02$ | $1.81 \pm 0.04$ | 1.99 |
| Greedy | $0.96 \pm 0.01$ | $1.90 \pm 0.02$ | 1.99 |
| Random | $0.99 \pm 0.00$ | $1.97 \pm 0.00$ | 1.99 |
| OneHot | $0.51 \pm 0.01$ | $1.01 \pm 0.02$ | 1.99 |

rithms, confirming $T \cdot V^{\mathrm{mix}}$ is a valid upper bound. Notably, OneHot achieves $\mathrm{CR}^* \approx 1.01$—it cannot exploit complementarity because it commits to a single configuration.

Complete scenario specifications, algorithm parameters, and extended results appear in Appendix G.

## 7. Conclusion

We studied a two-layer sequential decision problem combining configuration exploration and admission control under budget constraints. We construct a **switching-aware fluid oracle** which provides a benchmark that upper bounds the expected total reward of any online policy. This **switching-aware fluid oracle** also inspires the design of **SP-UCB–OLP** algorithm, which solves an optimistic saddle point problem at each round and uses the computed saddle point to select configurations and do admission control. We show that the regret of **SP-UCB–OLP** algorithm against the benchmark given by the switching-aware oracle is $\tilde{O}(\sqrt{KT})$. We also run numerical experiments to study the empirical performance of **SP-UCB–OLP** algorithm.

**Limitations and future work.** Our analysis assumes i.i.d. and bounded reward-resource pair of requests within each configuration. Extensions to non-stationary arrivals, contextual settings (where configurations depend on observed context), and delayed feedback are natural future research directions.

## Acknowledgements

PJ acknowledges funding from ONR grant N00014-24-1-2470 and AFOSR grant FA9550-23-1-0190.

## Impact Statement

Our framework for budgeted admission control has both beneficial applications and potential risks.

**Positive impacts.**

- **Energy efficiency**: Optimizing resource allocation under carbon budgets can reduce infrastructure emissions.

- **Cost reduction**: Improved utilization reduces costs, democratizing compute access.

- **Transparency**: Dual prices provide interpretable rejection signals, unlike black-box heuristics.

**Risks and limitations.**

- **Fairness**: Reward-maximizing admission may disadvantage low-priority users; fairness constraints should supplement efficiency objectives.

- **Strategic behavior**: Observable rejection patterns may incentivize priority inflation.

- **Information leakage**: Publishing dual prices reveals system scarcity.

**Recommendations for deployment.**

1. Incorporate fairness metrics alongside efficiency.

2. Provide clear explanations for rejections.

3. Monitor for strategic behavior.

4. Combine with absolute resource limits.

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

## Appendix Roadmap

**Roadmap.** This appendix provides complete proofs and extended experiments. Section A defines notation. Section B details the algorithm. Section C proves the oracle characterization (Theorems 1–4 and the saddle/KKT conditions). Section D establishes concentration bounds. Section E proves the regret bound (Theorem 5). Section F discusses implementation. Section G presents full experimental specifications and extended results.

## A. Notation

**Problem Parameters:**

- $T$: Time horizon

- $\mathbf{B} \in \mathbb{R}^d_+$: Initial resource budget vector

- $\mathbf{b} := \mathbf{B}/T$: Per-period budget

- $\mathbf{b}^{\text{safe}} := (1 - \varepsilon)\mathbf{b}$: Conservative (safe) per-period budget

- $\Theta = \{1, \ldots, K\}$: Set of configurations

- $d$: Resource dimensionality

- $\mathcal{P} = [0, P_{\max}]^d$: Price domain

**Distribution and Arrival Parameters:**

- $\mathcal{D}_\theta$: Distribution over $(r, \mathbf{a})$ induced by configuration $\theta$

- $(r_t, \mathbf{a}_t) \sim \mathcal{D}_{\theta_t}$: Arrival at time $t$

- $x_t \in \{0, 1\}$: Admission decision

- $R_{\max}$: Upper bound on rewards

- $A_{\max}$: Upper bound on resource consumption

**Key Functions:**

- $g_\theta(\mathbf{p}) := \mathbb{E}_\theta[(r - \langle \mathbf{p}, \mathbf{a} \rangle)_+]$: Surplus function

- $h_\theta(\mathbf{p}) := \mathbb{E}_\theta[\mathbf{a} \cdot \mathbf{1}\{r > \langle \mathbf{p}, \mathbf{a} \rangle\}]$: Strict-threshold consumption function (analysis uses $>$)

- $\widehat{g}_{\theta,t}(\mathbf{p})$: Empirical surplus from $N_\theta(t)$ samples

- $\widehat{h}_{\theta,t}(\mathbf{p}) := \frac{1}{N_\theta(t)} \sum_{(r,\mathbf{a}) \in \mathcal{S}_\theta(t)} \mathbf{a} \cdot \mathbf{1}\{r > \langle \mathbf{p}, \mathbf{a} \rangle\}$: Strict-threshold empirical consumption

**Remark 3** (Tie-Handling Convention). *Throughout the analysis, we use strict-threshold indicators ($r > \langle \mathbf{p}, \mathbf{a} \rangle$) for mathematical convenience. Under Assumption 5, this is equivalent to weak thresholds ($r \geq \langle \mathbf{p}, \mathbf{a} \rangle$) for online arrivals. However, empirical minimizers of piecewise-linear objectives can lie on breakpoints, so the analysis uses strict thresholds to avoid tie-ambiguity. See Lemma 11 for the tie-weighted subdifferential treatment.*

**Algorithm Variables:**

- $\mathbf{w}_t \in \Delta_K$: Mixture over configurations at time $t$ (algorithm output)

- $\theta_t \sim \mathbf{w}_t$: Configuration sampled from mixture at time $t$

- $\mathbf{p}_t \in \mathcal{P}$: Global bid price at time $t$

- $\beta_\theta(t)$: Confidence radius for configuration $\theta$ at time $t$

**Oracle and Regret:**

- $V^{\mathrm{mix}}(\mathbf{b}) = \min_{\mathbf{p} \in \mathcal{P}} \{\langle \mathbf{p}, \mathbf{b} \rangle + \max_\theta g_\theta(\mathbf{p})\}$: Switching-aware fluid oracle

- $\mathrm{Reg}^{\mathrm{mix}}(T) := TV^{\mathrm{mix}}(\mathbf{b}) - \mathbb{E}[R_T]$: Switching-aware regret

- $\widehat{V}_t^{\mathrm{opt}}(\mathbf{w}) := \min_{\mathbf{p} \in \mathcal{P}} \{\langle \mathbf{p}, \mathbf{b}^{\mathrm{safe}} \rangle + \sum_\theta w_\theta(\widehat{g}_{\theta,t}(\mathbf{p}) + \beta_\theta(t))\}$: Optimistic mixed value

# B. Algorithm Details

We restate Algorithm 1 (SP-UCB-OLP) from the main paper with additional implementation details.

---

**Algorithm 2** SP-UCB-OLP: Optimistic Saddle + Mixture Sampling + Bid-Price Admission

---

1: **Input:** Configurations $\Theta$ ($|\Theta| = K$), budget $\mathbf{B}$, horizon $T$, price box $\mathcal{P} = [0, P_{\max}]^d$, exploration parameter $\alpha$, slack $\varepsilon = \sqrt{\log T / T}$

2: **Initialize:** $\mathbf{B}^{\mathrm{rem}} \leftarrow \mathbf{B}$, $\mathbf{b} \leftarrow \mathbf{B}/T$, $\mathbf{b}^{\mathrm{safe}} \leftarrow (1 - \varepsilon)\mathbf{b}$

3: For each $\theta$: $N_\theta \leftarrow 0$, $\mathcal{S}_\theta \leftarrow \emptyset$

4: **for** $t = 1$ to $T$ **do**

5:     // **Warm start: round-robin exploration without consuming budget**

6:     **if** $t \leq K$ **then**

7:         $\theta_t \leftarrow t$ {Round-robin}

8:         Observe $(r_t, \mathbf{a}_t) \sim \mathcal{D}_{\theta_t}$

9:         $\mathcal{S}_{\theta_t} \leftarrow \mathcal{S}_{\theta_t} \cup \{(r_t, \mathbf{a}_t)\}$;    $N_{\theta_t} \leftarrow N_{\theta_t} + 1$

10:       $x_t \leftarrow 0$ {Do not consume budget during warm start}

11:       **continue**

12:     **end if**

13:     // **Compute confidence radii**

14:     **for** each $\theta \in \Theta$ **do**

15:         $\beta_\theta(t) \leftarrow \alpha \cdot c_g R_{\max} \sqrt{\frac{d \log t + \log(KT/\delta)}{N_\theta \vee 1}}$

16:     **end for**

17:     // **Empirical surpluses**

18:     $\widehat{g}_{\theta,t}(\mathbf{p}) \leftarrow \frac{1}{N_\theta \vee 1} \sum_{(r,\mathbf{a}) \in \mathcal{S}_\theta} (r - \langle \mathbf{p}, \mathbf{a} \rangle)_+$

19:     // **Solve optimistic saddle problem (exact or approximate)**

20:     $(\mathbf{w}_t, \mathbf{p}_t) \in \arg\max_{\mathbf{w} \in \Delta_K} \arg\min_{\mathbf{p} \in \mathcal{P}} \{\langle \mathbf{p}, \mathbf{b}^{\mathrm{safe}} \rangle + \sum_\theta w_\theta(\widehat{g}_{\theta,t}(\mathbf{p}) + \beta_\theta(t))\}$

21:     // **Sample a configuration from the mixture**

22:     Draw $\theta_t \sim \mathbf{w}_t$

23:     Observe $(r_t, \mathbf{a}_t) \sim \mathcal{D}_{\theta_t}$

24:     $\mathcal{S}_{\theta_t} \leftarrow \mathcal{S}_{\theta_t} \cup \{(r_t, \mathbf{a}_t)\}$;    $N_{\theta_t} \leftarrow N_{\theta_t} + 1$

25:     // **Admission decision (bid-price with hard feasibility)**

26:     **if** $\mathbf{a}_t \leq \mathbf{B}^{\mathrm{rem}}$ and $r_t \geq \langle \mathbf{p}_t, \mathbf{a}_t \rangle$ **then**

27:         Accept: $x_t \leftarrow 1$;    $\mathbf{B}^{\mathrm{rem}} \leftarrow \mathbf{B}^{\mathrm{rem}} - \mathbf{a}_t$

28:     **else**

29:         Reject: $x_t \leftarrow 0$

30:     **end if**

31: **end for**

---

**Key design features:**

1. **Warm start**: For $t \leq K$, sample each configuration once without consuming budget. This ensures $N_\theta(t) \geq 1$ for all $\theta$ when the saddle problem is first solved, avoiding "$N_\theta \vee 1$" hacks. The warm start contributes an additive $KR_{\max}$ to regret.

2. **Mixture $\mathbf{w}_t$ is part of the policy**: The algorithm outputs a mixture over configurations and *samples* $\theta_t \sim \mathbf{w}_t$. This is not equivalent to picking the single maximizer of the envelope—the mixture is essential for matching multi-resource budgets under complementary configurations.

3. **Global price $\mathbf{p}_t$**: The price is the dual variable of the switching-aware benchmark, not per-configuration.

4. **Optimistic saddle problem**: The term $\sum_\theta w_\theta(\widehat{g}_{\theta,t}(\mathbf{p}) + \beta_\theta(t))$ learns the mixed value with optimism.

5. **Sample storage**: All samples $(r, \mathbf{a})$ are stored regardless of admission decision, enabling unbiased estimation.

6. **Conservative slack**: Using $\mathbf{b}^{\mathrm{safe}} = (1 - \varepsilon)\mathbf{b}$ ensures budget feasibility with high probability.

7. **Strict vs. weak threshold**: The algorithm uses $r_t \geq \langle \mathbf{p}_t, \mathbf{a}_t \rangle$ in implementation; the analysis uses strict thresholds (see Remark 3).

**Important note on mixture vs. one-hot selection.** The benchmark $V^{\mathrm{mix}}$ is generally attained by a *mixture* over configurations, not a single configuration. If one instead computes $\mathbf{p}_t$ via envelope minimization and picks $\theta_t \in \arg\max_\theta(\widehat{g}_{\theta,t}(\mathbf{p}_t) + \beta_\theta(t))$, the policy may fail to match budget constraints when configurations have complementary resource profiles. Sampling from the mixture $\mathbf{w}_t$ ensures the algorithm's expected resource consumption aligns with the safe budget.

## C. Oracle Construction and Proofs

This section proves the primal–dual form of the switching-aware fluid oracle (Theorem 1), the saddle/KKT conditions (Theorem 2), the characterization of all saddle points (Theorem 3), and the oracle upper bound (Theorem 4).

### C.1. Bounded Dual Prices

We first justify that restricting dual prices to a compact box $\mathcal{P} = [0, P_{\max}]^d$ is without loss of generality.

**Lemma 6** (Bounded Dual Prices are W.L.O.G. (Safe Budget)). *Assume $0 \leq r \leq R_{\max}$ a.s. and $\mathbf{b}^{\mathrm{safe}} \in \mathbb{R}_+^d$ satisfies $b_i^{\mathrm{safe}} \geq b_{\min}^{\mathrm{safe}} > 0$ for all $i$. Then every minimizer of*

$$\min_{\mathbf{p} \geq 0} \left\{ \langle \mathbf{p}, \mathbf{b}^{\mathrm{safe}} \rangle + \max_{\theta \in \Theta} g_\theta(\mathbf{p}) \right\}$$

*lies in the box $\|\mathbf{p}\|_\infty \leq R_{\max}/b_{\min}^{\mathrm{safe}}$. Hence for any $P_{\max} \geq R_{\max}/b_{\min}^{\mathrm{safe}}$, restricting to $\mathcal{P} = [0, P_{\max}]^d$ does not change $V^{\mathrm{mix}}(\mathbf{b}^{\mathrm{safe}})$.*

*Proof.* Let $i$ be any coordinate and suppose $p_i > R_{\max}/b_i^{\mathrm{safe}}$. Then $\langle \mathbf{p}, \mathbf{b}^{\mathrm{safe}} \rangle \geq p_i b_i^{\mathrm{safe}} > R_{\max}$. Since $(r - \langle \mathbf{p}, \mathbf{a} \rangle)_+ \geq 0$, the objective at $\mathbf{p}$ exceeds $R_{\max}$.

At $\mathbf{p} = \mathbf{0}$, the objective equals $\max_\theta \mathbb{E}_\theta[(r)_+] = \max_\theta \mathbb{E}_\theta[r] \leq R_{\max}$. Therefore any point with $p_i > R_{\max}/b_i^{\mathrm{safe}}$ cannot be optimal. Applying this coordinatewise yields $\|\mathbf{p}^\star\|_\infty \leq R_{\max}/b_{\min}^{\mathrm{safe}}$. $\square$

**Importance.** This lemma ensures:

- The minimax theorem (Sion) applies with compact domain $\mathcal{P}$.

- The upper boundary constraint $p_i \leq P_{\max}$ is non-binding at optimality.

- KKT conditions for the inner minimization involve only the lower boundary $\mathbf{p} \geq 0$.

**Note:** Whenever the analysis claims "the upper box constraint $p_i \leq P_{\max}$ is non-binding," the correct condition is $P_{\max} > R_{\max}/b_{\min}^{\mathrm{safe}}$.

**Corollary 7** (Bounded Dual Prices for Empirical Objectives). *Fix any mixture $\mathbf{w} \in \Delta_K$ and any time $t$. Consider the empirical objective*

$$\min_{\mathbf{p} \geq 0} \left\{ \langle \mathbf{p}, \mathbf{b}^{\mathrm{safe}} \rangle + \sum_\theta w_\theta \widehat{g}_{\theta,t}(\mathbf{p}) \right\}, \qquad \widehat{g}_{\theta,t}(\mathbf{p}) = \frac{1}{N_\theta(t)} \sum_{(r,\mathbf{a}) \in \mathcal{S}_\theta(t)} (r - \langle \mathbf{p}, \mathbf{a} \rangle)_+.$$

*Then every minimizer satisfies $\|\mathbf{p}\|_\infty \leq R_{\max}/b_{\min}^{\mathrm{safe}}$. Hence if $P_{\max} > R_{\max}/b_{\min}^{\mathrm{safe}}$, the upper box constraint $p_i \leq P_{\max}$ is non-binding at the empirical minimizer(s) as well.*

*Proof.* The argument is identical to Lemma 6. If $p_i > R_{\max}/b_i^{\mathrm{safe}}$, then $\langle \mathbf{p}, \mathbf{b}^{\mathrm{safe}} \rangle \geq p_i b_i^{\mathrm{safe}} > R_{\max}$ while $\widehat{g}_{\theta,t}(\mathbf{p}) \geq 0$, so the objective exceeds $R_{\max}$. At $\mathbf{p} = \mathbf{0}$, the objective equals $\sum_\theta w_\theta \widehat{g}_{\theta,t}(\mathbf{0}) = \sum_\theta w_\theta \frac{1}{N_\theta} \sum r \leq R_{\max}$. Thus such $\mathbf{p}$ cannot be optimal. Apply coordinatewise. $\square$

### C.2. Nash Equilibrium Characterization

We prove that saddle points of zero-sum games are equivalent to Nash equilibria.

**Theorem 8** (Saddle Points $\Leftrightarrow$ Nash Equilibria). *For the zero-sum game with payoff function $L(\mathbf{w}, \mathbf{p})$, a pair $(\mathbf{w}^*, \mathbf{p}^*)$ is a saddle point of $L$ if and only if it is a Nash equilibrium.*

*Proof of Theorem 8.* Consider the zero-sum game with payoff $L(\mathbf{w}, \mathbf{p})$ where Player 1 receives $L(\mathbf{w}, \mathbf{p})$ and Player 2 receives $-L(\mathbf{w}, \mathbf{p})$.

**Definition (Saddle Point).** $(w^*, p^*)$ is a saddle point if:

$$L(\mathbf{w}, \mathbf{p}^*) \leq L(\mathbf{w}^*, \mathbf{p}^*) \leq L(\mathbf{w}^*, \mathbf{p}) \quad \forall \mathbf{w} \in \Delta_K, \forall \mathbf{p} \in \mathcal{P}.$$

**Definition (Nash Equilibrium).** $(w^*, p^*)$ is a Nash equilibrium if no player can improve by unilateral deviation:

1. $L(\mathbf{w}^*, \mathbf{p}^*) \geq L(\mathbf{w}, \mathbf{p}^*)$ for all $\mathbf{w} \in \Delta_K$ (Player 1 best-responds)

2. $-L(\mathbf{w}^*, \mathbf{p}^*) \geq -L(\mathbf{w}^*, \mathbf{p})$ for all $\mathbf{p} \in \mathcal{P}$ (Player 2 best-responds)

**($\Rightarrow$) Saddle Point implies Nash Equilibrium.**

Assume $(w^*, p^*)$ is a saddle point.

*Player 1's condition:* The left inequality gives $L(\mathbf{w}, \mathbf{p}^*) \leq L(\mathbf{w}^*, \mathbf{p}^*)$ for all $\mathbf{w}$, so $w^*$ maximizes $L(\cdot, \mathbf{p}^*)$. Player 1 has no profitable deviation.

*Player 2's condition:* The right inequality gives $L(\mathbf{w}^*, \mathbf{p}^*) \leq L(\mathbf{w}^*, \mathbf{p})$ for all $\mathbf{p}$, equivalently $-L(\mathbf{w}^*, \mathbf{p}) \leq -L(\mathbf{w}^*, \mathbf{p}^*)$. Thus $p^*$ minimizes $L(w^*, \cdot)$. Player 2 has no profitable deviation.

**($\Leftarrow$) Nash Equilibrium implies Saddle Point.**

Assume $(w^*, p^*)$ is a Nash equilibrium.

From Player 1's condition: $L(\mathbf{w}^*, \mathbf{p}^*) \geq L(\mathbf{w}, \mathbf{p}^*)$ for all $\mathbf{w}$, yielding the left saddle inequality.

From Player 2's condition: $-L(\mathbf{w}^*, \mathbf{p}^*) \geq -L(\mathbf{w}^*, \mathbf{p})$ for all $\mathbf{p}$, hence $L(\mathbf{w}^*, \mathbf{p}^*) \leq L(\mathbf{w}^*, \mathbf{p})$, yielding the right saddle inequality.

Combining: $L(\mathbf{w}, \mathbf{p}^*) \leq L(\mathbf{w}^*, \mathbf{p}^*) \leq L(\mathbf{w}^*, \mathbf{p})$ for all $\mathbf{w}, \mathbf{p}$. $\square$

**Remark 4** (Existence and Uniqueness). *By Sion's minimax theorem, a saddle point (and hence Nash equilibrium) exists when $\Delta_K$ and $\mathcal{P}$ are compact convex and $L$ is convex in $\mathbf{p}$ and linear (hence concave) in $\mathbf{w}$. The optimal mixture $\mathbf{w}^*$ may be unique (when budget constraints pin down resource utilization), while $\mathbf{p}^*$ may be non-unique (any price on the "tie surface" where configurations have equal surplus is optimal).*

### C.3. Pointwise Surplus Inequality

**Lemma 9** (Pointwise Surplus Inequality). *For any $\mathbf{p} \in \mathbb{R}^d_+$, any $(r, \mathbf{a}) \in \mathbb{R}_+ \times \mathbb{R}^d_+$, and any $x \in [0, 1]$:*

$$r \cdot x \leq \langle \mathbf{p}, \mathbf{a} \rangle \cdot x + (r - \langle \mathbf{p}, \mathbf{a} \rangle)_+.$$

*Proof.* Let $z := r - \langle \mathbf{p}, \mathbf{a} \rangle$. Then:

$$\langle \mathbf{p}, \mathbf{a} \rangle \cdot x + (r - \langle \mathbf{p}, \mathbf{a} \rangle)_+ = (r - z) \cdot x + z_+ = r \cdot x + (z_+ - z \cdot x).$$

We show $z_+ - z \cdot x \geq 0$:

- If $z \geq 0$: $z_+ - z \cdot x = z(1 - x) \geq 0$ since $x \leq 1$.

- If $z < 0$: $z_+ = 0$ and $z_+ - z \cdot x = -z \cdot x \geq 0$ since $z < 0$ and $x \geq 0$.

$\square$

### C.4. Optimal Acceptance Under Fixed Price

**Lemma 10** (Threshold Optimality). *Fix $\mathbf{p} \in \mathcal{P}$ and $\theta$. The optimization:*

$$\sup_{x_\theta : 0 \leq x_\theta \leq 1} \mathbb{E}_\theta \left[ (r - \langle \mathbf{p}, \mathbf{a} \rangle) \cdot x_\theta(r, \mathbf{a}) \right]$$

*equals $\mathbb{E}_\theta[(r - \langle \mathbf{p}, \mathbf{a} \rangle)_+] = g_\theta(\mathbf{p})$, achieved by the threshold rule $x_\theta(r, \mathbf{a}) = \mathbf{1}\{r > \langle \mathbf{p}, \mathbf{a} \rangle\}$ (ties arbitrary).*

*Proof.* For any fixed realization $(r, \mathbf{a})$, the quantity $(r - \langle \mathbf{p}, \mathbf{a} \rangle) \cdot x$ is maximized over $x \in [0, 1]$ by:

- $x = 1$ if $r - \langle \mathbf{p}, \mathbf{a} \rangle > 0$,

- $x = 0$ if $r - \langle \mathbf{p}, \mathbf{a} \rangle < 0$,

- Any $x \in [0, 1]$ if $r = \langle \mathbf{p}, \mathbf{a} \rangle$ (ties).

Thus the pointwise supremum equals $(r - \langle \mathbf{p}, \mathbf{a} \rangle)_+$, achieved by the threshold rule. Taking expectation yields $g_\theta(\mathbf{p})$. $\square$

### C.5. Proof of Theorem 1 (Primal–Dual Form)

*Proof of Theorem 1.* We prove:

$$V^{\mathrm{mix}}(\mathbf{b}) = \max_{\mathbf{w} \in \Delta_K} \min_{\mathbf{p} \in \mathcal{P}} \left\{ \langle \mathbf{p}, \mathbf{b} \rangle + \sum_\theta w_\theta g_\theta(\mathbf{p}) \right\} = \min_{\mathbf{p} \in \mathcal{P}} \left\{ \langle \mathbf{p}, \mathbf{b} \rangle + \max_\theta g_\theta(\mathbf{p}) \right\}.$$

**Step 1: Lagrangian dual for fixed w.** Fix any $\mathbf{w} \in \Delta_K$. The inner problem in the primal definition is:

$$V(\mathbf{w}) := \max_{\{x_\theta\}} \left\{ \sum_\theta w_\theta \mathbb{E}_\theta[r \cdot x_\theta] \,\Big|\, \sum_\theta w_\theta \mathbb{E}_\theta[\mathbf{a} \cdot x_\theta] \leq \mathbf{b} \right\}.$$

This is an infinite-dimensional linear program. Let $\mathcal{Z} := \mathbb{R}_+ \times \mathbb{R}^d_+$ denote the outcome space (reward-resource pairs). Each $x_\theta : \mathcal{Z} \to [0, 1]$ is a Borel-measurable acceptance function in $L^\infty(\mathcal{Z}, \mathcal{B}, \mathcal{D}_\theta)$. The objective and constraints are linear in $x_\theta$, and the Slater constraint qualification is satisfied since $x_\theta \equiv 0$ is strictly feasible: $\sum_\theta w_\theta \mathbb{E}_\theta[\mathbf{a} \cdot 0] = 0 \prec \mathbf{b}$ (componentwise strict inequality holds since $b_{\min} > 0$ by Assumption 3). The key observation is that while the primal space $L^\infty$ is infinite-dimensional, the constraint map $\{x_\theta\} \mapsto \sum_\theta w_\theta \mathbb{E}_\theta[\mathbf{a} \cdot x_\theta]$ takes values in $\mathbb{R}^d$, where the positive

cone $\mathbb{R}_+^d$ has non-empty interior. Under this structure, Slater's condition implies strong duality (see Luenberger (1969), Chapter 8, or Anderson & Nash (1987)):

$$V(\mathbf{w}) = \min_{\mathbf{p} \in \mathcal{P}} \left\{ \langle \mathbf{p}, \mathbf{b} \rangle + \sup_{\{x_\theta\}} \sum_\theta w_\theta \mathbb{E}_\theta[(r - \langle \mathbf{p}, \mathbf{a} \rangle) \cdot x_\theta] \right\}.$$

By Lemma 10, the inner supremum equals $\sum_\theta w_\theta g_\theta(\mathbf{p})$. Hence:

$$V(\mathbf{w}) = \min_{\mathbf{p} \in \mathcal{P}} \left\{ \langle \mathbf{p}, \mathbf{b} \rangle + \sum_\theta w_\theta g_\theta(\mathbf{p}) \right\}.$$

**Step 2: Maximize over w.** By definition, $V^{\mathrm{mix}}(\mathbf{b}) = \max_{\mathbf{w} \in \Delta_K} V(\mathbf{w})$, so:

$$V^{\mathrm{mix}}(\mathbf{b}) = \max_{\mathbf{w} \in \Delta_K} \min_{\mathbf{p} \in \mathcal{P}} \left\{ \langle \mathbf{p}, \mathbf{b} \rangle + \sum_\theta w_\theta g_\theta(\mathbf{p}) \right\}.$$

**Step 3: Minimax swap via Sion's theorem.** Define $\Phi(\mathbf{w}, \mathbf{p}) := \langle \mathbf{p}, \mathbf{b} \rangle + \sum_\theta w_\theta g_\theta(\mathbf{p})$. We verify the conditions for Sion's minimax theorem:

- $\Delta_K$ is compact and convex.

- $\mathcal{P} = [0, P_{\max}]^d$ is compact and convex.

- For fixed $\mathbf{p}$, $\Phi(\cdot, \mathbf{p})$ is linear (hence concave) in $\mathbf{w}$.

- For fixed $\mathbf{w}$, $\Phi(\mathbf{w}, \cdot)$ is convex in $\mathbf{p}$ because each $g_\theta(\mathbf{p}) = \mathbb{E}[(r - \langle \mathbf{p}, \mathbf{a} \rangle)_+]$ is convex (expectation of a convex function in $\mathbf{p}$).

- $\Phi$ is continuous and bounded by the boundedness assumptions.

By Sion's minimax theorem:

$$\max_{\mathbf{w} \in \Delta_K} \min_{\mathbf{p} \in \mathcal{P}} \Phi(\mathbf{w}, \mathbf{p}) = \min_{\mathbf{p} \in \mathcal{P}} \max_{\mathbf{w} \in \Delta_K} \Phi(\mathbf{w}, \mathbf{p}).$$

**Step 4: Simplify max over simplex.** For fixed $\mathbf{p}$:

$$\max_{\mathbf{w} \in \Delta_K} \sum_\theta w_\theta g_\theta(\mathbf{p}) = \max_\theta g_\theta(\mathbf{p}),$$

since a linear function over the simplex is maximized at an extreme point. Therefore:

$$V^{\mathrm{mix}}(\mathbf{b}) = \min_{\mathbf{p} \in \mathcal{P}} \left\{ \langle \mathbf{p}, \mathbf{b} \rangle + \max_\theta g_\theta(\mathbf{p}) \right\}.$$

$\square$

### C.6. Proof of Theorem 4 (Oracle Upper Bound)

*Proof of Theorem 4.* Let $\pi$ be any causal online policy satisfying $\sum_{t=1}^T \mathbf{a}_t x_t \le \mathbf{B}$ almost surely.

Fix any $\mathbf{p} \in \mathcal{P}$. Apply Lemma 9 at each time $t$ with $x = x_t$:

$$r_t x_t \le \langle \mathbf{p}, \mathbf{a}_t \rangle x_t + (r_t - \langle \mathbf{p}, \mathbf{a}_t \rangle)_+.$$

Summing over $t = 1, \ldots, T$:

$$R_T^\pi = \sum_{t=1}^T r_t x_t \le \left\langle \mathbf{p}, \sum_{t=1}^T \mathbf{a}_t x_t \right\rangle + \sum_{t=1}^T (r_t - \langle \mathbf{p}, \mathbf{a}_t \rangle)_+.$$

By pathwise feasibility:

$$R_T^\pi \leq \langle \mathbf{p}, \mathbf{B} \rangle + \sum_{t=1}^{T} (r_t - \langle \mathbf{p}, \mathbf{a}_t \rangle)_+.$$

Taking expectation:

$$\mathbb{E}[R_T^\pi] \leq \langle \mathbf{p}, \mathbf{B} \rangle + \sum_{t=1}^{T} \mathbb{E}\left[ (r_t - \langle \mathbf{p}, \mathbf{a}_t \rangle)_+ \right].$$

Condition on $\theta_t$. Since $(r_t, \mathbf{a}_t) \mid (\theta_t = \theta) \sim \mathcal{D}_\theta$:

$$\mathbb{E}\left[ (r_t - \langle \mathbf{p}, \mathbf{a}_t \rangle)_+ \mid \theta_t = \theta \right] = g_\theta(\mathbf{p}).$$

**Model assumption used:** The equality above uses Assumption 1, which ensures that conditional on $\theta_t = \theta$, the pair $(r_t, \mathbf{a}_t)$ is distributed according to $\mathcal{D}_\theta$.

Thus:

$$\mathbb{E}\left[ (r_t - \langle \mathbf{p}, \mathbf{a}_t \rangle)_+ \right] = \sum_\theta \Pr(\theta_t = \theta) \cdot g_\theta(\mathbf{p}).$$

Define the empirical mixture $\bar{w}_\theta := \frac{1}{T} \sum_{t=1}^{T} \Pr(\theta_t = \theta)$. Then $\bar{\mathbf{w}} \in \Delta_K$ and:

$$\frac{1}{T} \sum_{t=1}^{T} \mathbb{E}\left[ (r_t - \langle \mathbf{p}, \mathbf{a}_t \rangle)_+ \right] = \sum_\theta \bar{w}_\theta g_\theta(\mathbf{p}) \leq \max_\theta g_\theta(\mathbf{p}).$$

Therefore:

$$\mathbb{E}[R_T^\pi] \leq T \left( \langle \mathbf{p}, \mathbf{b} \rangle + \max_\theta g_\theta(\mathbf{p}) \right).$$

Minimizing over $\mathbf{p} \in \mathcal{P}$ and applying Theorem 1:

$$\mathbb{E}[R_T^\pi] \leq T \cdot V^{\mathrm{mix}}(\mathbf{b}).$$

$\square$

### C.7. No-Tie Condition and Subgradients

For the regret analysis, we require a technical assumption on online arrivals.

**Assumption 5** (No-Tie Condition). *For all $\theta \in \Theta$ and all $\mathbf{p} \in \mathcal{P}$,*

$$\Pr_{(r,\mathbf{a}) \sim \mathcal{D}_\theta} \left( r = \langle \mathbf{p}, \mathbf{a} \rangle \right) = 0.$$

See Remark 3 for why we use strict thresholds in the analysis.

**Lemma 11** (Subgradients and Tie-Weighted Consumption). *Fix any sample $(r, \mathbf{a})$ and define $\phi_{\mathbf{p}}(r, \mathbf{a}) := (r - \langle \mathbf{p}, \mathbf{a} \rangle)_+$. Then $\phi_{\mathbf{p}}$ is convex in $\mathbf{p}$ and its subdifferential is*

$$\partial_{\mathbf{p}} \phi_{\mathbf{p}}(r, \mathbf{a}) = \begin{cases} \{-\mathbf{a}\}, & r > \langle \mathbf{p}, \mathbf{a} \rangle, \\ \{-\lambda \mathbf{a} : \lambda \in [0,1]\}, & r = \langle \mathbf{p}, \mathbf{a} \rangle, \\ \{\mathbf{0}\}, & r < \langle \mathbf{p}, \mathbf{a} \rangle. \end{cases}$$

*Equivalently, for any $\lambda \in [0,1]$, the vector*

$$-\mathbf{a}\left( \mathbf{1}\{r > \langle \mathbf{p}, \mathbf{a} \rangle\} + \lambda \mathbf{1}\{r = \langle \mathbf{p}, \mathbf{a} \rangle\} \right)$$

*is a valid subgradient of $\phi_{\mathbf{p}}(r, \mathbf{a})$.*

*Now fix $\theta$ and time $t$ with samples $\mathcal{S}_\theta(t) = \{(r_j, \mathbf{a}_j)\}_{j=1}^{N_\theta(t)}$ and define*

$$\widehat{g}_{\theta,t}(\mathbf{p}) := \frac{1}{N_\theta(t)} \sum_{j=1}^{N_\theta(t)} (r_j - \langle \mathbf{p}, \mathbf{a}_j \rangle)_+.$$

*For any $\mathbf{p}$ and any tie-weight vector $\lambda \in [0,1]^{N_\theta(t)}$, define the tie-weighted empirical consumption*

$$\widehat{h}_{\theta,t}^\lambda(\mathbf{p}) := \frac{1}{N_\theta(t)} \sum_{j=1}^{N_\theta(t)} \mathbf{a}_j \Big( \mathbf{1}\{r_j > \langle \mathbf{p}, \mathbf{a}_j \rangle\} + \lambda_j \mathbf{1}\{r_j = \langle \mathbf{p}, \mathbf{a}_j \rangle\} \Big).$$

*Then*

$$\partial \widehat{g}_{\theta,t}(\mathbf{p}) = \Big\{ -\widehat{h}_{\theta,t}^\lambda(\mathbf{p}) : \lambda \in [0,1]^{N_\theta(t)} \Big\}.$$

*Moreover, componentwise for every $\mathbf{p}$ and $\lambda$,*

$$\widehat{h}_{\theta,t}^{>}(\mathbf{p}) \leq \widehat{h}_{\theta,t}^\lambda(\mathbf{p}) \leq \widehat{h}_{\theta,t}^{\geq}(\mathbf{p}),$$

*where $\widehat{h}_{\theta,t}^{>}(\mathbf{p})$ uses the strict indicator and $\widehat{h}_{\theta,t}^{\geq}(\mathbf{p})$ uses the weak indicator.*

*Proof.* The pointwise subdifferential formula is standard for the hinge composed with an affine map. For the empirical average $\widehat{g}_{\theta,t}$, the subdifferential is the average of the pointwise subdifferentials (finite sum of convex functions), hence it equals the set of averages of admissible pointwise subgradients; this is exactly the tie-weighted form above. The sandwich inequality holds because at a tie the contribution is $\lambda_j \mathbf{a}_j$ with $\lambda_j \in [0,1]$. $\square$

## C.8. Auxiliary Lemmas for Population Primal–Dual Analysis

**Lemma 12** (Upper Box Constraint is Inactive at Minimizers). *Assume $0 \leq r \leq R_{\max}$ a.s. and $b_{\min} > 0$. Fix any $\mathbf{w} \in \Delta_K$ and consider the convex function*

$$F_{\mathbf{w}}(\mathbf{p}) := \langle \mathbf{p}, \mathbf{b} \rangle + \sum_\theta w_\theta g_\theta(\mathbf{p}), \qquad \mathbf{p} \in \mathbb{R}_+^d.$$

*Then every minimizer of $\min_{\mathbf{p} \geq 0} F_{\mathbf{w}}(\mathbf{p})$ satisfies $\|\mathbf{p}\|_\infty \leq R_{\max}/b_{\min}$. In particular, if $P_{\max} > R_{\max}/b_{\min}$, then every minimizer of $\min_{\mathbf{p} \in \mathcal{P}} F_{\mathbf{w}}(\mathbf{p})$ lies in the strict interior of the upper box constraints, i.e., $p_i < P_{\max}$ for all $i$.*

*Proof.* Fix coordinate $i$ and suppose $p_i > R_{\max}/b_i$. Then

$$F_{\mathbf{w}}(\mathbf{p}) \geq \langle \mathbf{p}, \mathbf{b} \rangle \geq p_i b_i > R_{\max}.$$

On the other hand, at $\mathbf{p} = \mathbf{0}$,

$$F_{\mathbf{w}}(\mathbf{0}) = \sum_\theta w_\theta \mathbb{E}_\theta[r] \leq R_{\max}.$$

Hence no point with $p_i > R_{\max}/b_i$ can be optimal. Applying this argument to all $i$ gives $\|\mathbf{p}\|_\infty \leq R_{\max}/b_{\min}$. If $P_{\max} > R_{\max}/b_{\min}$, then $\|\mathbf{p}\|_\infty \leq R_{\max}/b_{\min}$ implies $\mathbf{p}$ cannot satisfy $p_i = P_{\max}$ for any $i$. $\square$

**Lemma 13** (Differentiability and Gradient Formula for $g_\theta$). *Assume Assumption 2 and Assumption 5 (no ties). Then for each $\theta$, $g_\theta$ is convex and continuously differentiable on $\mathcal{P}$, and*

$$\nabla g_\theta(\mathbf{p}) = -h_\theta(\mathbf{p}) = -\mathbb{E}_\theta[\mathbf{a}\, \mathbf{1}\{r > \langle \mathbf{p}, \mathbf{a} \rangle\}].$$

*Consequently, for any $\mathbf{w} \in \Delta_K$,*

$$\nabla_{\mathbf{p}} L(\mathbf{w}, \mathbf{p}) = \mathbf{b} - H(\mathbf{w}, \mathbf{p}).$$

*Proof.* Fix $\theta$ and define $\phi(\mathbf{p}; r, \mathbf{a}) := (r - \langle \mathbf{p}, \mathbf{a} \rangle)_+$, so $g_\theta(\mathbf{p}) = \mathbb{E}_\theta[\phi(\mathbf{p}; r, \mathbf{a})]$.

**Step 1 (convexity).** For each fixed $(r, \mathbf{a})$, $\mathbf{p} \mapsto r - \langle \mathbf{p}, \mathbf{a} \rangle$ is affine and $x \mapsto x_+$ is convex, hence $\mathbf{p} \mapsto \phi(\mathbf{p}; r, \mathbf{a})$ is convex. Expectation preserves convexity, so $g_\theta$ is convex.

**Step 2 (pointwise derivative).** Fix coordinate $i$ and consider the one-sided difference quotient:

$$D_t^{(i)}(r, \mathbf{a}) := \frac{\phi(\mathbf{p} + te_i; r, \mathbf{a}) - \phi(\mathbf{p}; r, \mathbf{a})}{t}, \qquad t \neq 0.$$

Because $x \mapsto x_+$ is 1-Lipschitz and the argument changes by $ta^{(i)}$, we have

$$|D_t^{(i)}(r, \mathbf{a})| \leq a^{(i)} \leq A_{\max} \qquad \text{a.s.}$$

Moreover, for any $(r, \mathbf{a})$ with $r \neq \langle \mathbf{p}, \mathbf{a} \rangle$ (which holds a.s. by Assumption 5), $\phi(\cdot; r, \mathbf{a})$ is differentiable at $\mathbf{p}$ and

$$\lim_{t \to 0} D_t^{(i)}(r, \mathbf{a}) = \frac{\partial}{\partial p_i} \phi(\mathbf{p}; r, \mathbf{a}) = \begin{cases} -a^{(i)}, & r > \langle \mathbf{p}, \mathbf{a} \rangle, \\ 0, & r < \langle \mathbf{p}, \mathbf{a} \rangle. \end{cases}$$

Equivalently,

$$\frac{\partial}{\partial p_i} \phi(\mathbf{p}; r, \mathbf{a}) = -a^{(i)} \mathbf{1}\{r > \langle \mathbf{p}, \mathbf{a} \rangle\} \qquad \text{a.s.}$$

**Step 3 (interchange derivative and expectation).** By the uniform bound $|D_t^{(i)}(r, \mathbf{a})| \leq A_{\max}$ and dominated convergence,

$$\frac{\partial}{\partial p_i} g_\theta(\mathbf{p}) = \frac{\partial}{\partial p_i} \mathbb{E}_\theta[\phi(\mathbf{p}; r, \mathbf{a})] = \mathbb{E}_\theta\left[\frac{\partial}{\partial p_i} \phi(\mathbf{p}; r, \mathbf{a})\right] = -\mathbb{E}_\theta\left[a^{(i)} \mathbf{1}\{r > \langle \mathbf{p}, \mathbf{a} \rangle\}\right].$$

Stacking coordinates yields $\nabla g_\theta(\mathbf{p}) = -h_\theta(\mathbf{p})$.

**Step 4 (continuity of the gradient).** Let $\mathbf{p}_n \to \mathbf{p}$. Then $\mathbf{1}\{r > \langle \mathbf{p}_n, \mathbf{a} \rangle\} \to \mathbf{1}\{r > \langle \mathbf{p}, \mathbf{a} \rangle\}$ pointwise for all $(r, \mathbf{a})$ such that $r \neq \langle \mathbf{p}, \mathbf{a} \rangle$; by Assumption 5 this holds a.s. Bounded convergence with $0 \leq a^{(i)} \leq A_{\max}$ implies $\mathbb{E}[a^{(i)} \mathbf{1}\{r > \langle \mathbf{p}_n, \mathbf{a} \rangle\}] \to \mathbb{E}[a^{(i)} \mathbf{1}\{r > \langle \mathbf{p}, \mathbf{a} \rangle\}]$, so $\nabla g_\theta$ is continuous on $\mathcal{P}$.

Finally, $\nabla_{\mathbf{p}} L(\mathbf{w}, \mathbf{p}) = \mathbf{b} + \sum_\theta w_\theta \nabla g_\theta(\mathbf{p}) = \mathbf{b} - H(\mathbf{w}, \mathbf{p})$. $\qquad \square$

**Lemma 14** (Nonnegative Dot Product Complementarity is Coordinatewise). *If $u, v \in \mathbb{R}_+^d$ and $\langle u, v \rangle = 0$, then $u_i v_i = 0$ for every $i \in [d]$.*

*Proof.* Each term $u_i v_i \geq 0$ and $\sum_i u_i v_i = 0$, hence each term must be zero. $\qquad \square$

### C.9. Proof of Theorem 2 (Saddle/KKT Optimality)

*Proof of Theorem 2.* Assume Assumptions 2–5, and assume $P_{\max} > R_{\max}/b_{\min}$ so that by Lemma 12 the upper box constraints are inactive at any minimizer of $\mathbf{p} \mapsto L(\mathbf{w}, \mathbf{p})$ for any fixed $\mathbf{w}$.

We prove the equivalence between:

(S) $(\mathbf{w}^\star, \mathbf{p}^\star)$ is a saddle point of $L$, i.e.,

$$L(\mathbf{w}, \mathbf{p}^\star) \leq L(\mathbf{w}^\star, \mathbf{p}^\star) \leq L(\mathbf{w}^\star, \mathbf{p}) \quad \forall \mathbf{w} \in \Delta_K, \forall \mathbf{p} \in \mathcal{P};$$

(KKT) conditions (i)–(iii) in the theorem statement.

**(S) $\Rightarrow$ (KKT).**

*Step 1 (Envelope support).* Fix $\mathbf{p}^\star$. The map $\mathbf{w} \mapsto L(\mathbf{w}, \mathbf{p}^\star) = \langle \mathbf{p}^\star, \mathbf{b} \rangle + \sum_\theta w_\theta g_\theta(\mathbf{p}^\star)$ is linear in $\mathbf{w}$ over the simplex. Therefore, any maximizer $\mathbf{w}^\star \in \arg\max_{\mathbf{w} \in \Delta_K} L(\mathbf{w}, \mathbf{p}^\star)$ must place all its mass on indices attaining the maximum coefficient $g_\theta(\mathbf{p}^\star)$:

$$\text{supp}(\mathbf{w}^\star) \subseteq A(\mathbf{p}^\star) := \arg\max_\theta g_\theta(\mathbf{p}^\star).$$

This is condition (i).

*Step 2 (First-order optimality for the $\mathbf{p}$-minimization).* Because $(\mathbf{w}^\star, \mathbf{p}^\star)$ is a saddle, $\mathbf{p}^\star \in \arg\min_{\mathbf{p}\in\mathcal{P}} L(\mathbf{w}^\star, \mathbf{p})$. By Lemma 12, $\mathbf{p}^\star$ lies in the interior of the upper box constraints, so the effective constraint set is only $\mathbf{p} \in \mathbb{R}_+^d$.

By Lemma 13, $L(\mathbf{w}^\star, \mathbf{p})$ is convex and differentiable in $\mathbf{p}$ with $\nabla_{\mathbf{p}} L(\mathbf{w}^\star, \mathbf{p}) = \mathbf{b} - H(\mathbf{w}^\star, \mathbf{p})$. The KKT condition for minimizing a convex differentiable function over $\mathbb{R}_+^d$ is

$$0 \in \nabla_{\mathbf{p}} L(\mathbf{w}^\star, \mathbf{p}^\star) + N_{\mathbb{R}_+^d}(\mathbf{p}^\star),$$

where $N_{\mathbb{R}_+^d}(\mathbf{p}^\star)$ is the normal cone to $\mathbb{R}_+^d$ at $\mathbf{p}^\star$. Equivalently, there exists $v \in N_{\mathbb{R}_+^d}(\mathbf{p}^\star)$ such that

$$0 = \mathbf{b} - H(\mathbf{w}^\star, \mathbf{p}^\star) + v.$$

The normal cone satisfies: $v_i = 0$ if $p_i^\star > 0$, and $v_i \leq 0$ if $p_i^\star = 0$. Thus:

$$p_i^\star > 0 \Rightarrow b_i - H_i(\mathbf{w}^\star, \mathbf{p}^\star) = 0, \qquad p_i^\star = 0 \Rightarrow b_i - H_i(\mathbf{w}^\star, \mathbf{p}^\star) \geq 0.$$

Hence $H(\mathbf{w}^\star, \mathbf{p}^\star) \leq \mathbf{b}$ componentwise, which is condition (ii). Also, $b_i - H_i(\mathbf{w}^\star, \mathbf{p}^\star) \geq 0$ and $p_i^\star \geq 0$ imply $\langle \mathbf{p}^\star, \mathbf{b} - H(\mathbf{w}^\star, \mathbf{p}^\star) \rangle \geq 0$. Moreover, because the coordinatewise implications above include $p_i^\star(b_i - H_i(\mathbf{w}^\star, \mathbf{p}^\star)) = 0$ for each $i$, we obtain

$$\langle \mathbf{p}^\star, \mathbf{b} - H(\mathbf{w}^\star, \mathbf{p}^\star) \rangle = 0,$$

which is condition (iii).

**(KKT) $\Rightarrow$ (S).**

Assume (i)–(iii).

*Step 1 ($\mathbf{w}^\star$ is a best response to $\mathbf{p}^\star$).* For fixed $\mathbf{p}^\star$, $\mathbf{w} \mapsto L(\mathbf{w}, \mathbf{p}^\star)$ is linear over $\Delta_K$, and its maximum value equals $\langle \mathbf{p}^\star, \mathbf{b} \rangle + \max_\theta g_\theta(\mathbf{p}^\star)$. Condition (i) implies $\sum_\theta w_\theta^\star g_\theta(\mathbf{p}^\star) = \max_\theta g_\theta(\mathbf{p}^\star)$, hence for all $\mathbf{w} \in \Delta_K$,

$$L(\mathbf{w}, \mathbf{p}^\star) \leq L(\mathbf{w}^\star, \mathbf{p}^\star).$$

*Step 2 ($\mathbf{p}^\star$ is a best response to $\mathbf{w}^\star$).* By Lemma 13, $L(\mathbf{w}^\star, \mathbf{p})$ is convex differentiable in $\mathbf{p}$. Since (ii) gives $\mathbf{b} - H(\mathbf{w}^\star, \mathbf{p}^\star) \in \mathbb{R}_+^d$ and (iii) gives $\langle \mathbf{p}^\star, \mathbf{b} - H(\mathbf{w}^\star, \mathbf{p}^\star) \rangle = 0$, Lemma 14 implies $p_i^\star(b_i - H_i(\mathbf{w}^\star, \mathbf{p}^\star)) = 0$ for each $i$. Equivalently:

$$p_i^\star > 0 \Rightarrow b_i - H_i(\mathbf{w}^\star, \mathbf{p}^\star) = 0, \qquad p_i^\star = 0 \Rightarrow b_i - H_i(\mathbf{w}^\star, \mathbf{p}^\star) \geq 0.$$

This is exactly the KKT condition $0 \in \nabla_{\mathbf{p}} L(\mathbf{w}^\star, \mathbf{p}^\star) + N_{\mathbb{R}_+^d}(\mathbf{p}^\star)$ for minimizing $L(\mathbf{w}^\star, \cdot)$ over $\mathbb{R}_+^d$ (upper box inactive by our choice of $P_{\max}$). Therefore $\mathbf{p}^\star \in \arg\min_{\mathbf{p}\in\mathcal{P}} L(\mathbf{w}^\star, \mathbf{p})$, and for all $\mathbf{p} \in \mathcal{P}$,

$$L(\mathbf{w}^\star, \mathbf{p}^\star) \leq L(\mathbf{w}^\star, \mathbf{p}).$$

Combining Steps 1 and 2 yields the saddle inequalities, so $(\mathbf{w}^\star, \mathbf{p}^\star)$ is a saddle point.

**Primal optimality of threshold admission and value identity.** Let $x(r, \mathbf{a}) := \mathbf{1}\{r > \langle \mathbf{p}^\star, \mathbf{a} \rangle\}$ and define the achieved (fluid) reward

$$U(\mathbf{w}^\star, \mathbf{p}^\star) := \sum_\theta w_\theta^\star \mathbb{E}_\theta[r\mathbf{1}\{r > \langle \mathbf{p}^\star, \mathbf{a} \rangle\}].$$

Using the identity $r\mathbf{1}\{r > \langle \mathbf{p}^\star, \mathbf{a} \rangle\} = \langle \mathbf{p}^\star, \mathbf{a} \rangle \mathbf{1}\{r > \langle \mathbf{p}^\star, \mathbf{a} \rangle\} + (r - \langle \mathbf{p}^\star, \mathbf{a} \rangle)_+$, we obtain

$$U(\mathbf{w}^\star, \mathbf{p}^\star) = \langle \mathbf{p}^\star, H(\mathbf{w}^\star, \mathbf{p}^\star) \rangle + \sum_\theta w_\theta^\star g_\theta(\mathbf{p}^\star).$$

Hence

$$L(\mathbf{w}^\star, \mathbf{p}^\star) = \langle \mathbf{p}^\star, \mathbf{b} \rangle + \sum_\theta w_\theta^\star g_\theta(\mathbf{p}^\star) = U(\mathbf{w}^\star, \mathbf{p}^\star) + \langle \mathbf{p}^\star, \mathbf{b} - H(\mathbf{w}^\star, \mathbf{p}^\star) \rangle.$$

By condition (iii), the last inner product is zero, so $U(\mathbf{w}^\star, \mathbf{p}^\star) = L(\mathbf{w}^\star, \mathbf{p}^\star)$. Condition (ii) implies the threshold rule is feasible for the primal constraint. Finally, since $(\mathbf{w}^\star, \mathbf{p}^\star)$ is a saddle point, $L(\mathbf{w}^\star, \mathbf{p}^\star)$ equals the minimax value $V^{\mathrm{mix}}(\mathbf{b})$ by Theorem 1. Therefore the mixture $\mathbf{w}^\star$ together with threshold admission attains $V^{\mathrm{mix}}(\mathbf{b})$ and is primal-optimal for (1).

$\square$

### C.10. Proof of Theorem 3 (All Saddle Points)

*Proof of Theorem 3.* Recall

$$f(\mathbf{p}) := \langle \mathbf{p}, \mathbf{b} \rangle + \max_{\theta \in \Theta} g_\theta(\mathbf{p}), \qquad \mathcal{P}^\star := \arg\min_{\mathbf{p} \in \mathcal{P}} f(\mathbf{p}),$$

and the candidate saddle set

$$\mathcal{S} = \Big\{ (\mathbf{w}, \mathbf{p}) \in \Delta_K \times \mathcal{P} : \mathbf{p} \in \mathcal{P}^\star, \mathrm{supp}(\mathbf{w}) \subseteq A(\mathbf{p}), H(\mathbf{w}, \mathbf{p}) \le \mathbf{b}, \langle \mathbf{p}, \mathbf{b} - H(\mathbf{w}, \mathbf{p}) \rangle = 0 \Big\}.$$

**Step 1 (Any saddle point lies in $\mathcal{S}$).** Let $(\mathbf{w}^\star, \mathbf{p}^\star)$ be any saddle point of $L$. By Theorem 2, we have $\mathrm{supp}(\mathbf{w}^\star) \subseteq A(\mathbf{p}^\star)$, $H(\mathbf{w}^\star, \mathbf{p}^\star) \le \mathbf{b}$, and $\langle \mathbf{p}^\star, \mathbf{b} - H(\mathbf{w}^\star, \mathbf{p}^\star) \rangle = 0$.

Moreover, $\mathrm{supp}(\mathbf{w}^\star) \subseteq A(\mathbf{p}^\star)$ implies $\sum_\theta w_\theta^\star g_\theta(\mathbf{p}^\star) = \max_\theta g_\theta(\mathbf{p}^\star)$, so

$$L(\mathbf{w}^\star, \mathbf{p}^\star) = \langle \mathbf{p}^\star, \mathbf{b} \rangle + \sum_\theta w_\theta^\star g_\theta(\mathbf{p}^\star) = \langle \mathbf{p}^\star, \mathbf{b} \rangle + \max_\theta g_\theta(\mathbf{p}^\star) = f(\mathbf{p}^\star).$$

Because $(\mathbf{w}^\star, \mathbf{p}^\star)$ is a saddle point, $L(\mathbf{w}^\star, \mathbf{p}^\star)$ equals the minimax value, and by Theorem 1 this value is $\min_{\mathbf{p} \in \mathcal{P}} f(\mathbf{p})$. Hence

$$f(\mathbf{p}^\star) = \min_{\mathbf{p} \in \mathcal{P}} f(\mathbf{p}),$$

so $\mathbf{p}^\star \in \mathcal{P}^\star$. Therefore $(\mathbf{w}^\star, \mathbf{p}^\star) \in \mathcal{S}$.

**Step 2 (Any point in $\mathcal{S}$ is a saddle point).** Now let $(\mathbf{w}, \mathbf{p}) \in \mathcal{S}$. Then $\mathrm{supp}(\mathbf{w}) \subseteq A(\mathbf{p})$, $H(\mathbf{w}, \mathbf{p}) \le \mathbf{b}$, and $\langle \mathbf{p}, \mathbf{b} - H(\mathbf{w}, \mathbf{p}) \rangle = 0$. These are exactly conditions (i)–(iii) of Theorem 2. Therefore $(\mathbf{w}, \mathbf{p})$ is a saddle point of $L$.

Combining Steps 1 and 2 proves that the set of all saddle points of $L$ is exactly $\mathcal{S}$. $\qquad \square$

## D. Uniform Concentration

We prove uniform concentration bounds for the empirical surplus and consumption functions.

### D.1. Concentration for Surplus Function

**Lemma 15** (Uniform Concentration for $g_\theta$ (Fixed $n$)). *Fix $\theta$ and $n \ge 1$. Let $\{(r_i, \mathbf{a}_i)\}_{i=1}^n$ be i.i.d. samples from $\mathcal{D}_\theta$ and define*

$$\widehat{g}_{\theta,n}(\mathbf{p}) := \frac{1}{n} \sum_{i=1}^n (r_i - \langle \mathbf{p}, \mathbf{a}_i \rangle)_+, \qquad g_\theta(\mathbf{p}) := \mathbb{E}_\theta[(r - \langle \mathbf{p}, \mathbf{a} \rangle)_+].$$

*There exist absolute constants $c_g, c_0 > 0$ such that for any $\delta \in (0,1)$, with probability at least $1 - \delta$,*

$$\sup_{\mathbf{p} \in \mathcal{P}} \big| \widehat{g}_{\theta,n}(\mathbf{p}) - g_\theta(\mathbf{p}) \big| \le c_g R_{\max} \sqrt{\frac{d \log\Big( \frac{c_0 d P_{\max} A_{\max} n}{R_{\max}} \Big) + \log(2/\delta)}{n}}.$$

*Proof.* **Step 1 (Pointwise Hoeffding).** For fixed $\mathbf{p}$, the random variable $(r - \langle \mathbf{p}, \mathbf{a} \rangle)_+$ lies in $[0, R_{\max}]$. Thus by Hoeffding,

$$\Pr\big( \big| \widehat{g}_{\theta,n}(\mathbf{p}) - g_\theta(\mathbf{p}) \big| > \epsilon \big) \le 2 \exp\Big( -\frac{2n\epsilon^2}{R_{\max}^2} \Big).$$

**Step 2 (Lipschitzness in p).** For any $\mathbf{p}, \boldsymbol{q} \in \mathcal{P}$ and any $(r, \mathbf{a})$,

$$\big| (r - \langle \mathbf{p}, \mathbf{a} \rangle)_+ - (r - \langle \boldsymbol{q}, \mathbf{a} \rangle)_+ \big| \le |\langle \mathbf{p} - \boldsymbol{q}, \mathbf{a} \rangle| \le A_{\max} \|\mathbf{p} - \boldsymbol{q}\|_1,$$

using $\|\mathbf{a}\|_\infty \le A_{\max}$. Hence both $\widehat{g}_{\theta,n}$ and $g_\theta$ are $A_{\max}$-Lipschitz in $\ell_1$.

**Step 3 ($\ell_1$-net + union bound).** Let $\eta > 0$ and let $\mathcal{N}$ be an $\ell_1$-net of $\mathcal{P} = [0, P_{\max}]^d$ with radius $\eta$. A standard volume argument gives $|\mathcal{N}| \leq \left(\frac{c_0 d P_{\max}}{\eta}\right)^d$ for an absolute constant $c_0 > 0$.

For any $\mathbf{p} \in \mathcal{P}$, pick $\boldsymbol{q} \in \mathcal{N}$ with $\|\mathbf{p} - \boldsymbol{q}\|_1 \leq \eta$. By Lipschitzness,

$$\left|\widehat{g}_{\theta,n}(\mathbf{p}) - g_\theta(\mathbf{p})\right| \leq \left|\widehat{g}_{\theta,n}(\boldsymbol{q}) - g_\theta(\boldsymbol{q})\right| + 2A_{\max}\eta.$$

Set $\eta = \epsilon/(4A_{\max})$. Then

$$\Pr\left(\sup_{\mathbf{p} \in \mathcal{P}} \left|\widehat{g}_{\theta,n}(\mathbf{p}) - g_\theta(\mathbf{p})\right| > \epsilon\right) \leq \Pr\left(\max_{\boldsymbol{q} \in \mathcal{N}} \left|\widehat{g}_{\theta,n}(\boldsymbol{q}) - g_\theta(\boldsymbol{q})\right| > \epsilon/2\right)$$

$$\leq \sum_{\boldsymbol{q} \in \mathcal{N}} \Pr\left(\left|\widehat{g}_{\theta,n}(\boldsymbol{q}) - g_\theta(\boldsymbol{q})\right| > \epsilon/2\right) \leq 2|\mathcal{N}| \exp\left(-\frac{2n(\epsilon/2)^2}{R_{\max}^2}\right) = 2|\mathcal{N}| \exp\left(-\frac{n\epsilon^2}{2R_{\max}^2}\right).$$

Substitute $|\mathcal{N}| \leq \left(\frac{4c_0 d P_{\max} A_{\max}}{\epsilon}\right)^d$ and solve for $\epsilon$ so that the RHS is at most $\delta$. This yields the stated bound. $\qquad\square$

**Corollary 16** (Anytime Version for Adaptive Sample Sizes). *Fix $\theta$ and horizon $T$. There exists an absolute constant $c'_g > 0$ such that for any $\delta \in (0, 1)$, with probability at least $1 - \delta$, the following holds simultaneously for all $n \in \{1, \ldots, T\}$:*

$$\sup_{\mathbf{p} \in \mathcal{P}} \left|\widehat{g}_{\theta,n}(\mathbf{p}) - g_\theta(\mathbf{p})\right| \leq c'_g R_{\max} \sqrt{\frac{d \log\left(\frac{c_0 d P_{\max} A_{\max} T}{R_{\max}}\right) + \log(2T/\delta)}{n}}.$$

*Consequently, at any time $t \leq T$, the bound holds with $n = N_\theta(t)$ even when $N_\theta(t)$ is chosen adaptively.*

*Proof.* Apply Lemma 15 with confidence level $\delta/T$ and take a union bound over $n = 1, \ldots, T$. $\qquad\square$

### D.2. Concentration for Threshold Consumption

The threshold consumption function involves weighted indicators $a^{(i)} \cdot \mathbf{1}\{r > \langle \mathbf{p}, \mathbf{a}\rangle\}$, which is not a pure indicator class. We use pseudo-dimension to obtain uniform concentration.

**Lemma 17** (Pseudo-Dimension of Weighted Threshold Consumption). *Fix a coordinate $i \in [d]$ and define the function class*

$$\mathcal{F}_i := \left\{f_{\mathbf{p}}(r, \mathbf{a}) := a^{(i)} \mathbf{1}\{r > \langle \mathbf{p}, \mathbf{a}\rangle\} : \mathbf{p} \in \mathcal{P}\right\}.$$

*Then $\mathrm{Pdim}(\mathcal{F}_i) \leq d + 2$.*

*Proof.* Recall the definition of pseudo-dimension: $\mathcal{F}_i$ pseudo-shatters $n$ points $z_1, \ldots, z_n$ if there exist thresholds $s_1, \ldots, s_n$ such that for every labeling $S \subseteq [n]$ there exists $f \in \mathcal{F}_i$ with $f(z_j) > s_j$ iff $j \in S$.

Take any candidate set $\{z_j = (r_j, \mathbf{a}_j)\}_{j=1}^n$ and thresholds $\{s_j\}$.

If $s_j < 0$, then $f_{\mathbf{p}}(z_j) \geq 0 > s_j$ for all $\mathbf{p}$, so the label of $j$ cannot vary across $S$. If $s_j \geq a_j^{(i)}$, then $f_{\mathbf{p}}(z_j) \leq a_j^{(i)} \leq s_j$ for all $\mathbf{p}$, so again the label cannot vary. Therefore, for a point to be label-flexible under pseudo-shattering, it must satisfy $0 \leq s_j < a_j^{(i)}$.

For such $j$, we have:

$$f_{\mathbf{p}}(z_j) > s_j \iff a_j^{(i)} \mathbf{1}\{r_j > \langle \mathbf{p}, \mathbf{a}_j\rangle\} > s_j \iff \mathbf{1}\{r_j > \langle \mathbf{p}, \mathbf{a}_j\rangle\} = 1.$$

Thus, on the subset of flexible points, pseudo-shattering by $\mathcal{F}_i$ reduces exactly to shattering by the halfspace indicator class

$$\mathcal{H} := \{(r, \mathbf{a}) \mapsto \mathbf{1}\{r > \langle \mathbf{p}, \mathbf{a}\rangle\} : \mathbf{p} \in \mathcal{P}\},$$

which is a class of halfspaces in $\mathbb{R}^{d+1}$ and satisfies $\mathrm{VCdim}(\mathcal{H}) \leq d + 2$. Therefore no more than $d + 2$ flexible points can be shattered, and hence $\mathrm{Pdim}(\mathcal{F}_i) \leq d + 2$. $\qquad\square$

**Theorem 18** (Uniform Deviation Bound for Pseudo-Dimension Classes). *Let $\mathcal{F}$ be a class of functions mapping into $[0,1]$ with pseudo-dimension $v$. Let $Z_1, \ldots, Z_n$ be i.i.d. samples from any distribution. Then for any $\delta \in (0,1)$, with probability at least $1 - \delta$,*

$$\sup_{f \in \mathcal{F}} \left| \mathbb{E}[f(Z)] - \frac{1}{n} \sum_{j=1}^{n} f(Z_j) \right| \leq c \sqrt{\frac{v \log(en) + \log(1/\delta)}{n}},$$

*for a universal constant $c > 0$.*

This is a standard result in statistical learning theory (see, e.g., Pollard (1984)).

**Lemma 19** (Uniform Concentration for $h_\theta$ (Strict and Weak Thresholds)). *Fix $\theta$, $n \geq 1$, and coordinate $i \in [d]$. Let $\{(r_j, \mathbf{a}_j)\}_{j=1}^{n}$ be i.i.d. from $\mathcal{D}_\theta$. Define the strict and weak empirical consumptions*

$$\widehat{h}_{\theta,n}^{>,(i)}(\mathbf{p}) := \frac{1}{n} \sum_{j=1}^{n} a_j^{(i)} \mathbf{1}\{r_j > \langle \mathbf{p}, \mathbf{a}_j \rangle\}, \qquad \widehat{h}_{\theta,n}^{\geq,(i)}(\mathbf{p}) := \frac{1}{n} \sum_{j=1}^{n} a_j^{(i)} \mathbf{1}\{r_j \geq \langle \mathbf{p}, \mathbf{a}_j \rangle\},$$

*and the population consumption (using strict threshold)*

$$h_\theta^{(i)}(\mathbf{p}) := \mathbb{E}_\theta\Big[ a^{(i)} \mathbf{1}\{r > \langle \mathbf{p}, \mathbf{a} \rangle\} \Big].$$

*Assume Assumption 5 (so that $\Pr(r = \langle \mathbf{p}, \mathbf{a} \rangle) = 0$ for all $\mathbf{p} \in \mathcal{P}$), hence $\mathbb{E}[a^{(i)} \mathbf{1}\{r \geq \langle \mathbf{p}, \mathbf{a} \rangle\}] = h_\theta^{(i)}(\mathbf{p})$ as well. Then there exists an absolute constant $c_h > 0$ such that for any $\delta \in (0,1)$, with probability at least $1 - \delta$,*

$$\sup_{\mathbf{p} \in \mathcal{P}} \left| \widehat{h}_{\theta,n}^{>,(i)}(\mathbf{p}) - h_\theta^{(i)}(\mathbf{p}) \right| \leq c_h A_{\max} \sqrt{\frac{(d+2) \log(en) + \log(4/\delta)}{n}},$$

*and simultaneously*

$$\sup_{\mathbf{p} \in \mathcal{P}} \left| \widehat{h}_{\theta,n}^{\geq,(i)}(\mathbf{p}) - h_\theta^{(i)}(\mathbf{p}) \right| \leq c_h A_{\max} \sqrt{\frac{(d+2) \log(en) + \log(4/\delta)}{n}}.$$

*Proof.* Both function classes $\{(r, \mathbf{a}) \mapsto a^{(i)} \mathbf{1}\{r > \langle \mathbf{p}, \mathbf{a} \rangle\} : \mathbf{p} \in \mathcal{P}\}$ and $\{(r, \mathbf{a}) \mapsto a^{(i)} \mathbf{1}\{r \geq \langle \mathbf{p}, \mathbf{a} \rangle\} : \mathbf{p} \in \mathcal{P}\}$ have pseudo-dimension at most $d + 2$ by the same argument as Lemma 17 (the strict vs. weak inequality does not change VC/pseudo-dimension). After scaling by $A_{\max}$ the functions map into $[0,1]$. Apply Theorem 18 to each class with confidence $\delta/2$ and union bound. Assumption 5 ensures both expectations coincide with $h_\theta^{(i)}(\mathbf{p})$. $\square$

**Corollary 20** (Anytime Version for Adaptive Sample Sizes). *Fix $\theta$ and horizon $T$. There exists a constant $c_h' > 0$ such that for any $\delta \in (0,1)$, with probability at least $1 - \delta$, the bounds in Lemma 19 hold simultaneously for all $n \in \{1, \ldots, T\}$ (for both strict and weak thresholds) with the RHS replaced by*

$$c_h' A_{\max} \sqrt{\frac{(d+2) \log(eT) + \log(4Td/\delta)}{n}}.$$

*Consequently, at any time $t \leq T$, the bound holds with $n = N_\theta(t)$ under adaptive sampling.*

*Proof.* Apply Lemma 19 with confidence $\delta/T$ and union bound over $n \leq T$ (and over coordinates $i$ if desired). $\square$

### D.3. Good Event

**Good Event Definition.** Fix a global confidence $\delta_{\text{tot}} := T^{-2}$. For each $\theta$, apply Corollary 16 with confidence $\delta_g := \delta_{\text{tot}}/(2K)$ and apply Corollary 20 with confidence $\delta_h := \delta_{\text{tot}}/(2K)$ (and union bound over coordinates $i \in [d]$ inside the corollary if desired). Let $\mathcal{E}$ be the event that for every $\theta$ and every $n \leq T$ simultaneously:

- $\sup_{\mathbf{p} \in \mathcal{P}} |\widehat{g}_{\theta,n}(\mathbf{p}) - g_\theta(\mathbf{p})| \leq \beta_{g,\theta}(n)$,

- for every coordinate $i$, both strict and weak empirical consumptions satisfy $\sup_{\mathbf{p} \in \mathcal{P}} |\widehat{h}_{\theta,n}^{>,(i)}(\mathbf{p}) - h_\theta^{(i)}(\mathbf{p})| \leq \beta_{h,\theta}(n)$ and $\sup_{\mathbf{p} \in \mathcal{P}} |\widehat{h}_{\theta,n}^{\geq,(i)}(\mathbf{p}) - h_\theta^{(i)}(\mathbf{p})| \leq \beta_{h,\theta}(n)$.

Then $\Pr(\mathcal{E}^c) \leq \delta_{\text{tot}}$.

**Confidence Radii.** For definiteness, one may take (for $n \geq 1$)

$$\beta_{g,\theta}(n) := c'_g R_{\max} \sqrt{\frac{d \log\left(\frac{c_0 d P_{\max} A_{\max} T}{R_{\max}}\right) + \log(2KT/\delta_{\text{tot}})}{n}},$$

$$\beta_{h,\theta}(n) := c'_h A_{\max} \sqrt{\frac{(d+2) \log(eT) + \log(4KTd/\delta_{\text{tot}})}{n}}.$$

In the regret proof, we evaluate these at $n = N_\theta(t)$.

# E. Regret Proof for the Main Theorem

We prove the main theorem following a five-step decomposition.

## E.1. Setup

Let $\varepsilon = \sqrt{\log T / T}$ and $\mathbf{b}^{\text{safe}} = (1 - \varepsilon)\mathbf{b}$. For mixture $\mathbf{w} \in \Delta_K$, define:

$$V(\mathbf{w}) := \min_{\mathbf{p} \in \mathcal{P}} \left\{ \langle \mathbf{p}, \mathbf{b}^{\text{safe}} \rangle + \sum_\theta w_\theta g_\theta(\mathbf{p}) \right\}, \quad V^{\text{mix}}(\mathbf{b}^{\text{safe}}) = \max_{\mathbf{w} \in \Delta_K} V(\mathbf{w}).$$

Define the **optimistic mixed value**:

$$\widehat{V}_t^{\text{opt}}(\mathbf{w}) := \min_{\mathbf{p} \in \mathcal{P}} \left\{ \langle \mathbf{p}, \mathbf{b}^{\text{safe}} \rangle + \sum_\theta w_\theta \big(\widehat{g}_{\theta,t}(\mathbf{p}) + \beta_{g,\theta}(t)\big) \right\}.$$

The algorithm chooses $\mathbf{w}_t \in \arg\max_{\mathbf{w} \in \Delta_K} \widehat{V}_t^{\text{opt}}(\mathbf{w})$.

**Confidence Radii Used in the Analysis.** Fix $\delta_{\text{tot}} := T^{-2}$. Let $\beta_{g,\theta}(t)$ be any sequence satisfying $\sup_{\mathbf{p} \in \mathcal{P}} |\widehat{g}_{\theta,t}(\mathbf{p}) - g_\theta(\mathbf{p})| \leq \beta_{g,\theta}(t)$ on $\mathcal{E}$. For concreteness, by Corollary 16 one may take

$$\beta_{g,\theta}(t) := \alpha \, c'_g R_{\max} \sqrt{\frac{d \log\left(\frac{c_0 d P_{\max} A_{\max} T}{R_{\max}}\right) + \log\left(\frac{2KT}{\delta_{\text{tot}}}\right)}{N_\theta(t) \vee 1}},$$

with any fixed $\alpha \geq 1$.

Similarly, by Corollary 20 one may take

$$\beta_{h,\theta}(t) := c'_h A_{\max} \sqrt{\frac{(d+2) \log(eT) + \log\left(\frac{4KTd}{\delta_{\text{tot}}}\right)}{N_\theta(t) \vee 1}}.$$

## E.2. Step 1: Good Event

On the good event $\mathcal{E}$ (probability $\geq 1 - O(1/T)$):

- $\sup_{\mathbf{p} \in \mathcal{P}} |\widehat{g}_{\theta,t}(\mathbf{p}) - g_\theta(\mathbf{p})| \leq \beta_{g,\theta}(t)$ for all $\theta, t$.

- $\sup_{\mathbf{p} \in \mathcal{P}} |\widehat{h}_{\theta,t}^{(i)}(\mathbf{p}) - h_\theta^{(i)}(\mathbf{p})| \leq \beta_{h,\theta}(t)$ for all $\theta, t, i$.

The bad event contributes at most

$$\mathbb{E}\big[\text{Reg}^{\text{mix}}(T)\mathbf{1}_{\mathcal{E}^c}\big] \leq TR_{\max} \Pr(\mathcal{E}^c) \leq TR_{\max} \cdot T^{-2} = \frac{R_{\max}}{T},$$

which is negligible compared to the main terms (and can be absorbed into constants).

**Lemma 21** (Mixture-Weight Bridge). *Since $\theta_t \sim \mathbf{w}_t$ and $\beta_\theta(t)$ is $\mathcal{F}_{t-1}$-measurable,*

$$\mathbb{E}[\beta_{\theta_t}(t) \mid \mathcal{F}_{t-1}] = \sum_\theta w_{t,\theta}\beta_\theta(t).$$

*Therefore, for any realization,*

$$\mathbb{E}\left[\sum_{t=1}^T \beta_{\theta_t}(t)\right] = \mathbb{E}\left[\sum_{t=1}^T \sum_\theta w_{t,\theta}\beta_\theta(t)\right],$$

*and standard concentration bounds apply to the realized arm sequence $\{\theta_t\}_{t=1}^T$.*

*Proof.* Fix any round $t$. Since $\mathbf{w}_t = (w_{t,1}, \dots, w_{t,K})$ is determined by the history up to round $t-1$ (i.e., $\mathbf{w}_t \in \mathcal{F}_{t-1}$) and $\theta_t$ is drawn from the categorical distribution with weights $\mathbf{w}_t$, we have by definition:

$$\mathbb{E}[\beta_{\theta_t}(t) \mid \mathcal{F}_{t-1}] = \sum_{\theta=1}^K \Pr(\theta_t = \theta \mid \mathcal{F}_{t-1}) \cdot \beta_\theta(t) = \sum_{\theta=1}^K w_{t,\theta}\beta_\theta(t).$$

Summing over $t$ and taking total expectation via the tower property yields the result. This lemma bridges the gap between our algorithm's mixture sampling and classical chosen-arm concentration lemmas, justifying why bounds on $\sum_\theta w_{t,\theta}\beta_\theta(t)$ translate to bounds on $\sum_t \beta_{\theta_t}(t)$. □

### E.3. Step 2: Mixture-Value Regret

**Lemma 22** (Optimism and True Value). *On $\mathcal{E}$, for all $t \leq T$ and all $\mathbf{w} \in \Delta_K$:*

$$V(\mathbf{w}) \leq \widehat{V}_t^{\mathrm{opt}}(\mathbf{w}) \leq V(\mathbf{w}) + 2\sum_\theta w_\theta \beta_{g,\theta}(t).$$

*In particular, $\widehat{V}_t^{\mathrm{opt}}(\mathbf{w})$ is an optimistic upper bound on $V(\mathbf{w})$.*

*Proof.* For any $\mathbf{p}$, on $\mathcal{E}$:

$$\sum_\theta w_\theta g_\theta(\mathbf{p}) \leq \sum_\theta w_\theta(\widehat{g}_{\theta,t}(\mathbf{p}) + \beta_{g,\theta}(t)).$$

Adding $\langle \mathbf{p}, \mathbf{b}^{\mathrm{safe}} \rangle$ and taking $\min_{\mathbf{p} \in \mathcal{P}}$ gives the lower bound $V(\mathbf{w}) \leq \widehat{V}_t^{\mathrm{opt}}(\mathbf{w})$.

For the upper bound, on $\mathcal{E}$:

$$\sum_\theta w_\theta(\widehat{g}_{\theta,t}(\mathbf{p}) + \beta_{g,\theta}(t)) \leq \sum_\theta w_\theta(g_\theta(\mathbf{p}) + 2\beta_{g,\theta}(t)).$$

Adding $\langle \mathbf{p}, \mathbf{b}^{\mathrm{safe}} \rangle$ and taking $\min_{\mathbf{p} \in \mathcal{P}}$ gives the result. □

Let $\mathbf{w}^\star \in \arg\max_{\mathbf{w}} V(\mathbf{w})$ so that $V(\mathbf{w}^\star) = V^{\mathrm{mix}}(\mathbf{b}^{\mathrm{safe}})$.

**Lemma 23** (Per-Round Mixture-Value Gap). *On $\mathcal{E}$, for all $t \leq T$:*

$$V^{\mathrm{mix}}(\mathbf{b}^{\mathrm{safe}}) - V(\mathbf{w}_t) \leq 2\sum_\theta w_{t,\theta}\beta_{g,\theta}(t).$$

*Proof.* Since $\mathbf{w}_t$ maximizes $\widehat{V}_t^{\mathrm{opt}}(\mathbf{w})$ over $\Delta_K$:

$$V(\mathbf{w}^\star) \leq \widehat{V}_t^{\mathrm{opt}}(\mathbf{w}^\star) \leq \widehat{V}_t^{\mathrm{opt}}(\mathbf{w}_t),$$

where the first inequality uses Lemma 22. By the upper bound in Lemma 22:

$$\widehat{V}_t^{\mathrm{opt}}(\mathbf{w}_t) \leq V(\mathbf{w}_t) + 2\sum_\theta w_{t,\theta}\beta_{g,\theta}(t).$$

Combining gives the result. □

**E.4. Step 3: Admission-Price Error and KKT Feasibility**

Define the unconstrained threshold acceptance $\tilde{x}_t := \mathbf{1}\{r_t > \langle \mathbf{p}_t, \mathbf{a}_t \rangle\}$ and reward $\tilde{R}_T := \sum_{t=1}^T r_t \tilde{x}_t$.

For any mixture $\mathbf{w}$ and price $\mathbf{p}$, define (using strict threshold $>$ per our convention):

$$U(\mathbf{w}, \mathbf{p}) := \sum_\theta w_\theta \mathbb{E}_\theta[r \cdot \mathbf{1}\{r > \langle \mathbf{p}, \mathbf{a} \rangle\}],$$

$$H(\mathbf{w}, \mathbf{p}) := \sum_\theta w_\theta h_\theta(\mathbf{p}),$$

$$\widehat{H}_t(\mathbf{w}, \mathbf{p}) := \sum_\theta w_\theta \widehat{h}_{\theta,t}(\mathbf{p}).$$

**Lemma 24** (Primal–Dual Inequality for Threshold Rules). *For any mixture $\mathbf{w}$ and price $\mathbf{p} \in \mathbb{R}_+^d$:*

$$V(\mathbf{w}) - U(\mathbf{w}, \mathbf{p}) \le \langle \mathbf{p}, \mathbf{b}^{\mathrm{safe}} - H(\mathbf{w}, \mathbf{p}) \rangle.$$

*Proof.* Fix any mixture $\mathbf{w}$ and price $\mathbf{p} \in \mathbb{R}_+^d$.

**Step 1 (dual upper bound on $V(\mathbf{w})$).** By the dual representation of $V(\mathbf{w})$ in Theorem 1, for any fixed $\mathbf{p}$ we have

$$V(\mathbf{w}) = \min_{\boldsymbol{q} \in \mathcal{P}} \left\{ \langle \boldsymbol{q}, \mathbf{b}^{\mathrm{safe}} \rangle + \sum_\theta w_\theta g_\theta(\boldsymbol{q}) \right\} \le \langle \mathbf{p}, \mathbf{b}^{\mathrm{safe}} \rangle + \sum_\theta w_\theta g_\theta(\mathbf{p}).$$

**Step 2 (identity for the threshold value).** For strict-threshold admission at price $\mathbf{p}$,

$$r\,\mathbf{1}\{r > \langle \mathbf{p}, \mathbf{a} \rangle\} = \langle \mathbf{p}, \mathbf{a} \rangle\,\mathbf{1}\{r > \langle \mathbf{p}, \mathbf{a} \rangle\} + (r - \langle \mathbf{p}, \mathbf{a} \rangle)_+.$$

Taking expectation under $\mathcal{D}_\theta$ and summing with weights $w_\theta$ yields

$$U(\mathbf{w}, \mathbf{p}) = \sum_\theta w_\theta \mathbb{E}_\theta[r\,\mathbf{1}\{r > \langle \mathbf{p}, \mathbf{a} \rangle\}] = \langle \mathbf{p}, H(\mathbf{w}, \mathbf{p}) \rangle + \sum_\theta w_\theta g_\theta(\mathbf{p}),$$

where $H(\mathbf{w}, \mathbf{p}) = \sum_\theta w_\theta h_\theta(\mathbf{p})$ is the strict-threshold expected consumption.

**Step 3 (combine).** Subtract the expression for $U(\mathbf{w}, \mathbf{p})$ from the dual upper bound on $V(\mathbf{w})$:

$$V(\mathbf{w}) - U(\mathbf{w}, \mathbf{p}) \le \langle \mathbf{p}, \mathbf{b}^{\mathrm{safe}} \rangle + \sum_\theta w_\theta g_\theta(\mathbf{p}) - \left( \langle \mathbf{p}, H(\mathbf{w}, \mathbf{p}) \rangle + \sum_\theta w_\theta g_\theta(\mathbf{p}) \right) = \langle \mathbf{p}, \mathbf{b}^{\mathrm{safe}} - H(\mathbf{w}, \mathbf{p}) \rangle.$$

$\square$

The following lemma establishes drift control via KKT conditions using tie-weighted consumption.

**Lemma 25** (KKT Implies Tie-Weighted Empirical Complementarity). *Fix a mixture $\mathbf{w} \in \Delta_K$ and time $t$ with $N_\theta(t) \ge 1$ for all $\theta$. Consider the convex problem*

$$\min_{\mathbf{p} \in \mathcal{P}} \left\{ \langle \mathbf{p}, \mathbf{b}^{\mathrm{safe}} \rangle + \sum_\theta w_\theta \widehat{g}_{\theta,t}(\mathbf{p}) \right\}.$$

*Assume $P_{\max} > R_{\max}/b_{\min}^{\mathrm{safe}}$ so that the upper box constraint is non-binding at minimizers (Corollary 7). Let $\mathbf{p}^\star(\mathbf{w})$ be any minimizer. Then there exist tie-weight vectors $\lambda_\theta \in [0,1]^{N_\theta(t)}$ (one per configuration) such that, with the tie-weighted empirical consumptions $\widehat{h}_{\theta,t}^{\lambda_\theta}(\mathbf{p}^\star)$ from Lemma 11, the mixture tie-weighted empirical consumption*

$$\widehat{H}_t^{\mathrm{kkt}}(\mathbf{w}, \mathbf{p}^\star) := \sum_\theta w_\theta \widehat{h}_{\theta,t}^{\lambda_\theta}(\mathbf{p}^\star)$$

*satisfies*

$$\widehat{H}_t^{\mathrm{kkt}}(\mathbf{w}, \mathbf{p}^\star) \le \mathbf{b}^{\mathrm{safe}} \quad \textit{(componentwise)} \qquad \textit{and} \qquad \langle \mathbf{p}^\star, \mathbf{b}^{\mathrm{safe}} - \widehat{H}_t^{\mathrm{kkt}}(\mathbf{w}, \mathbf{p}^\star) \rangle = 0.$$

*Proof.* By optimality of $\mathbf{p}^\star$ for the constrained convex problem with effective domain $\mathbf{p} \in \mathbb{R}^d_+$ (upper box non-binding), the KKT condition is

$$0 \in \mathbf{b}^{\text{safe}} + \sum_\theta w_\theta\, \partial\widehat{g}_{\theta,t}(\mathbf{p}^\star) + N_{\mathbb{R}^d_+}(\mathbf{p}^\star).$$

Thus there exist subgradients $s_\theta \in \partial\widehat{g}_{\theta,t}(\mathbf{p}^\star)$ and a normal-cone vector $v \in N_{\mathbb{R}^d_+}(\mathbf{p}^\star)$ such that

$$0 = \mathbf{b}^{\text{safe}} + \sum_\theta w_\theta s_\theta + v.$$

By Lemma 11, each $s_\theta$ can be written as $s_\theta = -\widehat{h}^{\lambda_\theta}_{\theta,t}(\mathbf{p}^\star)$ for some tie-weight vector $\lambda_\theta$. Define $\widehat{H}^{\text{kkt}}_t(\mathbf{w}, \mathbf{p}^\star) := \sum_\theta w_\theta \widehat{h}^{\lambda_\theta}_{\theta,t}(\mathbf{p}^\star)$. Then the KKT condition becomes

$$0 = \mathbf{b}^{\text{safe}} - \widehat{H}^{\text{kkt}}_t(\mathbf{w}, \mathbf{p}^\star) + v, \quad \text{i.e.,} \quad v = \widehat{H}^{\text{kkt}}_t(\mathbf{w}, \mathbf{p}^\star) - \mathbf{b}^{\text{safe}}.$$

The normal cone $N_{\mathbb{R}^d_+}(\mathbf{p}^\star)$ satisfies: $v_i = 0$ if $p^\star_i > 0$ and $v_i \le 0$ if $p^\star_i = 0$. Hence if $p^\star_i > 0$ then $\widehat{H}^{\text{kkt}}_{t,i} = b^{\text{safe}}_i$, and if $p^\star_i = 0$ then $\widehat{H}^{\text{kkt}}_{t,i} \le b^{\text{safe}}_i$. This yields the componentwise feasibility. The complementarity identity follows because for every coordinate $i$, either $p^\star_i = 0$ or $b^{\text{safe}}_i - \widehat{H}^{\text{kkt}}_{t,i} = 0$, hence $\sum_i p^\star_i(b^{\text{safe}}_i - \widehat{H}^{\text{kkt}}_{t,i}) = 0$. $\square$

**Lemma 26** (Consumption Drift Bound). *On the good event $\mathcal{E}$ (defined in Section D with Lemma 19 holding for both strict and weak thresholds), for all $t$,*

$$H(\mathbf{w}_t, \mathbf{p}_t) \le \mathbf{b}^{\text{safe}} + \Big(\sum_\theta w_{t,\theta}\beta_{h,\theta}(t)\Big)\mathbf{1}_d,$$

*where $H(\mathbf{w}, \mathbf{p}) := \sum_\theta w_\theta h_\theta(\mathbf{p})$ is the population consumption under the strict threshold (which equals the weak threshold by Assumption 5).*

*Proof.* Fix $t$ and apply Lemma 25 with $\mathbf{w} = \mathbf{w}_t$ and $\mathbf{p}^\star = \mathbf{p}_t$ to obtain $\widehat{H}^{\text{kkt}}_t(\mathbf{w}_t, \mathbf{p}_t) \le \mathbf{b}^{\text{safe}}$ componentwise. For each $\theta$, the tie-weighted empirical consumption $\widehat{h}^{\lambda_\theta}_{\theta,t}(\mathbf{p}_t)$ satisfies

$$\widehat{h}^>_{\theta,t}(\mathbf{p}_t) \le \widehat{h}^{\lambda_\theta}_{\theta,t}(\mathbf{p}_t) \le \widehat{h}^{\ge}_{\theta,t}(\mathbf{p}_t) \quad \text{(componentwise)}$$

by Lemma 11. On $\mathcal{E}$, Lemma 19 implies both $\|\widehat{h}^>_{\theta,t}(\mathbf{p}_t) - h_\theta(\mathbf{p}_t)\|_\infty \le \beta_{h,\theta}(t)$ and $\|\widehat{h}^{\ge}_{\theta,t}(\mathbf{p}_t) - h_\theta(\mathbf{p}_t)\|_\infty \le \beta_{h,\theta}(t)$. Since $\widehat{h}^{\lambda_\theta}_{\theta,t}(\mathbf{p}_t)$ lies between the strict and weak empirical quantities coordinatewise and $h_\theta(\mathbf{p}_t)$ equals the strict/weak population expectation, it follows that $\|\widehat{h}^{\lambda_\theta}_{\theta,t}(\mathbf{p}_t) - h_\theta(\mathbf{p}_t)\|_\infty \le \beta_{h,\theta}(t)$ as well. Therefore,

$$\begin{aligned}
H(\mathbf{w}_t, \mathbf{p}_t) &= \sum_\theta w_{t,\theta} h_\theta(\mathbf{p}_t) \\
&\le \sum_\theta w_{t,\theta}\Big(\widehat{h}^{\lambda_\theta}_{\theta,t}(\mathbf{p}_t) + \beta_{h,\theta}(t)\mathbf{1}_d\Big) \\
&= \widehat{H}^{\text{kkt}}_t(\mathbf{w}_t, \mathbf{p}_t) + \Big(\sum_\theta w_{t,\theta}\beta_{h,\theta}(t)\Big)\mathbf{1}_d \\
&\le \mathbf{b}^{\text{safe}} + \Big(\sum_\theta w_{t,\theta}\beta_{h,\theta}(t)\Big)\mathbf{1}_d.
\end{aligned}$$

$\square$

**Lemma 27** (Price-Error Bound). *On the good event $\mathcal{E}$, for all $t \le T$,*

$$V(\mathbf{w}_t) - U(\mathbf{w}_t, \mathbf{p}_t) \le dP_{\max}\sum_\theta w_{t,\theta}\beta_{h,\theta}(t),$$

*where $U(\mathbf{w}, \mathbf{p}) := \sum_\theta w_\theta\, \mathbb{E}_\theta[r\mathbf{1}\{r > \langle\mathbf{p}, \mathbf{a}\rangle\}]$.*

*Proof.* By Lemma 24,
$$V(\mathbf{w}_t) - U(\mathbf{w}_t, \mathbf{p}_t) \leq \langle \mathbf{p}_t, \mathbf{b}^{\text{safe}} - H(\mathbf{w}_t, \mathbf{p}_t) \rangle.$$

Apply Lemma 25 (with $\mathbf{w} = \mathbf{w}_t$) to obtain a tie-weighted empirical consumption $\widehat{H}_t^{\text{kkt}}(\mathbf{w}_t, \mathbf{p}_t)$ satisfying $\langle \mathbf{p}_t, \mathbf{b}^{\text{safe}} - \widehat{H}_t^{\text{kkt}}(\mathbf{w}_t, \mathbf{p}_t) \rangle = 0$. Hence

$$\begin{aligned} \langle \mathbf{p}_t, \mathbf{b}^{\text{safe}} - H(\mathbf{w}_t, \mathbf{p}_t) \rangle &= \langle \mathbf{p}_t, \mathbf{b}^{\text{safe}} - \widehat{H}_t^{\text{kkt}}(\mathbf{w}_t, \mathbf{p}_t) \rangle \\ &\quad + \langle \mathbf{p}_t, \widehat{H}_t^{\text{kkt}}(\mathbf{w}_t, \mathbf{p}_t) - H(\mathbf{w}_t, \mathbf{p}_t) \rangle \\ &= \langle \mathbf{p}_t, \widehat{H}_t^{\text{kkt}}(\mathbf{w}_t, \mathbf{p}_t) - H(\mathbf{w}_t, \mathbf{p}_t) \rangle. \end{aligned}$$

By Hölder and $\|\mathbf{p}_t\|_1 \leq dP_{\max}$,

$$\langle \mathbf{p}_t, \widehat{H}_t^{\text{kkt}}(\mathbf{w}_t, \mathbf{p}_t) - H(\mathbf{w}_t, \mathbf{p}_t) \rangle \leq \|\mathbf{p}_t\|_1 \cdot \|\widehat{H}_t^{\text{kkt}}(\mathbf{w}_t, \mathbf{p}_t) - H(\mathbf{w}_t, \mathbf{p}_t)\|_\infty \leq dP_{\max} \cdot \sum_\theta w_{t,\theta} \beta_{h,\theta}(t),$$

where the last inequality uses the same sandwich argument as in Lemma 26. $\qquad\square$

### E.5. Step 4: Budget-Feasibility Loss

The real algorithm uses budget checks, producing reward $R_T \leq \tilde{R}_T$. We bound $\mathbb{E}[\tilde{R}_T - R_T]$ using a clean martingale-maximum approach.

**Setup.** Let $\tilde{x}_t := \mathbf{1}\{r_t > \langle \mathbf{p}_t, \mathbf{a}_t \rangle\}$ be the unconstrained threshold decision (strict inequality by Assumption 4), and let $\tilde{R}_T := \sum_{t=1}^T r_t \tilde{x}_t$. For each resource coordinate $i$, define $\tilde{C}_t^{(i)} := \sum_{s=1}^t a_s^{(i)} \tilde{x}_s$ and the violation time $\tau_i := \inf\{t \geq 1 : \tilde{C}_t^{(i)} > B_i\}$, with the convention $\inf \emptyset = \infty$. Let $\tau := \min_{i \in [d]} \tau_i$.

**Lemma 28** (Budget Loss Controlled by Violation Times). *Almost surely,*

$$\tilde{R}_T - R_T \leq R_{\max}(T - \tau + 1)_+ \leq R_{\max} \sum_{i=1}^d (T - \tau_i + 1)_+.$$

*Proof.* If no coordinate violates, then the budget check never rejects an unconstrained acceptance, so $\tilde{R}_T = R_T$. Otherwise, after the first violation time $\tau$, the real algorithm may lose at most $R_{\max}$ reward per remaining period, giving the first inequality. The second inequality uses $\max_i z_i \leq \sum_i z_i$ with $z_i = (T - \tau_i + 1)_+$. $\qquad\square$

**Martingale structure.** For coordinate $i$, let $Y_{t,i} := a_t^{(i)} \tilde{x}_t \in [0, A_{\max}]$. Since $\theta_t \sim \mathbf{w}_t$ and $(r_t, \mathbf{a}_t) \sim \mathcal{D}_{\theta_t}$:

$$\mathbb{E}[Y_{t,i} | \mathcal{F}_{t-1}] = \sum_\theta w_{t,\theta} \mathbb{E}_\theta \left[ a^{(i)} \cdot \mathbf{1}\{r > \langle \mathbf{p}_t, \mathbf{a} \rangle\} \right] = H^{(i)}(\mathbf{w}_t, \mathbf{p}_t).$$

Define the martingale $M_{t,i} := \sum_{s=1}^t (Y_{s,i} - \mathbb{E}[Y_{s,i} | \mathcal{F}_{s-1}])$ with bounded increments $|M_{t,i} - M_{t-1,i}| \leq A_{\max}$.

**Lemma 29** (Violation Time Bounded by Martingale Maximum and Drift). *Assume the drift bound holds:* $\mathbb{E}[Y_{t,i} | \mathcal{F}_{t-1}] \leq b_i^{\text{safe}} + \Delta_t$ *for all* $t$, *where* $\Delta_t \geq 0$ *is* $\mathcal{F}_{t-1}$-*measurable. Let* $\Delta_{1:T} := \sum_{t=1}^T \Delta_t$. *Then for the violation time* $\tau_i$,

$$(T - \tau_i + 1)_+ \leq 1 + \frac{\sup_{t \leq T}(M_{t,i})_+ + \Delta_{1:T}}{b_i}.$$

*Proof.* If $\tau_i = \infty$ the left side is 0 and the inequality is trivial. Otherwise, by definition $\tilde{C}_{\tau_i}^{(i)} > B_i = Tb_i$. By the martingale decomposition,

$$\tilde{C}_{\tau_i}^{(i)} = \sum_{s=1}^{\tau_i} \mathbb{E}[Y_{s,i} | \mathcal{F}_{s-1}] + M_{\tau_i,i} \leq \tau_i b_i^{\text{safe}} + \sum_{s=1}^{\tau_i} \Delta_s + M_{\tau_i,i} \leq \tau_i(1 - \varepsilon)b_i + \Delta_{1:T} + M_{\tau_i,i}.$$

Thus

$$M_{\tau_i,i} > Tb_i - \tau_i(1 - \varepsilon)b_i - \Delta_{1:T} = (T - \tau_i)b_i + \varepsilon \tau_i b_i - \Delta_{1:T} \geq (T - \tau_i)b_i - \Delta_{1:T}.$$

Rearranging gives

$$T - \tau_i < \frac{(M_{\tau_i,i})_+ + \Delta_{1:T}}{b_i} \leq \frac{\sup_{t \leq T}(M_{t,i})_+ + \Delta_{1:T}}{b_i}.$$

Adding 1 yields the claim. □

**Lemma 30** (Expected Maximum of Bounded-Increment Martingale). *Let $(M_t)_{t=0}^T$ be a martingale with $M_0 = 0$ and increments bounded as $|M_t - M_{t-1}| \leq A_{\max}$ almost surely. Then*

$$\mathbb{E}\left[\sup_{t \leq T}(M_t)_+\right] \leq A_{\max}\sqrt{\frac{\pi T}{2}}.$$

*Proof.* By Azuma–Hoeffding,

$$\Pr\left(\sup_{t \leq T} M_t \geq u\right) \leq \exp\left(-\frac{u^2}{2TA_{\max}^2}\right) \quad \text{for all } u \geq 0.$$

(One may prove this by applying Azuma to the stopped process at the first hitting time of level $u$.)

Integrate the tail bound:

$$\mathbb{E}\left[\sup_{t \leq T}(M_t)_+\right] = \int_0^\infty \Pr\left(\sup_{t \leq T} M_t \geq u\right) du \leq \int_0^\infty \exp\left(-\frac{u^2}{2TA_{\max}^2}\right) du = A_{\max}\sqrt{\frac{\pi T}{2}}.$$

□

**Corollary 31** (Expected Feasibility Loss Bound). *Under the drift condition in Lemma 29,*

$$\mathbb{E}[\tilde{R}_T - R_T] \leq R_{\max}\sum_{i=1}^d \mathbb{E}[(T - \tau_i + 1)_+] \leq R_{\max}\left(d + \frac{d}{b_{\min}}\mathbb{E}[\Delta_{1:T}] + \frac{dA_{\max}}{b_{\min}}\sqrt{\frac{\pi T}{2}}\right).$$

*Proof.* Combine Lemma 28 and Lemma 29, take expectations, apply Lemma 30, and use $b_i \geq b_{\min}$. □

**Drift bound verification.** By Lemma 26, on the good event $\mathcal{E}$:

$$H(\mathbf{w}_t, \mathbf{p}_t) \leq \mathbf{b}^{\text{safe}} + \sum_\theta w_{t,\theta}\beta_{h,\theta}(t) \cdot \mathbf{1}_d,$$

so $\mathbb{E}[Y_{t,i}|\mathcal{F}_{t-1}] \leq b_i^{\text{safe}} + \Delta_t$ with $\Delta_t := \sum_\theta w_{t,\theta}\beta_{h,\theta}(t)$. By Corollary 31:

$$\mathbb{E}[\tilde{R}_T - R_T|\mathcal{E}] \leq R_{\max}\left(d + \frac{d}{b_{\min}}\mathbb{E}[\Delta_{1:T}|\mathcal{E}] + \frac{dA_{\max}}{b_{\min}}\sqrt{\frac{\pi T}{2}}\right).$$

### E.6. Step 5: Summing Confidence Radii and Slack Loss

**Lemma 32** (Summation of $1/\sqrt{N}$). *Let $\theta_t$ be the arm pulled at time $t$. Then pathwise:*

$$\sum_{t=1}^T \frac{1}{\sqrt{N_{\theta_t}(t) \vee 1}} \leq 2\sum_{\theta=1}^K \sqrt{N_\theta(T) \vee 1} \leq 2\sqrt{K(T + K)}.$$

*Proof.* For each $\theta$, the $j$-th pull contributes $1/\sqrt{j}$; $\sum_{j=1}^n 1/\sqrt{j} \leq 1 + \int_1^n s^{-1/2}ds \leq 2\sqrt{n}$. Then apply Cauchy–Schwarz: $\sum_\theta \sqrt{N_\theta} \leq \sqrt{K\sum_\theta N_\theta} = \sqrt{KT}$. □

Since $\beta_g(n), \beta_h(n) = O(\sqrt{d \log T/n})$, Lemma 32 implies:

$$\sum_{t=1}^{T} \sum_{\theta} w_{t,\theta} \beta_{g,\theta}(t) = O\left(\alpha \, c_g R_{\max} \sqrt{KT \cdot d \log T}\right),$$

and similarly:

$$\mathbb{E}[\Delta_{1:T}] = O\left(A_{\max} \sqrt{KT \cdot d \log T}\right).$$

**Lemma 33** (Lipschitzness and Monotonicity in the Budget). *For any* $\mathbf{b}, \mathbf{b}' \in \mathbb{R}_+^d$,

$$\left| V^{\mathrm{mix}}(\mathbf{b}) - V^{\mathrm{mix}}(\mathbf{b}') \right| \leq P_{\max} \|\mathbf{b} - \mathbf{b}'\|_1.$$

*Moreover,* $V^{\mathrm{mix}}$ *is monotone: if* $\mathbf{b} \geq \mathbf{b}'$ *componentwise, then* $V^{\mathrm{mix}}(\mathbf{b}) \geq V^{\mathrm{mix}}(\mathbf{b}')$. *In particular, with* $\mathbf{b}^{\mathrm{safe}} = (1-\varepsilon)\mathbf{b} \leq \mathbf{b}$,

$$0 \leq T\left(V^{\mathrm{mix}}(\mathbf{b}) - V^{\mathrm{mix}}(\mathbf{b}^{\mathrm{safe}})\right) \leq P_{\max} \|\mathbf{b} - \mathbf{b}^{\mathrm{safe}}\|_1 T = P_{\max} \|\mathbf{b}\|_1 \, \varepsilon T.$$

*Proof.* Using the dual form,
$$V^{\mathrm{mix}}(\mathbf{b}) = \min_{\mathbf{p} \in \mathcal{P}} \{\langle \mathbf{p}, \mathbf{b} \rangle + \max_\theta g_\theta(\mathbf{p})\}.$$

For any fixed $\mathbf{p} \in \mathcal{P}$,
$$\langle \mathbf{p}, \mathbf{b} \rangle + \max_\theta g_\theta(\mathbf{p}) - \left(\langle \mathbf{p}, \mathbf{b}' \rangle + \max_\theta g_\theta(\mathbf{p})\right) = \langle \mathbf{p}, \mathbf{b} - \mathbf{b}' \rangle,$$

hence $V^{\mathrm{mix}}(\mathbf{b}) - V^{\mathrm{mix}}(\mathbf{b}') \leq \sup_{\mathbf{p} \in \mathcal{P}} \langle \mathbf{p}, \mathbf{b} - \mathbf{b}' \rangle \leq P_{\max} \|\mathbf{b} - \mathbf{b}'\|_1$. Swap $\mathbf{b}, \mathbf{b}'$ to get the absolute-value bound. Monotonicity follows since $\langle \mathbf{p}, \mathbf{b} \rangle$ is monotone in $\mathbf{b}$ and the minimum over $\mathbf{p}$ preserves monotonicity. $\square$

### E.7. Combining All Pieces (Unconditional Expectation; No Conditioning on $\mathcal{E}$)

We now combine the previous steps in a way that is fully rigorous: we *never* condition on the global good event $\mathcal{E}$ (which depends on the entire trajectory), and instead split expectations using indicators $1_{\mathcal{E}}$ and $1_{\mathcal{E}^c}$.

**Warm-Start Definition.** For warm-start rounds $t \leq K$, the algorithm sets $x_t = 0$ (pure observation). We define $\tilde{x}_t := 0$ and $U(\mathbf{w}_t, \mathbf{p}_t) := 0$ for $t \leq K$, as saddle-point outputs $(\mathbf{w}_t, \mathbf{p}_t)$ are only computed for $t \geq K+1$. Accordingly, all sums involving $U(\mathbf{w}_t, \mathbf{p}_t)$ or $\tilde{x}_t$ below are taken over $t = K+1, \ldots, T$.

**Warm-Start Loss.** During rounds $t \leq K$, the algorithm forces $x_t = 0$, hence it can lose at most $R_{\max}$ reward per round relative to any benchmark. Therefore the warm start contributes at most $KR_{\max}$ to regret.

**Step A: Relate $\mathbb{E}[\tilde{R}_T]$ to $U(\mathbf{w}_t, \mathbf{p}_t)$.** For $t > K$, recall the unconstrained strict-threshold decision $\tilde{x}_t := \mathbf{1}\{r_t > \langle \mathbf{p}_t, \mathbf{a}_t \rangle\}$; for $t \leq K$, we set $\tilde{x}_t := 0$ by convention. Define $\tilde{R}_T := \sum_{t=K+1}^{T} r_t \tilde{x}_t$. Because $\mathbf{w}_t$ and $\mathbf{p}_t$ are $\mathcal{F}_{t-1}$-measurable and $\theta_t \sim \mathbf{w}_t$, for $t > K$ we have

$$\mathbb{E}[r_t \tilde{x}_t \mid \mathcal{F}_{t-1}] = \sum_\theta w_{t,\theta} \, \mathbb{E}_\theta[r \, \mathbf{1}\{r > \langle \mathbf{p}_t, \mathbf{a} \rangle\}] = U(\mathbf{w}_t, \mathbf{p}_t).$$

Taking total expectation and summing over $t = K+1, \ldots, T$ (tower property) gives

$$\mathbb{E}[\tilde{R}_T] = \sum_{t=K+1}^{T} \mathbb{E}[U(\mathbf{w}_t, \mathbf{p}_t)].$$

Hence

$$TV^{\mathrm{mix}}(\mathbf{b}^{\mathrm{safe}}) - \mathbb{E}[\tilde{R}_T] = KV^{\mathrm{mix}}(\mathbf{b}^{\mathrm{safe}}) + \sum_{t=K+1}^{T} \mathbb{E}\left[V^{\mathrm{mix}}(\mathbf{b}^{\mathrm{safe}}) - U(\mathbf{w}_t, \mathbf{p}_t)\right].$$

**Step B: Per-Round Expected Gap on $\mathcal{E}$.** On the good event $\mathcal{E}$, Lemmas 23 and 27 imply for every round $t > K$:

$$V^{\text{mix}}(\mathbf{b}^{\text{safe}}) - U(\mathbf{w}_t, \mathbf{p}_t) = \big(V^{\text{mix}}(\mathbf{b}^{\text{safe}}) - V(\mathbf{w}_t)\big) + \big(V(\mathbf{w}_t) - U(\mathbf{w}_t, \mathbf{p}_t)\big) \leq 2 \sum_\theta w_{t,\theta} \beta_{g,\theta}(t) + dP_{\max} \sum_\theta w_{t,\theta} \beta_{h,\theta}(t).$$

Multiplying by $1_{\mathcal{E}}$ and using nonnegativity of the RHS yields the unconditional bound

$$\mathbb{E}\big[(V^{\text{mix}}(\mathbf{b}^{\text{safe}}) - U(\mathbf{w}_t, \mathbf{p}_t))1_{\mathcal{E}}\big] \leq 2 \, \mathbb{E}\left[\sum_\theta w_{t,\theta} \beta_{g,\theta}(t)\right] + dP_{\max} \, \mathbb{E}\left[\sum_\theta w_{t,\theta} \beta_{h,\theta}(t)\right].$$

On $\mathcal{E}^c$, we use the crude bound $0 \leq U(\mathbf{w}_t, \mathbf{p}_t) \leq R_{\max}$ and $0 \leq V^{\text{mix}}(\mathbf{b}^{\text{safe}}) \leq R_{\max}$, hence

$$\mathbb{E}\big[(V^{\text{mix}}(\mathbf{b}^{\text{safe}}) - U(\mathbf{w}_t, \mathbf{p}_t))1_{\mathcal{E}^c}\big] \leq R_{\max} \Pr(\mathcal{E}^c).$$

Summing over $t$ and adding the warm start loss gives

$$TV^{\text{mix}}(\mathbf{b}^{\text{safe}}) - \mathbb{E}[\tilde{R}_T] \leq K R_{\max} + 2 \sum_{t=K+1}^{T} \mathbb{E}[\sum_\theta w_{t,\theta} \beta_{g,\theta}(t)] + dP_{\max} \sum_{t=K+1}^{T} \mathbb{E}[\sum_\theta w_{t,\theta} \beta_{h,\theta}(t)] + T R_{\max} \Pr(\mathcal{E}^c).$$

**Step C: Feasibility Loss $\mathbb{E}[\tilde{R}_T - R_T]$.** We split

$$\mathbb{E}[\tilde{R}_T - R_T] = \mathbb{E}[(\tilde{R}_T - R_T)1_{\mathcal{E}}] + \mathbb{E}[(\tilde{R}_T - R_T)1_{\mathcal{E}^c}].$$

The bad-event term satisfies $\mathbb{E}[(\tilde{R}_T - R_T)1_{\mathcal{E}^c}] \leq T R_{\max} \Pr(\mathcal{E}^c)$.

On $\mathcal{E}$, the drift condition in Lemma 29 holds with $\Delta_t := \sum_\theta w_{t,\theta} \beta_{h,\theta}(t)$ by Lemma 26, so Lemmas 28–29 imply (for each coordinate $i$)

$$(T - \tau_i + 1)_+ 1_{\mathcal{E}} \leq \left(1 + \frac{\sup_{t \leq T}(M_{t,i})_+ + \Delta_{1:T}}{b_i}\right) 1_{\mathcal{E}}.$$

Taking expectations and using $\mathbb{E}[\sup_{t \leq T}(M_{t,i})_+ 1_{\mathcal{E}}] \leq \mathbb{E}[\sup_{t \leq T}(M_{t,i})_+]$ together with Lemma 30 gives

$$\mathbb{E}[(T - \tau_i + 1)_+ 1_{\mathcal{E}}] \leq 1 + \frac{A_{\max}\sqrt{\pi T/2}}{b_i} + \frac{\mathbb{E}[\Delta_{1:T}]}{b_i}.$$

Combining with Lemma 28 and $b_i \geq b_{\min}$ yields

$$\mathbb{E}[(\tilde{R}_T - R_T)1_{\mathcal{E}}] \leq R_{\max}\left(d + \frac{dA_{\max}}{b_{\min}}\sqrt{\frac{\pi T}{2}} + \frac{d}{b_{\min}}\mathbb{E}[\Delta_{1:T}]\right).$$

Therefore,

$$\mathbb{E}[\tilde{R}_T - R_T] \leq R_{\max}\left(d + \frac{dA_{\max}}{b_{\min}}\sqrt{\frac{\pi T}{2}} + \frac{d}{b_{\min}}\mathbb{E}[\Delta_{1:T}]\right) + T R_{\max} \Pr(\mathcal{E}^c).$$

**Step D: Slack Loss.** By Lemma 33,

$$0 \leq T\big(V^{\text{mix}}(\mathbf{b}) - V^{\text{mix}}(\mathbf{b}^{\text{safe}})\big) \leq P_{\max}\|\mathbf{b} - \mathbf{b}^{\text{safe}}\|_1 T = P_{\max}\|\mathbf{b}\|_1 \sqrt{T \log T}.$$

**Step E: Bound the Confidence Sums.** Using Lemma 21 and Lemma 32, and the definitions of $\beta_{g,\theta}(t)$ and $\beta_{h,\theta}(t)$ (both scaling as $O(1/\sqrt{N_\theta(t)})$), we obtain

$$\sum_{t=K+1}^{T} \mathbb{E}\left[\sum_\theta w_{t,\theta} \beta_{g,\theta}(t)\right] = \tilde{O}\big(\sqrt{KT \cdot d}\big), \qquad \sum_{t=K+1}^{T} \mathbb{E}\left[\sum_\theta w_{t,\theta} \beta_{h,\theta}(t)\right] = \tilde{O}\big(\sqrt{KT \cdot d}\big),$$

and similarly $\mathbb{E}[\Delta_{1:T}] = \tilde{O}(\sqrt{KT \cdot d})$.

**Step F: Bad Event Probability.** By construction of $\mathcal{E}$ we have $\Pr(\mathcal{E}^c) \leq \delta_{\text{tot}} = T^{-2}$, hence the total bad-event contribution is at most $O(R_{\max}/T)$.

**Conclusion.** Combining Steps B–F and recalling $R_T \leq \tilde{R}_T$ yields

$$\text{Reg}^{\text{mix}}(T) = TV^{\text{mix}}(\mathbf{b}) - \mathbb{E}[R_T] \leq C_1 \, \alpha \, \sqrt{KT \cdot d \log T} + C_2 \sqrt{T \log T} + KR_{\max},$$

for constants $C_1, C_2$ depending only on $(d, R_{\max}, A_{\max}, P_{\max}, b_{\min}, \|\mathbf{b}\|_1)$. This completes the proof of Theorem 5. $\square$

## F. Implementation Notes

### F.1. Compute Saddle Point via Linear Programming

Let $\bar{\mathbf{b}}$ denote the per-period budget vector used in the saddle problem; in Algorithm 1, $\bar{\mathbf{b}} = \mathbf{b}^{\text{safe}}$. Define

$$L(\mathbf{w}, \mathbf{p}) = \bar{\mathbf{b}}^T \mathbf{p} + \sum_{\theta \in \Theta} w_\theta \left( \frac{1}{N_\theta} \sum_{j=1}^{N_\theta} (r_{j,\theta} - \mathbf{a}_{j,\theta}^T \mathbf{p})^+ + \beta_\theta \right)$$

By Assumption 2 and the argument in Lemma 6, we have

$$\max_{\mathbf{w} \in \Delta_K} \min_{\mathbf{p} \geq 0} L(\mathbf{w}, \mathbf{p}) = \max_{\mathbf{w} \in \Delta_K} \min_{\mathbf{p} \in \mathcal{P}} L(\mathbf{w}, \mathbf{p}) = \min_{\mathbf{p} \in \mathcal{P}} \max_{\mathbf{w} \in \Delta_K} L(\mathbf{w}, \mathbf{p}) = \min_{\mathbf{p} \geq 0} \max_{\mathbf{w} \in \Delta_K} L(\mathbf{w}, \mathbf{p}).$$

Thus the empirical saddle point is obtained by solving the following LP:

$$
\begin{aligned}
Min \quad & \bar{\mathbf{b}}^T \mathbf{p} + z \\
S.T \quad & z \geq \frac{1}{N_\theta} \sum_{j=1}^{N_\theta} y_{j,\theta} + \beta_\theta \quad \forall \theta \\
& y_{j,\theta} \geq r_{j,\theta} - \mathbf{a}_{j,\theta}^T \mathbf{p} \quad \forall \theta, j \\
& \mathbf{p} \geq 0, \ y_{j,\theta} \geq 0 \quad \forall \theta, j.
\end{aligned}
$$

Let $\mathbf{p}^*$ be the optimal solution of this LP and let $\mathbf{w}^* = \{w_\theta^*\}_{\theta \in \Theta}$ be an optimal dual solution corresponding to the constraints $z \geq \frac{1}{N_\theta} \sum_{j=1}^{N_\theta} y_{j,\theta} + \beta_\theta \ \forall \theta$. Then $(\mathbf{w}^*, \mathbf{p}^*)$ is a saddle point.

*Proof.* By dual feasibility, we have $w_\theta^* \geq 0$ and $\sum_{\theta \in \Theta} w_\theta^* = 1$. Thus, $\mathbf{w}^* \in \Delta_K$. Let $z^*$ be the optimal solution of the LP. For any $\mathbf{w} \in \Delta_K$,

$$L(\mathbf{w}, \mathbf{p}^*) \leq \bar{\mathbf{b}}^T \mathbf{p}^* + \max_{\theta \in \Theta} \frac{1}{N_\theta} \sum_{j=1}^{N_\theta} (r_{j,\theta} - \mathbf{a}_{j,\theta}^T \mathbf{p}^*)^+ + \beta_\theta = \bar{\mathbf{b}}^T \mathbf{p}^* + z^*$$

By complementary slackness, $w_\theta^*$ is positive only if

$$\frac{1}{N_\theta} \sum_{j=1}^{N_\theta} (r_{j,\theta} - \mathbf{a}_{j,\theta}^T \mathbf{p}^*)^+ + \beta_\theta = z^*$$

Thus,

$$\sum_{\theta \in \Theta} w_\theta^* \left( \frac{1}{N_\theta} \sum_{j=1}^{N_\theta} (r_{j,\theta} - \mathbf{a}_{j,\theta}^T \mathbf{p}^*)^+ + \beta_\theta \right) = z^*$$

Thus,

$$L(\mathbf{w}, \mathbf{p}^*) \leq L(\mathbf{w}^*, \mathbf{p}^*) \quad \forall \mathbf{w} \in \Delta_K \tag{4}$$

For any $\mathbf{p} \geq 0$,

$$L(\mathbf{w}^*, \mathbf{p}) = \bar{\mathbf{b}}^T \mathbf{p} + \sum_{\theta \in \Theta} w_\theta^* \beta_\theta + \sum_{\theta \in \Theta} \sum_{j=1}^{N_\theta} \max_{0 \leq \eta_{\theta,j} \leq \frac{w_\theta^*}{N_\theta}} \eta_{\theta,j} (r_{j,\theta} - \mathbf{a}_{j,\theta}^T \mathbf{p})$$

Let $\eta_{\theta,j}^* = \frac{w_\theta^*}{N_\theta} \mathbf{1}\{r_{j,\theta} > \mathbf{a}_{j,\theta}^T \mathbf{p}^*\}$. Then,

$$L(\mathbf{w}^*, \mathbf{p}^*) = \bar{\mathbf{b}}^T \mathbf{p}^* + \sum_{\theta \in \Theta} w_\theta^* \beta_\theta + \sum_{\theta \in \Theta} \sum_{j=1}^{N_\theta} \eta_{\theta,j}^* (r_{j,\theta} - \mathbf{a}_{j,\theta}^T \mathbf{p}^*)$$

In addition, for any $\mathbf{p} \geq 0$,

$$L(\mathbf{w}^*, \mathbf{p}) \geq \bar{\mathbf{b}}^T \mathbf{p} + \sum_{\theta \in \Theta} w_\theta^* \beta_\theta + \sum_{\theta \in \Theta} \sum_{j=1}^{N_\theta} \eta_{\theta,j}^* (r_{j,\theta} - \mathbf{a}_{j,\theta}^T \mathbf{p})$$

$$= \sum_{\theta \in \Theta} w_\theta^* \beta_\theta + \sum_{\theta \in \Theta} \sum_{j=1}^{N_\theta} \eta_{\theta,j}^* r_{j,\theta} + \mathbf{p}^T (\bar{\mathbf{b}} - \sum_{\theta \in \Theta} \sum_{j=1}^{N_\theta} \eta_{\theta,j}^* \mathbf{a}_{j,\theta})$$

By dual feasibility and complementary slackness,

$$\bar{\mathbf{b}} - \sum_{\theta \in \Theta} \sum_{j=1}^{N_\theta} \eta_{\theta,j}^* \mathbf{a}_{j,\theta} \geq 0$$

$$\mathbf{p}^{*T} (\bar{\mathbf{b}} - \sum_{\theta \in \Theta} \sum_{j=1}^{N_\theta} \eta_{\theta,j}^* \mathbf{a}_{j,\theta}) = 0$$

Thus,

$$L(\mathbf{w}^*, \mathbf{p}^*) = \sum_{\theta \in \Theta} w_\theta^* \beta_\theta + \sum_{\theta \in \Theta} \sum_{j=1}^{N_\theta} \eta_{\theta,j}^* r_{j,\theta}$$

$$\mathbf{p}^T (\bar{\mathbf{b}} - \sum_{\theta \in \Theta} \sum_{j=1}^{N_\theta} \eta_{\theta,j}^* \mathbf{a}_{j,\theta}) \geq 0 \quad \forall \mathbf{p} \geq 0$$

Thus,

$$L(\mathbf{w}^*, \mathbf{p}^*) \leq L(\mathbf{w}^*, \mathbf{p}) \quad \forall \mathbf{p} \geq 0 \tag{5}$$

(4) and (5) complete the proof. $\qquad \square$

### F.2. Epoch/Doubling Schedule

For computational efficiency, one can update $(\mathbf{w}_t, \mathbf{p}_t)$ only when some $N_\theta$ doubles (i.e., at times $\tau \in \{1, 2, 4, 8, \ldots\}$ in $N_\theta$). This reduces the number of saddle problem solves from $T$ to $O(K \log T)$ while preserving the regret bound up to constants.

**Rationale**: The confidence radius $\beta_\theta(t) \propto 1/\sqrt{N_\theta}$ changes by at most a factor of $\sqrt{2}$ between doubling updates, so the regret analysis remains valid with adjusted constants.

## G. Experimental Details

This section provides complete specifications for reproducibility.

### G.1. Scenario Specifications

**S4: Complementarity.** $K = 2$ configurations with $d = 2$ resources, designed as a near-deterministic diagnostic. Both configurations have reward $r = 1 + \xi$ with $\xi \sim \text{Uniform}(-0.01, 0.01)$ (small noise satisfies the no-ties assumption). Consumption profiles are orthogonal:

| $\theta$ | Reward | Consumption $\mathbf{a}$ | Resource usage |
|---|---|---|---|
| 0 | $1.0 \pm 0.01$ | $[1.0, 0.0] \pm 0.01$ | Resource 1 only |
| 1 | $1.0 \pm 0.01$ | $[0.0, 1.0] \pm 0.01$ | Resource 2 only |

**Budget**: $\mathbf{b}_0 = [0.5, 0.5]$ (baseline per-period budget), so $\mathbf{b} = \rho \cdot [0.5, 0.5]$. With $\rho = 0.7$, $\mathbf{b} = [0.35, 0.35]$.

**Oracle values**: $V^*(\mathbf{b}) = 0.5$ (any fixed configuration wastes one resource), $V^{\mathrm{mix}}(\mathbf{b}) = 1.0$ (alternating uses both), yielding gap $V^{\mathrm{mix}}/V^* = 2.0$.

**Experiment**: $T = 5{,}000$, 10 seeds, $\alpha = 0.1$.

**S0: Theory-Compliant Regret Validation.** $K = 5$ configurations with $d = 3$ resources and truncated Gaussian arrivals. For each configuration $\theta$, rewards $r \sim \mathcal{N}(\mu_r^\theta, (\sigma_r^\theta)^2)$ are truncated to $[0.01, R_{\mathrm{max}}]$ and per-resource consumptions $a_j \sim \mathcal{N}(\mu_{a,j}^\theta, (\sigma_{a,j}^\theta)^2)$ are truncated to $[0.01, A_{\mathrm{max}}]$, with $R_{\mathrm{max}} = A_{\mathrm{max}} = 2.0$. This scenario is used for validating $\sqrt{T}$ regret scaling with the theory-compliant exploration parameter $\alpha = 1.5$ (Section 6.1).

| $\theta$ | $\mu_r$ | $\sigma_r$ | $\mu_a$ | $\sigma_a$ |
|---|---|---|---|---|
| 0 | 1.0 | 0.3 | [0.8, 0.2, 0.2] | [0.2, 0.2, 0.2] |
| 1 | 0.8 | 0.3 | [0.2, 0.7, 0.2] | [0.2, 0.2, 0.2] |
| 2 | 0.6 | 0.3 | [0.2, 0.2, 0.6] | [0.2, 0.2, 0.2] |
| 3 | 0.9 | 0.3 | [0.5, 0.5, 0.5] | [0.2, 0.2, 0.2] |
| 4 | 0.4 | 0.3 | [0.1, 0.1, 0.1] | [0.2, 0.2, 0.2] |

**Budget**: The baseline per-period budget $\mathbf{b}_0$ is derived from the mean consumption of the most resource-intensive configuration (config 0: $\bar{\mathbf{a}}_0 = [0.8, 0.2, 0.2]$); with $\rho = 0.7$, $\mathbf{b} = 0.7 \cdot \mathbf{b}_0$.

**Algorithm parameters**: $\alpha = 1.5$, $R_{\mathrm{max}} = A_{\mathrm{max}} = 2.0$, $P_{\mathrm{max}} = 2.0$, $\delta = T^{-2}$, $\varepsilon = \sqrt{\log T/T}$, $c_g = 0.0707$, warm-start rounds $= K$, saddle-point solve on doubling schedule, $\rho = 0.7$, $T \in \{100, 200, 500, 1000, 2000\}$, 50 seeds.

**Boundedness Enforcement.** In all synthetic experiments, we enforce the boundedness assumptions (Assumption 2 in the main paper) by clipping rewards to $[0, R_{\mathrm{max}}]$ and per-coordinate consumptions to $[0, A_{\mathrm{max}}]$. To reduce empirical ties at threshold boundaries, we add continuous jitter $\xi \sim \mathrm{Unif}(-\eta, \eta)$ with $\eta = 10^{-6}$ to rewards. Note that hard clipping can create atoms at boundary values; in our experiments, boundary events are rare (clipping occurs in $< 0.1\%$ of samples), so empirical tie frequency remains negligible. For stricter compliance with Assumption 5, one could use truncation (rejection sampling) instead of clipping.

### G.2. Algorithm Parameters

| Parameter | Value |
|---|---|
| $R_{\mathrm{max}}$ | 10.0 |
| $A_{\mathrm{max}}$ | 2.0 |
| $P_{\mathrm{max}}$ | $2R_{\mathrm{max}}/b_{\mathrm{min}}$[1] |
| $\delta$ | $1/T^2$ |
| Slack $\varepsilon$ | $\sqrt{\log T/T}$ |
| Warm start | First $K$ rounds (round-robin, no budget consumption) |
| Saddle solve frequency | Doubling schedule (when any $N_\theta$ doubles) |

### G.3. Confidence Radius

The exploration parameter $\alpha$ scales the confidence radius:

$$\beta_\theta(t) = \alpha \cdot c_g R_{\mathrm{max}} \cdot \sqrt{\frac{d \log t + \log(KT/\delta)}{N_\theta(t)}},$$

where $c_g > 0$ is an absolute constant from the concentration lemma (Lemma 15). The factor $R_{\mathrm{max}}$ (not $R_{\mathrm{max}} + P_{\mathrm{max}} A_{\mathrm{max}}$) arises because $(r - \langle \mathbf{p}, \mathbf{a} \rangle)_+ \in [0, R_{\mathrm{max}}]$.

Setting $\alpha = 0$ yields pure exploitation (greedy); $\alpha = 0.1$ is the standard UCB scaling. *Note:* Theoretical guarantees (Theorem 5 in the main paper) hold for $\alpha \geq 1$; values $\alpha < 1$ are heuristics that often improve empirical performance but are not covered by the high-probability analysis.

## G.4. E1: Benchmark Validation (Complementarity Gap)

A core claim is that the switching-aware benchmark $V^{\text{mix}}$ properly upper-bounds all policies while the fixed-configuration benchmark $V^*$ can be beaten. We test this on S4 (Complementarity).

*Table 4.* E1: Benchmark Validation on S4 Complementarity (10 seeds each). $CR^* > 1$ confirms the fixed oracle is beatable; $CR^{\text{mix}} \leq 1$ confirms $V^{\text{mix}}$ is a valid upper bound.

| Algorithm | $\rho$ | $CR^{\text{mix}}$ | $CR^*$ | Gap |
|---|---|---|---|---|
| Greedy | 0.3 | $0.956 \pm 0.011$ | $1.908 \pm 0.022$ | 1.996 |
| Greedy | 0.5 | $0.956 \pm 0.014$ | $1.904 \pm 0.027$ | 1.992 |
| Greedy | 0.7 | $0.956 \pm 0.012$ | $1.900 \pm 0.023$ | 1.987 |
| Greedy | 0.9 | $0.960 \pm 0.009$ | $1.902 \pm 0.018$ | 1.982 |
| Greedy | 1.2 | $1.000 \pm 0.000$ | $1.652 \pm 0.000$ | 1.652 |
| SP-UCB-OLP | 0.3 | $0.891 \pm 0.054$ | $1.778 \pm 0.109$ | 1.996 |
| SP-UCB-OLP | 0.5 | $0.853 \pm 0.045$ | $1.700 \pm 0.090$ | 1.992 |
| SP-UCB-OLP | 0.7 | $0.909 \pm 0.022$ | $1.807 \pm 0.043$ | 1.987 |
| SP-UCB-OLP | 0.9 | $0.950 \pm 0.029$ | $1.883 \pm 0.057$ | 1.982 |
| SP-UCB-OLP | 1.2 | $0.993 \pm 0.005$ | $1.640 \pm 0.009$ | 1.652 |
| Random | 0.7 | $0.989 \pm 0.000$ | $1.965 \pm 0.000$ | 1.987 |
| OneHot | 0.7 | $0.507 \pm 0.012$ | $1.007 \pm 0.022$ | 1.988 |

## G.5. Alibaba Trace Experiments

We validate SP-UCB-OLP on real-world cluster traces from Alibaba (Alibaba Group, 2018). The trace is processed in *original temporal order*, preserving realistic non-stationary arrival patterns.

**Data Source.** Alibaba Cluster Trace v2018 (`batch_task.csv`), containing 13.4 million task records over 8.9 days of cluster operations.

**Experimental Configuration.**

| Parameter | Value |
|---|---|
| $T$ (time horizon) | 5,000 |
| $K$ (regimes) | 3 |
| $d$ (resources) | 2 (CPU, Memory) |
| Seeds | 42–91 (50 seeds) |
| $\rho$ (budget scaling) | 1.0 |
| Noise $\sigma$ | 0.1 |

**Resource Consumption.** Real values from trace, normalized to $[0, 1]$:

- CPU: $\text{cpu} = \text{plan\_cpu}/100$, mean $= 0.836$, std $= 0.635$

- Memory: $\text{mem} = \text{plan\_mem}/100$, mean $= 0.349$, std $= 0.299$

The consumption vector is $\mathbf{a} = [\text{cpu}, \text{mem}]$.

**Reward Construction.**

$$r[\theta] = c_1[\theta] \cdot \text{cpu} + c_2[\theta] \cdot \text{mem} + \epsilon, \quad \epsilon \sim \mathcal{N}(0, 0.01)$$

This formula creates a stationary LP structure where the optimal regime depends on resource availability. Given the trace's CPU-heavy profile (mean CPU > mean memory), Regime 0 (CPU-heavy) tends to dominate.

**Regime Specifications.**

| Regime | Name | $c_1$ | $c_2$ | Description |
|---|---|---|---|---|
| 0 | CPU-heavy | 2.0 | 0.5 | High reward per CPU unit |
| 1 | Memory-heavy | 0.5 | 2.0 | High reward per memory unit |
| 2 | Balanced | 1.2 | 1.2 | Equal reward per resource |

**Budget Computation.** Nominal budget: $\mathbf{b}_0 = 0.5 \cdot T \cdot \bar{\mathbf{a}}$, where $\bar{\mathbf{a}}$ is mean consumption. Scaled budget: $\mathbf{b} = \rho \cdot \mathbf{b}_0$.

**Oracle Estimation.** $V^{\mathrm{mix}}$ estimated via Monte Carlo with 10,000 samples per regime, solving the dual LP (Section F).

**Complete Results.**

| Algorithm | $\alpha$ | Mean CR | Std CR | Range |
|---|---|---|---|---|
| Oracle | – | 99.95% | 0.02% | [99.88%, 100.00%] |
| SP-UCB-OLP | 0.01 | **97.38%** | **0.59%** | [94.72%, 98.17%] |
| SP-UCB-OLP | 0.10 | 95.21% | 0.52% | [93.96%, 96.03%] |
| SP-UCB-OLP | 1.00 | 91.98% | 0.57% | [90.52%, 92.99%] |
| Greedy | 0.00 | 84.81% | 14.24% | [60.88%, 98.03%] |
| Random | – | 61.05% | 0.25% | [60.33%, 61.55%] |

**Key Findings.**

1. **Exploration prevents lock-in**: Greedy ($\alpha = 0$) has 14.2% std with 10/50 seeds stuck below 70% CR; $\alpha = 0.01$ reduces std to 0.6%.

2. **Minimal exploration suffices**: $\alpha = 0.01$ achieves 97.4% CR, only 2.6% below Oracle.

3. **Diminishing returns**: Increasing $\alpha$ beyond 0.01 hurts performance (95.2% at $\alpha = 0.1$, 92.0% at $\alpha = 1.0$).

4. **Oracle validates benchmark**: 99.95% CR confirms Monte Carlo estimation accuracy.

5. **Random baseline**: 61.1% CR provides lower bound for uninformed policy.

