# OpenReview forum: "A Two-Layer Framework for Joint Online Configuration Selection and Admission Control"
_ICML.cc/2026/Conference — ICML 2026 regular_

### Official Review · Reviewer_DCoV · 2026-03-03

**Soundness:** 4
**Presentation:** 3
**Significance:** 3
**Originality:** 3
**Overall Recommendation:** 4
**Confidence:** 4

**Summary:**

This paper studies an online decision problem with two layers. In each period, the decision maker first chooses one of K configurations. That configuration determines the distribution of the reward and resource vector of the incoming request. After observing the realized reward and resource, the decision maker decides whether to accept the request, subject to a cumulative resource budget over T periods.

The key feature is that reward and resource are observed before the admission decision. This makes the setting different from standard bandits with knapsacks, where budget is consumed immediately when an arm is pulled. At the same time, because the reward-resource distribution depends on the chosen configuration, it is not the same as classical online linear programming either.

To evaluate performance, the paper introduces a switching-aware fluid oracle. Instead of fixing one configuration, this oracle allows mixing configurations over time. The authors write the oracle as a max-min problem and characterize its saddle points using primal-dual conditions. The result is that the optimal oracle policy consists of a mixture over configurations together with a threshold rule for admission based on a dual price vector.

The proposed algorithm, called SP-UCB-OLP, maintains empirical surplus estimates for each configuration and builds confidence bounds. In each round it solves an optimistic saddle-point problem, samples a configuration from the resulting mixture, and then applies a threshold rule for admission. The regret is shown to scale on the order of the square root of K times T, up to logarithmic factors, measured against the switching-aware oracle.

**Compliance With Llm Reviewing Policy:**

Affirmed.

**Key Questions For Authors:**

The regret is measured against the switching-aware fluid oracle. Can you comment on how far this oracle can be from the true offline optimal policy that enforces the budget exactly?

What is the computational cost of solving the saddle-point problem each round? How does it scale in K and in the number of resource types?

How does the resource dimension enter the regret bound? Is there any dependence that could become large in high dimension?

How sensitive is the algorithm to the assumption that reward and resource are fully observed before the admission decision?

**Limitations:**

The paper builds its analysis around a fluid benchmark and the assumption that reward and resource are observed before making the admission decision. These are reasonable assumptions for the setting the authors have in mind, but they also narrow the scope of where the results apply. I am also a bit concerned about the need to solve a saddle-point problem in every round, since this could become expensive in larger systems.

**Strengths And Weaknesses:**

I think the structural part of the paper is solid. The switching-aware oracle is well motivated and avoids the simple but weak benchmark of fixing one configuration. The max-min formulation is clear, and the link between configuration mixing and threshold-based admission is explained carefully. The dual price interpretation is easy to follow.

The regret result is also meaningful. Learning both a distribution over configurations and a dual price at the same time is not trivial, and the analysis is detailed. The rate in terms of K and T is reasonable for this type of problem.

That said, I have some concerns.

First, the benchmark is a fluid relaxation. It ignores the pathwise budget constraint and enforces feasibility only in expectation. This is common in related literature, but it does mean the regret is measured against a relaxed oracle. I would like to understand better how far this oracle can be from the true offline optimal policy that must satisfy the budget exactly.

Second, the algorithm solves a saddle-point problem in every round. The paper focuses on the regret analysis but does not discuss the computational cost very much. It is not clear how expensive this step is in practice, especially when K or the resource dimension is large.

Third, the regret bound is written mainly in terms of K and T. It would help to see how the resource dimension affects the bound and whether it introduces additional scaling issues.

Finally, the whole framework depends on observing reward and resource before making the admission decision. If this assumption is weakened, the threshold rule would no longer apply in the same way. A short discussion of how sensitive the method is to this assumption would make the scope of the contribution clearer.

Overall, I think this is a strong theory paper with a clear structure and a careful analysis. My main hesitation is about the strength of the benchmark and the computational cost of the algorithm.

---

> ### Author Rebuttal · Authors · 2026-03-29
>
> We thank the reviewer for the careful assessment. To define an offline policy that satisfies the budget constraint almost surely, we must first specify what information that offline policy is allowed to observe before making any decisions. In our setting this is subtle because the first-stage configuration decision determines which reward-resource realization is revealed. If the offline policy is allowed to observe the realizations of all the requests under all configurations, then the regret of any online policy against the total expected reward of such an offline policy can already be \\(\\Omega(T)\\). Consider two identical configurations, one resource with deterministic consumption \\(a=1\\), budget \\(B=T\\), and rewards \\(r\_{t,1},r\_{t,2}\\stackrel{i.i.d.}{\\sim}\\mathrm{Unif}[0,1]\\). Any online policy must choose one configuration before seeing its reward, so its expected total reward is at most \\(T/2\\). By contrast, a clairvoyant benchmark that first observes realizations from both configurations earns
> \\[T\\cdot \\mathbb{E}\\big[\\max\\{r\_{t,1},r\_{t,2}\\}\\big] = T\\cdot \\frac{2}{3}.\\]
> Thus the gap is \\(T/6=\\Omega(T)\\), so such a benchmark is too strong to serve as a meaningful regret comparator. On the other hand, as shown in Example 1, if the offline policy is restricted to observing realizations of requests from only one fixed configuration, then it can fail to upper-bound feasible online policies. The fact that the decision maker controls what can be observed through the first-layer configuration choice is exactly what makes designing an offline policy that satisfy budget constraint almost surely subtle in this model. A main contribution of our switching-aware benchmark is to address this difficulty. Our benchmark is sensitive to configuration switching and a valid upper bound of every feasible online policy. In addition, we want to emphasize that, although \\(T\\,V^{\\mathrm{mix}}(b)\\) is a fluid relaxation rather than the exact pathwise offline optimum that enforces the budget almost surely, the regret of our algorithm with respect to this benchmark is already sub-linear. We will include improving regret bound and designing an offline policy that both provides a valid upper bound and satisfies budget constraint almost surely as an important future extension in the revision.
>
> Computationally, the optimistic saddle-point problem is formulated as a linear program in Appendix F.1, and its size grows linearly in \\(d\\), \\(K\\), and the number of stored samples. Although the main theorem is stated for the version that re-solves this problem every round after the warm start, Appendix F.2 explains the implementation-level doubling schedule, under which we re-solve only when some \\(N\_\\theta\\) doubles. This reduces the total number of LP solves to \\(O(K\\log T)\\) while preserving the same regret order up to constant factors. Empirically the runtime is modest, about \\(0.9\\)s at \\(T=1\\text{,}000\\) and \\(5.2\\)s at \\(T=5\\text{,}000\\) on the Alibaba experiment. Regarding scaling, the resource dimension enters the regret analysis through the concentration of the \\(d\\)-dimensional price vector, which gives an explicit \\(\\widetilde O(\\sqrt{KdT})\\) dependence.
>
> Finally, the observe-then-decide assumption is central to the model, but in the intended applications the relevant quantities can often be estimated upstream. In LLM serving, for example, prompt length, predicted output length, expected KV-cache footprint, and latency/quality models under each configuration can provide an estimate of \\((r\_t,a\_t)\\) before commitment. If these quantities are estimated rather than exactly observed, the proof structure is designed to track error propagation at the level of surplus and consumption estimates, so the additional estimation error can be incorporated in the same way. We will clarify that the current theorem assumes exact revelation and that noisy upstream estimation is a natural extension.

---

> > ### Author Rebuttal · Reviewer_DCoV · 2026-03-31
> >
> > Thank you for your detailed response. I appreciate the clarification and will keep my score as is.

---

### Official Review · Reviewer_F5eC · 2026-03-09

**Soundness:** 3
**Presentation:** 3
**Significance:** 3
**Originality:** 3
**Overall Recommendation:** 4
**Confidence:** 3

**Summary:**

This paper studies an online joint configuration selection (demand model selection) and admission control (resource allocation) problem. In this problem, a configuration is first selected from a small finite set, which determines the distribution of the demand. Then the demand (resource consumption and the corresponding reward) is realized, and the algorithm decides whether to admit this demand. This model lies between the bandits with knapsack model, which only considers the configuration selection part, and the online linear programming model, which assumes that a fixed configuration is selected.

The main contribution is to propose a benchmark for this new model, called the mixing oracle. First, this oracle provides an upper bound on the expected reward of any online policy. Second, the saddle point formulation in the oracle motivates the design of an optimistic saddle point problem in a UCB-type algorithm to jointly determine the configuration selection and admission control. The proposed algorithm is shown to achieve a regret of $\tilde{O}(\sqrt{T})$. Experiments show that the proposed algorithm based on the mixing oracle outperforms the so-called OneHot selection approach that assumes a fixed configuration oracle.

**Compliance With Llm Reviewing Policy:**

Affirmed.

**Final Justification:**

The authors addressed my comments.

**Key Questions For Authors:**

- To what extent does the no-tie assumption (Assumption 4) affect the results, both theoretically and empirically? This assumption is discussed in several parts of the paper, but it is unclear to me whether it is introduced mainly for convenience in the theoretical analysis.

- Could the authors provide a more thorough discussion of the setup of the LLM serving scenarios and the rationale behind these settings? When generating the scenarios, it seems that a base reward and resource consumption are first sampled from a distribution, and then different configurations use different scaling factors to scale the samples. Would it make more sense to sample separately from different distributions (with scaled parameters)?

- Please comment on the computational complexity of the proposed algorithm.

**Limitations:**

See weakness and questions.

**Strengths And Weaknesses:**

Strength
- The joint configuration selection and admission control model appears new to me. The configuration selection provides additional flexibility to control the demand, which represents a middle ground between stationary and non-stationary models.

- The formal characterization of the mixing oracle seems novel and is shown to be effective in the design of the UCB-type algorithm, both theoretically and empirically.

Weakness
- The most interesting part of the work lies in the characterization of the mixing oracle. However, one key step of the transformation relies on the dual form, which is already known from existing work (Li et al., 2022). I understand that the paper does not claim novelty for this part, but I feel that relying on this known result somewhat weakens the overall novelty of the work.

- Although the model is theoretically interesting, the paper lacks a clear motivation for this model. In the experiments, an LLM serving scenario is used to validate the algorithm; however, the overall scenario appears somewhat artificial.

- The computational complexity of the proposed algorithm is not discussed. Solving the optimistic saddle-point problem requires solving a linear program (lines 1736–1740), which depends on the number of past demands (whether rejected or admitted). As the horizon increases, this could become a concern.

---

> ### Author Rebuttal · Authors · 2026-03-29
>
> We thank the reviewer for the careful comments. We agree that the Li--Ye dual characterization is a building block rather than the main novelty. The new difficulty begins once the configuration-selection layer is added: the first-stage choice makes arrivals endogenous, the correct benchmark becomes a switching-aware mixture oracle rather than a fixed-distribution OLP benchmark, and the algorithm must jointly learn a mixture over configurations together with a shared dual price. In particular, the fixed-configuration oracle can fail to upper-bound feasible online policies, while the envelope-support/complementarity characterization of the mixed oracle is specific to this two-stage setting.
>
> Assumption 4 is a no-tie condition: conditional on \\(a\\), the reward distribution is continuous, equivalently \\(\\Pr(r=\\langle p,a\\rangle)=0\\) for every price vector \\(p\\). This assumption is used mainly to make strict and weak threshold rules equivalent and to obtain the identity \\(\\nabla g\_\\theta(p)=-h\_\\theta(p)\\). A parallel discrete version can be derived by replacing gradients with subgradients and using randomized tie-breaking: when \\(r=\\langle p,a\\rangle\\), accept with probability \\(\\lambda\\in[0,1]\\), which yields
> \\[h\_\\theta^{\\lambda}(p) = \\mathbb{E}\_\\theta\\!\\left[ a\\Bigl( \\mathbf{1}\\{r > \\langle p,a\\rangle\\} + \\lambda\\,\\mathbf{1}\\{r=\\langle p,a\\rangle\\} \\Bigr) \\right].\\]
> The same saddle-point/KKT argument then goes through with subgradients. More precisely, \\(\\lambda\\) should be chosen so that the resulting tie-weighted expected (or empirical) consumption satisfies the KKT feasibility/complementarity condition at the minimizing price \\(p^\\star\\). In one dimension, this amounts to interpolating between the strict-threshold and weak-threshold consumptions; in multiple dimensions, the formal object is a vector of tie weights over the tie observations. Thus Assumption 4 is primarily for analytical cleanliness rather than a substantive modeling restriction.
>
> A concrete LLM-serving interpretation is as follows. At time \\(t\\), the system first chooses
> \\[\\theta\_t\\in\\{\\text{8-bit quantization},\\ \\text{4-bit quantization},\\ \\text{dynamic batching}\\}.\\]
> This first-stage choice changes the downstream reward-resource distribution \\(D\_{\\theta\_t}\\): \\(4\\)-bit quantization reduces memory usage but may reduce output quality/reward, dynamic batching increases throughput but may increase latency or CPU usage, and \\(8\\)-bit quantization acts as a balanced baseline. After the request arrives, upstream metadata such as prompt length, predicted output length, expected KV-cache footprint, and latency/QoS predictions under the chosen configuration determine the realized or predicted pair \\((r\_t,a\_t)\\). The scheduler then accepts the request if \\(r\_t > \\langle p\_t,a\_t\\rangle\\) and the remaining capacity is sufficient. This is exactly the two-stage control captured by our model. For the Alibaba experiment, we use a shared base trace together with configuration-specific multipliers because the dataset does not provide the same job executed under multiple configurations; this preserves a job-level counterfactual coupling across configurations.
>
> Computationally, the optimistic saddle-point problem is formulated as a linear program in Appendix F.1, whose size grows linearly in the resource dimension \\(d\\), the number of configurations \\(K\\), and the number of stored samples. The main-text Algorithm 1 recomputes the optimistic saddle point at every round after the warm start, whereas Appendix F.2 describes the implementation-level doubling schedule: we re-solve only when some \\(N\_\\theta\\) doubles, so the total number of saddle solves is \\(O(K\\log T)\\). This preserves the same \\(\\widetilde O(\\sqrt{KT})\\) regret order up to constants while substantially reducing computational overhead. Empirically, SP-UCB--OLP takes approximately \\(0.9\\)s at \\(T=1\\text{,}000\\) and \\(5.2\\)s at \\(T=5\\text{,}000\\) on the Alibaba trace, which is consistent with near-linear scaling in \\(T\\).

---

> > ### Author Rebuttal · Reviewer_F5eC · 2026-04-03
> >
> > Thanks for the detailed response. I will keep my rating.

---

### Official Review · Reviewer_NB3D · 2026-03-11

**Soundness:** 3
**Presentation:** 2
**Significance:** 3
**Originality:** 3
**Overall Recommendation:** 4
**Confidence:** 4

**Summary:**

This paper studies a two-layer online decision problem where an agent first selects a configuration inducing a reward–resource distribution, then makes admission decisions under multi-resource budgets. The authors introduce a switching-aware fluid oracle that permits configuration mixing, derive a primal–dual/saddle-point characterization of this oracle, and design SP-UCB–OLP, an optimistic saddle-point algorithm achieving $\tilde{O}(\alpha\sqrt{KT})$ regret.

**Compliance With Llm Reviewing Policy:**

Affirmed.

**Final Justification:**

I am satisfied with the rebuttal.

**Key Questions For Authors:**

- How does your approach relate (in guarantees or benchmarks) to the online stochastic packing literature, beyond prophet inequalities?
- In the complementarity experiment (Section 6.3), what is the intuition for why the gap between algorithms and the mixed-fluid oracle varies across algorithms? Does this suggest room for algorithmic improvement?

**Limitations:**

yes

**Strengths And Weaknesses:**

Strengths
- The switching-aware fluid oracle is technically novel and well-motivated. The counterexample (Example 1) concretely shows the fixed-configuration oracle can be loose, and the complementarity experiment (Table 2, S4) demonstrates the gap can approach 2 times. The mixed oracle provides a valid upper bound (Theorem 8).
- The saddle-point formulation (Theorem 5) is elegant, with KKT-style characterizations (Theorems 6–7) that naturally connect threshold admission rules to configuration mixing via envelope support, feasibility, and complementarity.
- Experiments are rigorous: empirical regret scales as $\tilde{O}(\sqrt{T})$ (Figure 2), complementarity experiments validate the benchmark (Table 2), and ablations (Table 3) isolate contributions of global dual pricing, mixture sampling, and other components.
- The framework addresses a practically relevant class of problems (LLM serving, cluster scheduling, revenue management) lying between BwK and OLP, offering a unifying benchmark and methodology.

Weaknesses

- Several presentation inconsistencies reduce readability. For example, Appendix E is titled "Regret Proof for Theorem 3" but the corresponding main-text theorem is Theorem 9, and the proof of Theorem 8 (Appendix C.6) references "Theorem 1" instead of Theorem 5. Also, Algorithm 1 (main text) uses strict inequality $r_t > \langle p_t, a_t \rangle$ while Algorithm 2 (Appendix B) uses $\geq$; the equivalence under the No-Tie assumption is clarified only in the appendix, and a brief note in the main text would help.
- The related work section covers OLP and prophet inequalities but could be expanded to discuss the broader online stochastic packing literature, where observe-then-decide thresholds and LP-based benchmarks are also central. Positioning the switching-aware oracle relative to these existing benchmarks would strengthen the contribution.
- Minor typo: Section 4.1 defines $\theta^* = \arg\min_{\theta} V^{\mathrm{off}}_\theta(b)$, but the surrounding text explicitly states "the one whose reward-resource distribution gives the highest optimal value," so the intended definition is $\arg\max$.

---

> ### Author Rebuttal · Authors · 2026-03-29
>
> We thank the reviewer for asking for sharper positioning relative to the broader online stochastic packing literature. Our model connects to both OLP and BwK through the switching-aware oracle. As already noted in Section 4.2 (current lines 165--175), when \\(K=1\\), the model reduces to OLP and \\(V^{\\mathrm{mix}}(b)\\) coincides with the standard fluid relaxation; when the admission rules \\(\\{x\_\\theta\\}\_{\\theta\\in\\Theta}\\) are fixed ex ante, the model becomes a BwK-type LP relaxation. Compared with OLP / online-stochastic-packing, the new ingredient is that configuration choice makes arrivals endogenous. Example 1 highlights why a single-distribution OLP / online-stochastic-packing benchmark is insufficient once the arrival distribution itself is controlled by the first-stage configuration choice. Compared with BwK, the new ingredient is that a request can be rejected after we observe its reward and resource consumption. Because the first-stage decision changes the distribution of the second-stage request, neither a single-distribution OLP benchmark nor a fixed-configuration BwK benchmark is sufficient. This is why the correct oracle takes the form
> \\[V^{\\mathrm{mix}}(b) = \\max\_{w\\in\\Delta\_K}\\min\_{p\\in P} \\left\\{ \\langle p,b\\rangle+\\sum\_{\\theta\\in\\Theta} w\_\\theta g\_\\theta(p) \\right\\},\\]
> and why the algorithm must jointly learn both the mixture \\(w\\) and the global price vector \\(p\\). We will expand the related-work discussion accordingly, and we will also correct the typos in the revision.
>
> The complementarity experiment in Section 6.3 is intended to show precisely why this distinction matters. On scenario S4, the gap
> \\[\\frac{T\\,V^{\\mathrm{mix}}(b)}{V^{\\ast}(b)}\\]
> is close to \\(2\\). Empirically, several online algorithms achieve \\(\\mathrm{CR}^{\\ast} > 1\\), i.e., they outperform the fixed-configuration candidate oracle, while their performance relative to the switching-aware oracle remains below \\(1\\), typically in the range \\(0.5\\) to \\(0.95\\). This confirms that \\(V^{\\ast}(b)\\) is not a true upper bound, whereas \\(V^{\\mathrm{mix}}(b)\\) is. The significance of the gap is therefore conceptual as well as quantitative: it identifies the correct benchmark for the framework. Without a valid upper bound, one can observe negative regret or competitive ratios above \\(1\\), which makes algorithmic comparisons misleading.
>
> Regarding the variation of the gap between algorithms and the mixed-fluid oracle across algorithms, the bad performance of OneHot Algorithm is due to the fact that OneHot is restricted to a single configuration and cannot exploit complementary resource profiles. For the other three algorithms, although Greedy and Random perform slightly better than SP-UCB-OLP, we do not believe this implies that there is room for algorithmic improvement. It is just the design of Greedy and Random makes them fit very well to scenario S4. In other scenarios, SP-UCB-OLP performs better than them.
>
> We also provide here a concrete illustration of the difference between BwK and our problem by modifying Example 1 to \\(b=[0.25,0.25]\\). Suppose there are two configurations, both with reward \\(r\\sim \\mathrm{Unif}[0,2]\\), while configuration \\(1\\) always uses resource vector \\([1,0]\\) and configuration \\(2\\) always uses \\([0,1]\\). Fix \\(T\\). In BwK, since a selected request must be admitted, the best policy is to choose each configuration \\(T/4\\) times, which yields expected total reward \\(T/2\\). In our model, the switching-aware oracle uses the mixture \\(w=(1/2,1/2)\\) and threshold \\(p=(1,1)\\). Indeed, under \\(r\\sim \\mathrm{Unif}[0,2]\\), we have \\(\\mathbb{P}(r > 1)=1/2\\) and \\(\\mathbb{E}[r\\,\\mathbf{1}\\{r > 1\\}]=3/4\\), so the expected consumption rate of each resource is exactly \\(1/4\\), while the per-period value is \\(V^{\\mathrm{mix}}(b)=0.75\\). Therefore, by Theorem 9, SP-UCB--OLP achieves expected total reward
> \\[0.75T-\\widetilde O(\\sqrt{KT})=0.75T-\\widetilde O(\\sqrt{T}),\\]
> since \\(K=2\\) is fixed. Hence the value of being able to reject after observing \\((r\_t,a\_t)\\) is \\(\\Omega(T)\\), not a lower-order effect.

---

> > ### Author Rebuttal · Reviewer_NB3D · 2026-04-03
> >
> > I am satisfied with the rebuttal.

---

### Official Review · Reviewer_yJdj · 2026-03-13

**Soundness:** 3
**Presentation:** 4
**Significance:** 2
**Originality:** 4
**Overall Recommendation:** 4
**Confidence:** 2

**Summary:**

The paper presents a variation of the multi-armed bandits with knapsacks problem where, once the algorithm discovers the outcome of a given option, it has the additional possibility to accept or reject the reward and the costs.
The paper has two main theoretical contributions.
First, it presents a new oracle which is constrained by the knapsack constraints only in expectation, leaving the possibility that the oracle suggests impossible executions; it shows that this oracle dominates any possible strategy, making it a valid choice for regret optimization.
Second, it introduces an algorithm and show its regret against the proposed oracle is upper-bounded by a function of the square root of the number of time steps T and the number of configurations K.

**Compliance With Llm Reviewing Policy:**

Affirmed.

**Final Justification:**

Thanks for your rebuttal, it addresses my comments.

**Key Questions For Authors:**

I find the paper lacks comparison with the closest related work Bandits with Knapsacks.
For a non-expert point of view, the change in model is quite small, only adding the possibility that the algorithm rejects the observed random outcomes.
However, the paper suggests this change in model justifies 1) a change of oracle (while the Bandits with Knapsacks paper sticked with the simple, best possible strategy) and 2) a regret which now depends on the number of rounds T (which is not the case in BwK and seems way worse).
Could you justify why those changes are needed or point me towards an example where this becomes clearer?
I am ready to increase my grade then.

**Limitations:**

Yes

**Strengths And Weaknesses:**

Soundness:
The results look solid, I did not dig so much.

Presentation:
The paper is very well presented.
I would recommend to mention earlier the fact that the proposed switching-aware oracle dominates any feasible strategy (Theorem 8).

Significance:
The paper's significance is hard to judge for me, as I could not compare with the closest related work Bandits with Knapsacks.
In particular, the regret of this paper seems way worse the one in Bandits with Knapsacks, as it depends on the number of steps.
Without a clear reason why this is the case, I cannot call the innovations of this paper significant ones.

Originality:
The paper brings together two fields and merge them into a model that seems coherent. In particular, the fact that the introduced oracle dominates all correct strategies may start an interesting line of work with an oracle that is simpler to work with.

---

> ### Author Rebuttal · Authors · 2026-03-29
>
> We thank the reviewer for asking why the gap from Bandits with Knapsacks (BwK) is substantial enough to require both a new oracle and a new algorithm. The distinction is structural instead of incremental. In BwK, once an arm/configuration is selected, the request is committed and resources are consumed immediately. In our model, each round has two sequential decisions: first choose a configuration \\(\\theta\_t\\), which determines the downstream distribution \\(D\_{\\theta\_t}\\); then, after observing \\((r\_t,a\_t)\\sim D\_{\\theta\_t}\\), decide whether to admit.
>
> A concrete illustration is obtained by modifying Example 1 to \\(b=[0.25,0.25]\\). Suppose there are two configurations, both with reward \\(r\\sim \\mathrm{Unif}[0,2]\\), while configuration \\(1\\) always uses resource vector \\([1,0]\\) and configuration \\(2\\) always uses \\([0,1]\\). Fix \\(T\\). In BwK, since a selected request must be admitted, the best policy is to choose each configuration \\(T/4\\) times, which yields expected total reward \\(T/2\\). In our model, the switching-aware oracle uses the mixture \\(w=(1/2,1/2)\\) and threshold \\(p=(1,1)\\). Indeed, under \\(r\\sim \\mathrm{Unif}[0,2]\\), we have \\(\\mathbb{P}(r > 1)=1/2\\) and \\(\\mathbb{E}[r\\,\\mathbf{1}\\{r > 1\\}]=3/4\\), so the expected consumption rate of each resource is exactly \\(1/4\\), while the per-period value is \\(V^{\\mathrm{mix}}(b)=0.75\\). Therefore, by Theorem 9, SP-UCB--OLP achieves expected total reward
> \\[0.75T-\\widetilde O(\\sqrt{KT})=0.75T-\\widetilde O(\\sqrt{T}),\\]
> since \\(K=2\\) is fixed. Hence the value of being able to reject after observing \\((r\_t,a\_t)\\) is \\(\\Omega(T)\\), not a lower-order effect. To exploit this gain, one must learn more than arm means: the algorithm must estimate the full surplus family
> \\[g\_\\theta(p)=\\mathbb{E}\_\\theta\\!\\left[(r-\\langle p,a\\rangle)\_+\\right]\\]
> and jointly solve for a mixture \\(w\\in\\Delta\_K\\) and a shared dual price \\(p\\in\\mathbb{R}^d\_+\\). This is exactly why a new benchmark and a new algorithmic design are needed.
>
> Relative to standard OLP, the additional difficulty is that arrivals are endogenous: the first-stage choice changes the distribution of the second-stage observation. These two features---observe-then-decide admission and endogenous arrivals---are precisely why both the benchmark and the learning problem change.
>
> Finally, dependence on the horizon \\(T\\) is standard in the large-budget regime \\(B=T b\\) adopted in Assumption 3; under this scaling, the benchmark itself is \\(T\\,V^{\\mathrm{mix}}(b)\\). Studying how regret grows with \\(T\\) is common in both BwK and OLP; see, e.g., Agrawal et al. 2016 and Li and Ye 2022. Our guarantee is sublinear,
> \\[\\operatorname{Reg}^{\\mathrm{mix}}\_{\\pi}(T)=\\widetilde O(\\sqrt{KT}),\\]
> so the average regret vanishes as \\(T\\to\\infty\\).
>
> References
>
> [1] Shipra Agrawal, Nikhil R. Devanur, and Lihong Li. An efficient algorithm for contextual bandits with knapsacks, and an extension to concave objectives. In Proceedings of the 29th Annual Conference on Learning Theory (COLT), volume 49 of PMLR, pages 4–18, 2016.
>
> [2] Xiaocheng Li and Yinyu Ye. Online linear programming: Dual convergence, new algorithms, and regret bounds. Operations Research, 70(5):2948–2966, 2022. arXiv:1909.05499.

---

> > ### Author Rebuttal · Reviewer_yJdj · 2026-04-03
> >
> > Thanks for your rebuttal, it addresses my comments.

---

### Decision · Program_Chairs · 2026-04-30

**Decision:**

Accept (regular)

**Comment:**

The four reviewers agree that this is a technically solid and original paper, and they all recommend weak acceptance. I also support acceptance. The paper introduces a novel two-layer formulation for joint online configuration selection and admission control, together with a switching-aware fluid oracle and a principled saddle-point-based algorithmic framework. In my view, these are interesting contributions that can be of value to the community.

The reviewers raised several important points, including the need for clearer positioning with respect to related work (especially BwK, OLP, and stochastic packing), a sharper discussion of the benchmark and computational aspects, and some presentation issues and minor inconsistencies. However, the authors’ rebuttal addressed these concerns satisfactorily, and the reviewers explicitly acknowledged that their main questions were resolved. Overall, I believe the remaining issues are manageable through a minor revision aimed primarily at clarifying the motivation, positioning, and presentation of the contributions, rather than requiring substantial technical changes. Said that, since the paper presents several weaknesses, I consider it as a borderline paper.